# Future Sea Level Contribution from Antarctica inferred from CMIP5 Model Forcing and its Dependence on Precipitation Ansatz

Christian B. Rodehacke[1,2], Madlene Pfeiffer[1], Tido Semmler[1], Özgür Gurses[1], and Thomas Kleiner[1]

[1]Alfred Wegener Institute Helmholtz Centre for Polar and Marine Research, D-27570 Bremerhaven, Germany
[2]Danish Meteorological Institute, DK-2100 Copenhagen Ø, Denmark

**Correspondence:** Christian Rodehacke (christian.rodehacke@awi.de)

**Abstract.** Various observational estimates indicate growing mass loss at Antarctica's margins but also heavier precipitation across the continent. Simulated future projections reveal that heavier precipitation, fallen on Antarctica, may counteract amplified iceberg discharge and increased basal melting of floating ice shelves driven by a warming ocean. Here, we test how the ansatz (implementation in a mathematical framework) of the precipitation boundary condition shapes Antarctica's sea-level contribution in an ensemble of ice-sheet simulations. We test two precipitation conditions. We either apply the precipitation anomalies coming from CMIP5 models directly or scale the precipitation by the air temperature anomalies from the CMIP5 models. In the scaling approach, it is common to use a relative precipitation increment per degree warming as an invariant scaling constant. From nine CMIP5 models, we use future climate projections, ranging from strong mitigation efforts to business-as-usual, to perform simulations from 1850 to 5000. We take advantage of individual climate projections by exploiting its full temporal and spatial structure. The CMIP5 projections beyond 2100 are prolonged with reiterated forcing that includes decadal variability; hence, our study may underestimate ice loss past 2100. In contrast to various former studies that apply an evolving temporal forcing averaged spatially across the entire Antarctic Ice Sheet, our simulations consider the spatial structure in the forcing coined by various climate patterns. This fundamental difference reproduces regions of decreasing precipitation despite general warming. Regardless of the applied boundary and forcing conditions, our ensemble study suggests that some areas will lose ice in the future, such as the glaciers from the West Antarctic Ice Sheet draining into the Amundsen Sea. In general, the simulated ice-sheet thickness grows along the coast, where incoming storms deliver topographically controlled precipitation. There the ice thickness differences are largest between the applied precipitation methods. On average, Antarctica shrinks for all future scenarios if the air temperature anomalies scale the precipitation. In contrast, Antarctica gains mass in our simulations if we apply the simulated precipitation anomalies directly. The analysis reveals that the mean scaling inferred from climate models is larger than the commonly used values deduced from ice cores; besides, it varies spatially: Highest scaling across the East Antarctic Ice Sheet and lowest scaling around the Siple Coast east of the Ross Ice Shelf. The discrepancies in response to both precipitation ansatzes illustrate the principal uncertainty in projections of Antarctica's sea-level contribution.

**Plain Language Summary** In the warmer future, the ice sheet of Antarctica will lose more ice at the margin, because more icebergs may calve and the warming ocean melts more floating ice shelves from below. However, the hydrological cycle is also stronger in a warmer world. As a consequence, more snowfall precipitates on Antarctica, which may balance the amplified marginal ice loss. In this study, we have used future climate scenarios from various global climate models to perform numerous

ice-sheet simulations. These simulations represent the Antarctic Ice Sheet. We analyze whether Antarctica will grow or shrink. In all our simulations, we find that certain areas will lose ice under all circumstances. However, depending on the method used to describe the precipitation reaching Antarctica in our simulations, parts of the Antarctic Ice Sheet may either grow or shrink in the future. The discrepancy of the simulation results between both methods describing the precipitation illustrates the uncertainty of the possible range of future precipitation growth in a warming atmosphere. Furthermore, the dissimilarity is pronounced differently between the West Antarctic and East Antarctic Ice Sheet. Since we use only the available climate scenarios until the year 2100, any additional warming after 2100 may turn the ice gain into an ice loss under a strongly changing climate.

## 1  Introduction

Sea level rise as a symptom of the progressing climate warming is of paramount importance for coastal societies because it impacts numerous economic activities globally and threatens the population along coasts. An ice Sheet's contribution to the future sea level is projected by statistical approaches that take advantage of the deduced past behavior (Church et al., 2013a) or process-based model simulations, e.g. ice-sheet models (Goelzer et al., 2018; Seroussi et al., 2019a). Adequate forcing fields are required to perform ice-sheet model simulations covering centuries to glacial-interglacial (100,000 years) periods (e.g., Golledge et al., 2015; Winkelmann et al., 2012; Pollard and DeConto, 2009). Those forcing fields are either descriptions based on linear multiple-regression analysis (e.g., surface elevation and latitude dependence (Fortuin and Oerlemans, 1990)) or originate from regional climate models or climatological data sets.

It is common for changing climate experiments to deduce simplified forcing anomalies across selected climate scenarios, which come from CMIP models, for instance. As a further step of simplification, often, the anomalies forced through time are spatially homogeneous in these experiments. For example, a spatial homogeneous air temperature anomaly is applied across the entire Antarctic Ice Sheet on top of a background field representing the presently observed state. Compared to air temperature, the evolution of precipitation is even more uncertain in both observational records (Hartmann et al., 2013) and models (Flato et al., 2013). Therefore, the precipitation forcing anomalies are commonly developed from the temperature forcing anomalies in prescribing a percentage increase of precipitation with temperature increase. The motivation behind this is the Clausius-Clapeyron process, where the saturation pressure of water vapor scales exponentially by about $7\,\%$ per Kelvin warming (Held and Soden, 2006) — it is implicitly assumed that the relative humidity does not change.

However, when considered globally, this rate — hereafter, referred to as mean precipitation scaling in the following — is less than the theoretical value deduced from thermodynamic principles. Climate modeling studies representing the Last Glacial Maximum (LGM), the pre-industrial (piControl), and historical period as well as climate warming scenarios (1pctCO2, abrupt4xCO2) show that the global precipitation increases in warmer climates and decreases in colder climates with a rate of $1\,\%\,\mathrm{K}^{-1}$ to $4\,\%\,\mathrm{K}^{-1}$ (Held and Soden, 2006; Li et al., 2013).

Decreasing precipitation rates with global warming in the dry subtropics (Sun et al., 2007), covering a substantial part of the globe, exposes a limitation of this scaling. It indicates that the assumption of a homogeneous increase of precipitation with

global warming is not expected; for instance, future simulations of the 21st century show that the scaling in the Arctic with $4.5\,\%\,\mathrm{K}^{-1}$ is much larger than the global value of $1.6\,\%\,\mathrm{K}^{-1}$ – $1.9\,\%\,\mathrm{K}^{-1}$ because retreating sea ice amplifies the hydrological cycle in the Arctic (Bintanja and Selten, 2014).

Both dynamical and thermodynamical processes contribute to the actual precipitation change, even if dynamical changes, such as an altering circulation, play a secondary role globally (Emori and Brown, 2005). The balance of radiative fluxes in/out of the troposphere and the latent energy flux at the surface limits evaporation, which restricts water vapor supply regionally and therefore limits the scaling (Allen and Ingram, 2002).

A global analysis of observed precipitation and air temperature changes reveals a low or even negative scaling in tropical land regions driven by decreasing soil moisture, a near Clausius-Clapeyron scaling $\approx 7\%\,\mathrm{K}^{-1}$ over the open ocean, and a super Clausius-Clapeyron scaling ($>7\,\%\,\mathrm{K}^{-1}$) along extratropical coasts (Yin et al., 2018). In the latter case, the moisture supply by the ocean in concert with the atmospheric circulation generates extreme precipitation events inland. These events cause a high temperature scaling factor for precipitation onshore. Ultimately, the local availability and recycling of moisture and the atmospheric dynamics determine the size of the precipitation-temperature scaling (Yin et al., 2018). Since the interplay between thermodynamic and atmospheric dynamics governs the scaling, it is unlikely to represent this scaling by a single value across Antarctica.

To overcome some of the limitations of forcing strategies of ice sheet models in previous studies, we exploit the full temporal and spatial pattern of the atmospheric and oceanographic forcing anomalies from CMIP5 models to perform transient simulations; this approach is unprecedented. We use the historical climate scenario (1850–2004) followed by three future climate scenarios (RCP2.6, RCP4.5, RCP8.5; 2005–2100) from a compilation of nine CMIP5 models (Table 1) to drive for each climate projection one simulation with the Parallel Ice Sheet Model (PISM, e.g., Bueler and Brown, 2009; Winkelmann et al., 2011). We compare these ice sheet simulations considering the spatial inhomogeneities in transient climate forcing with more traditional simulations in which the precipitation forcing anomaly is scaled with the temperature forcing anomaly.

The following subsection provides an overview of observed and simulated precipitation changes over Antarctica from previous studies. Following, is the description of factors influencing the precipitation and its scaling in Antarctica. Afterward, a subsection highlights processes that control Antarctica's mass balance.

## 1.1 Processes Linked to Precipitation Scaling

While in most regions both the local availability and recycling of moisture and the atmospheric dynamics determine the size of the precipitation-temperature scaling (Yin et al., 2018), atmospheric dynamics dominate over the deep-frozen interior Antarctic continent, probably due to negligible water buffering capacity of the frozen ground. The ocean surface conditions around Antarctica set the lower boundary condition for the atmosphere, which accounts for the spread of the precipitation scaling among climate models. Atmosphere simulations over Antarctica, which are driven by boundary conditions from a small ensemble of historical and future climate scenarios, show a weak impact of changed atmospheric conditions or enhanced radiative forcing on the scaling factor (Krinner et al., 2014). In contrast, the ocean conditions are crucial for the precipitation scaling in Antarctica (Krinner et al., 2014) because those conditions shape the atmospheric circulation, determining the moisture flux

that maintains the precipitation (Wang et al., 2020). Bracegirdle et al. (2015) show that the sea ice cover has a decisive impact and that the mean historical sea ice concentration is more important than the sea ice retreat rate.

Across Antarctica, the patterns of increasing as well as decreasing precipitation are consistent with the variability of the large-scale moisture transport resembling the known Southern Hemispheric modes of variability, such as Amundsen Sea Low (ASL) or Southern Annular Mode (SAM) (Fyke et al., 2017). Also, the baroclinic annular mode (BAM) and the two Pacific-South American teleconnections (PSA1 and PSA2) indices influence precipitation over Antarctica (Marshall et al., 2017). An enhanced baroclinic annular mode, which corresponds to high storm amplitudes, increases the precipitation over the coastal East Antarctic Ice Sheet, while an enhanced SAM causes stronger precipitation across the West Antarctic Ice Sheet (WAIS) and the neighboring Antarctic Peninsula. The two Pacific-South American teleconnections impact mainly precipitation over the West Antarctic Ice Sheet beside other regions across the Antarctic continent.

In reanalysis products, no robust or statistically significant precipitation trend exists over Antarctica (Bromwich et al., 2011). This result is in agreement with precipitation observations over the Southern Ocean (Bromwich et al., 2011). However, shallow ice cores across Antarctica reveal a tendency for a positive precipitation trend over the last 50 years and 100 years. Since 1800, the increase of the surface mass balance (SMB) is estimated to be $7 \pm 1.3\,\mathrm{Gt\,decade}^{-1}$ (Thomas et al., 2017).

In contrast, over the western region of the West Antarctic Ice Sheet (WAIS), next to the Ross Ice Shelf, a negative snow accumulation trend has been detected in monthly reanalysis products (ERA-Interim: 1979–2010 and ERA-20C: 1900–2010), which is confirmed by a composite of 17 firn cores (Wang et al., 2017). The flow of available atmospheric moisture, which feeds the precipitation across the WAIS, is dominated by the Amundsen Sea Low (Thomas et al., 2017). A location shift of the Amundsen Sea Low (ASL), expressed by its longitudinal position, exposes different regions to the inland-directed circulation branch of moisture-rich air masses — on the eastern side of the Low's center — or isolates them from moisture supply due to the offshore directed circulation branch — on the western side. Furthermore, the deepening of the ASL enhances the cyclonic circulation, which strengthens precipitation — southeast of the Low's center — over the Antarctic Peninsula and eastern WAIS. However, less moisture-rich air masses reach the western WAIS, which ultimately leads to an accumulation deficit. Enlarged sea ice extent in the Ross Sea (Haumann et al., 2016; Liu, 2004) damps evaporation to the atmosphere. In contrast, a decreasing sea ice trend in the Amundsen Sea and Bellingshausen Sea (Haumann et al., 2016; Jacobs, 2006) enhances the moisture supply. To conclude, across the West Antarctic Ice Sheet, the observed accumulation reduction is driven by the deepening of the ASL and reinforced by a more extensive sea ice extent in the Ross Sea (Wang et al., 2017).

## 1.2 Scaling between Precipitation and Air Temperature Changes over Antarctica

In Antarctica (Figure 1), global model simulations until the end of the century show an average scaling of about $7.4\,\%\,\mathrm{K}^{-1}$ (range: $5.5\,\%\,\mathrm{K}^{-1}$ – $24.5\,\%\,\mathrm{K}^{-1}$, Palerme et al., 2017), which is in agreement with CloudSat estimates (years 2007 – 2010, $7.1\,\%\,\mathrm{K}^{-1}$, Palerme et al., 2017). For the last deglaciation, global climate models suggest a value of about $6\,\%\,\mathrm{K}^{-1}$ in Antarctica, while future projections of a high-resolution regional climate model show a lower value of $4.9\,\%\,\mathrm{K}^{-1}$ in contrast to a value of $6.1 \pm 2.6\,\%\,\mathrm{K}^{-1}$ coming from an ensemble of global climate system models (Frieler et al., 2015) — which is similar to the value from global climate models for the last deglaciation. Ice core data covering 10,000 years of marked temperature changes

reveal a value of $5 \pm 1\,\%\,\mathrm{K}^{-1}$ (Frieler et al., 2015). In standalone ice-sheet modeling studies, the commonly used temperature scaling factor for precipitation amounts to approximately $5\,\%\,\mathrm{K}^{-1}$ (e.g. Gregory and Huybrechts, 2006) in Antarctica, as the latitudinal relation obtained from a CMIP5 model ensemble suggests (Golledge et al., 2015). We consider $5\,\%\,\mathrm{K}^{-1}$ as the reference value from this point on.

## 1.3 Mass Balance of Antarctica

Processes governing the balance between mass gain and mass loss determine if Antarctica contributes to a rising sea level. Antarctica's surface mass balance controls mass gain, while mass loss occurs predominantly by ocean-driven basal melting of ice shelves and iceberg calving in concert with dynamical grounding line migration (Wingham et al., 2018). For individual ice shelves, the fraction between basal melting and iceberg calving ranges from $10\,\%$ to $90\,\%$ in the period 1995–2009 (Depoorter et al., 2013). Estimates about the total mass loss agree within the uncertainties, even if they range from $2200\,\mathrm{Gt\,year}^{-1}$ to $2800\,\mathrm{Gt\,year}^{-1}$ (Depoorter et al., 2013; Liu et al., 2015; Rignot et al., 2013). However, they differ on the relative contribution between basal melting and calving. Past studies suggest that the overall mass loss is either driven by a nearly equal share between calving ($1321 \pm 144\,\mathrm{Gt\,year}^{-1}$) and basal melting ($1454 \pm 174\,\mathrm{Gt\,year}^{-1}$) (Depoorter et al., 2013), or the basal melting ($1516 \pm 106\,\mathrm{Gt\,year}^{-1}$) contribution is twice as much as the calving ($755 \pm 24\,\mathrm{Gt\,year}^{-1}$) contribution (Liu et al., 2015). The surface mass balance is the difference between mass gain by precipitation — and here predominantly snowfall — and surface meltwater that runs off, because it is not refrozen nor retained in the snowpack. Surface melt ponds (Kingslake et al., 2017) and runoff exist on Antarctic ice shelves (Bell et al., 2017), but their contribution to the total mass balance is considered to be negligible (Van Wessem et al., 2014), except for the (northern) Antarctic Peninsula (Adusumilli et al., 2018).

The focus of this paper is to identify common features of an ensemble of ice-sheet simulations forced by a multimodel forcing data set. After the discussion on the temporal and spatial evolution of the climatic boundary conditions from nine CMIP5 models, we diagnose the temperature scaling of the precipitation of these climate models. Afterwards, we investigate how the deduced scaling impacts the simulated ice-sheet thickness in contrast to spatially homogeneous scaling, e.g. inferred from ice core data. Before we discuss our results and conclude, we estimate differences in Antarctica's sea-level contribution for the variety of applied forcing and precipitation boundary conditions. Specific aspects of the work are compiled in the appendix.

## 2 Material and Methods

The full temporally and spatially varying forcings are obtained from a compilation of CMIP5 models representing a suite of climate scenarios. These climate forcings drive the Parallel Ice Sheet Model (PISM) in order to estimate Antarctica's future sea-level contribution. Here, we test our hypothesis if the ansatz (implementation in a mathematical framework) of the precipitation determines whether the global sea level rises or falls. We consider two precipitation boundary conditions. (1) We utilize both the ocean and air temperature anomalies and the precipitation anomalies from CMIP5 models on top of the reference background distributions (see Table 2) that were used to drive the ice-sheet model during spin-up. (2) We take only the ocean and air

temperature anomalies from CMIP5 models and compute the precipitation anomalies scaled by the air temperature anomalies. Also, the second set of anomalies is added to the reference fields (see Table 2). The second approach is commonly used, in particular, in paleo applications (e.g., Applegate et al., 2012; Bakker et al., 2016; de Boer et al., 2013), while some sensitivity studies keep the surface mass balance constant (Feldmann and Levermann, 2015; Hughes et al., 2017). According to these pure thermodynamical considerations, negative temperature scaling is unexpected (Frieler et al., 2012); however, in reality, atmospheric dynamics may dominate in certain regions, questioning the usage of constant scaling across Antarctica.

## 2.1 CMIP5 Forcing Data Set to Drive Ice Sheet Simulations

Nine CMIP5 models deliver the following climate scenarios (see Table 1, Taylor et al. (2012)): control run under pre-industrial conditions (piControl), the historical period (1850–2004), as well as RCP2.6, RCP4.5, and RCP8.5 (2005–2100) Vuuren et al. (2011)). These models stem from different model families (Knutti et al., 2013) and cover the range of current atmospheric (Agosta et al., 2015) and oceanographic (Sallée et al., 2013a) model uncertainties, although model deficiencies such as insufficient resolution can exist across all models. The transient forcing from 1850 until 2100 comprises the historical and scenario periods. Beyond 2100, the last 30 years (2071–2100) are repeated until the model year 5000. This procedure produces a chain of 30 year long climate forcing periods that inherits the climate variability of this period — an alternative approach to repeat the last year (2100) would not contain any decadal variability.

The length of the control climate simulation (piControl) depends on the model and varies between five and ten centuries (Table 1). From this control simulation, we extract the first or the last 50 years of the available forcing. Since these two periods are subject to variation of the long-term variability besides a potential long-term drift (identified by comparing both periods) during the control run, these two 50 year periods are generally slightly different. Therefore, anomaly forcing differs if it is computed relative to the first or last 50 years of control run. Hence, this procedure doubles the data set size of anomaly forcing. In the following, the first 50 years act as our reference.

The repetition of the last 30 years of climate forcing beyond the year 2100 is a simplification, which is not entirely consistent with the applied climate scenarios. An ongoing growing atmospheric greenhouse concentration would be expected to trigger changes in the climate system. While the atmospheric radiation reacts immediately, the redistribution of the accompanied heating within the global ocean is much slower (Hansen et al., 2011). This delay is critical because most of the additional heat ends up in the worldwide ocean (Church et al., 2011, 2013b). Consequently, further warming is inevitable after the cessation of greenhouse gas emissions (Hansen et al., 2005). Note well, our simulations do not reflect this ongoing warming. Also, over longer timescales, some feedbacks are not captured by our simulations. For instance, a disintegrating Greenland Ice Sheet will raise the global sea level. As a consequence of Greenland's reduced gravitational pull (Whitehouse, 2018), the sea level rise is particularly pronounced around Antarctica (Mitrovica et al., 2001) by this "remote" effect of gravitation. This rising sea level potentially migrates the grounding lines inshore, which ultimately destabilizes ice shelves and causes a more vulnerable Antarctic Ice Sheet. On the other hand, locally the gravitational effect may buttress Antarctica, whether Antarctica's ice loss is slow enough (Gomez et al., 2010) and Greenland stabilizes. However, the ocean's ongoing thermal expansion is currently the dominant driver behind the rising sea level (Rietbroek et al., 2016). This rise will likely destabilize Antarctica. Therefore, since

only 21st century climate conditions are used to force the ensemble after 2100, our ensemble of ice sheet simulations beyond
this year should not be considered a projection.

Atmospheric and oceanic forcing is applied as annual mean forcing on top of the forcing used to spin-up the ice-sheet model (Table 2). Since CMIP5 models do not resolve ice shelves, ocean temperatures are extrapolated horizontally into the ice shelves to mimic isopycnical flow: The operator "fillmiss2" of the Climate Data Operators' (cdo) tool kit acts on the original CMIP5 ocean grid. To allow for surface melting under a warming climate, the surface mass balance (SMB) is calculated following the positive degree day (PDD) approach (Braithwaite, 1995; Hock, 2005; Ohmura, 2001) as implemented in the PISM model (The PISM Authors, 2015a, b). Section 2.3: "Surface Mass Balance" describes the PDD setup.

## 2.2 Parallel Ice Sheet Model

The ice-sheet model PISM — based on version 0.7 — runs on a 16 km equidistant polar stereographic grid and it utilizes a hybrid system combining the Shallow Ice Approximation (SIA) and Shallow Shelf Approximation (SSA). The model employs a generalized version of the viscoelastic Lingle-Clark bedrock deformation model (Bueler et al., 2007; Lingle and Clark, 1985). In our simulations, only the viscous part has been used because of known implementation flaws in the elastic part in our and later PISM versions. The basal resistance is described as plastic till by a Mohr-Coulomb formula to perform the yield stress computation (Bueler and Brown, 2009; Schoof, 2006). The basal melting of ice shelves is proportional to the squared thermal ocean temperature forcing ($\Delta T_{\text{force}}^2$), which is the difference between the pressure-dependent melting temperature of the ice and the actual ocean temperature above melting. Here, the parameterization considers the full depth-dependence of the ocean temperature field, as described in Sutter et al. (2019). Basal ice-shelf melting occurs only in fully floating grid points, while the grounding line position is determined on a sub-grid space (Feldmann et al., 2014) to interpolate basal friction.

The calving occurs at the ice-shelf margin, and three sub-schemes determine it. (1) At the ocean-ice-shelf margin, ice-shelf grid points with a thickness of less than $150\,\text{m}$ calve. (2) Ice shelves calve that extend across the continental shelf edge and progress into the depth ocean (defined by the $1500\,\text{m}$ depth contour). (3) The Eigen-calving parameterization exploits the divergence of the strain/velocity field (Levermann et al., 2012), with the proportionality constant of $1 \cdot 10^{18}\,\text{m\,s}$ or $1 \cdot 10^{17}\,\text{m\,s}$, respectively. Two independent spin-up runs delivering our initial conditions (PISM1Eq and PISM2Eq) utilize these constants. Ocean temperatures from the World Ocean Atlas 2009 (Locarnini et al., 2010) and the multi-year mean surface mass balance (SMB) from the RACMO 2.3/ANT model (Van Wessem et al., 2014) drive PISM during spin-up (Table 2). A similar model setup has taken part in the initMIP-Antarctica exercise under the name AWI_PISM1Eq with an adjusted Eigen-calving proportionality constant of $2 \cdot 10^{18}$ and no bed deformation (Seroussi et al., 2019a).

## 2.3 Surface Mass Balance

The surface mass balance (SMB) is computed via the PDD method, where the hydrological year starts on day 91. The PDD factor for snow and ice are $0.3296\,\text{cm(IE)\,Kelvin}^{-1}\,\text{day}^{-1}$ and $0.8792\,\text{cm(IE)\,Kelvin}^{-1}\,\text{day}^{-1}$, respectively. The temporal evolving annual 2m-air temperature standard deviation is derived from daily CMIP5 model values for each CMIP5 model at each ice sheet model grid-cell.

The reference data set (Table 2) drives three special ice sheet control runs "control 1", "control 2," and "control 3" (Table 3). These are performed to check whether a disturbance occurs when we replace the SMB used during the spin-up. The simulation named "control 1" is a continuation of the spin-up, where the SMB (Figure A16a) equals the precipitation from the reference

data set (Table 2). The utilization of the PDD approach provides the SMB in "control 2". In the simulation "control 3", the SMB is computed via PDD and considers a potential height difference between the reference data set and the evolving ice sheet surface (Figure A16b). For the height difference, we consider a lapse rate of $-7\,\mathrm{K\,km^{-1}}$. Since the height difference is zero at the beginning of this test, it initially does not influence the SMB. However, a lowering ice sheet surface in progressing simulations increases the air temperature used to compute the SMB via PDD. All the SMB distributions ("control 1" to "control

3") are numerically identical across Antarctica (Figure A16c) because Antarctica is too cold to experience melting via PDD (Figure 2a; A detailed analysis of the climate follows below in section 3.1: "CMIP5 Forcing Data Set"). Therefore, the altered computation does not trigger any disturbance, though the SMB computed via PDD does allow for melting.

## 2.4   Precipitation Scaling

Inspired by the Clausius-Clapeyron process, it is often assumed that with a warming atmosphere, the precipitation also in-

creases. Together with the contemporary climate fields as a reference, the air temperature scaling of precipitation is

$$S(t,\boldsymbol{x}) = \frac{1}{\Delta T(t,\boldsymbol{x})} \frac{P_{t=0}(\boldsymbol{x})}{\Delta P(t,\boldsymbol{x})} \cdot [100\%], \tag{1}$$

where $\Delta T$ is the air temperature anomaly, $\Delta P$ the precipitation anomaly, and $P_{t=0} = P(t_{\mathrm{ref}})$ the precipitation reference field. The scaled precipitation is

$$P(t,\boldsymbol{x}) = \Delta P(t,\boldsymbol{x}) + P_{t=0}(\boldsymbol{x}) = \Delta P(t,\boldsymbol{x})\,[1 + \Delta T(t,\boldsymbol{x}) \cdot S(t,\boldsymbol{x})]. \tag{2}$$

As reported above (section 1.2: "Scaling between Precipitation and Air Temperature Changes over Antarctica"), in the modeling context, it is often assumed that the scaling is constant: $S(t,\boldsymbol{x}) = S$.

## 3   Results

Depending on the applied CMIP5 forcing scenario, the ensemble mean climate signal is weaker for those scenarios following an aggressive mitigation path and, hence, releasing less carbon dioxide (e.g. RCP2.6). Around Antarctica, the here analyzed ensemble follows the same pattern (Figure 2 and Figure 3). Since in the past decade greenhouse gas concentrations have followed most closely the high-emission RCP8.5 scenario path, we will focus on RCP8.5 if not otherwise stated.

## 3.1   CMIP5 Forcing Data Set

From 1850 until the end of the 21st century, the CMIP5 data set spatial mean 2m-air temperature in Antarctica (see the map of Figure 3d) rises steadily by $6\,\mathrm{K}$ with a spread of $1\,\mathrm{K}$ (one standard deviation; Figure 3a) while the mean precipitation accumu-

lates $9 \pm 3 \, \mathrm{cm} \, \mathrm{year}^{-1}$ (water equivalent) in addition (Figure 3b). The average potential ocean temperature in the depth range of 150m to 500m depth along Antarctica's coast (see the map of Figure 3e) warms by nearly $1 \pm 0.18°\mathrm{C}$ in the same period (Figure 3c). In particular, since the beginning of the 21st century, these warming trends becomes stronger in the atmosphere and ocean.

### 3.1.1 Spatial Patterns in the Atmosphere

These changes are not homogeneous across the Antarctic continent (Figure 2d-l). The atmosphere warms strongest along the Antarctic Peninsula (Mulvaney et al., 2012; Thomas et al., 2009, in agreement with current observed trends), the high plateau of the East Antarctic Ice Sheet (EAIS) and to a lesser degree around the Filchner-Ronne-Ice Shelf region (Figure 2d, g, j). The warming is lowest in the coastal areas of East and West Antarctica that extend (clockwise) from the Greenwich Meridian via Wilkens Land and the Ross Ice Shelf to the Marie Byrd Land, respectively. The Amery Ice Shelf interrupts the coastal band of low 2-m air temperature rise. In general, warming trends are less pronounced over the adjacent ocean and ice-sheet interior.

The precipitation increases marginally across the high plateau of the EAIS and east of the Ross Ice Shelf as part of the WAIS (Figure 2e, h, k). In contrast, the coastal areas, where air masses with much precipitable water make landfall, receive more precipitation. Since these air masses on their way into the interior are uplifted by the steep topography, the precipitation along the coasts is topographically controlled. Areas of heavy precipitation under the reference climate (Figure 2b) also receive the highest increments. The precipitation increases strongest along the western Antarctic Peninsula, where the lifting of eastward flowing air masses by mountain ranges leads to topographic precipitation, which is firmly enhanced; this resembles the observed positive precipitation trend of the Antarctic Peninsula since 1900 (Wang et al., 2017).

### 3.1.2 Spatial Patterns in the Ocean

Under the control climate, the coldest potential ocean temperatures in the depth range from $150 \, \mathrm{m}$ to $500 \, \mathrm{m}$ exist offshore the coasts of Antarctica (Figure 2c). We detect the lowest ocean temperatures in front of the Filchner-Ronne, Amery, and Ross Ice Shelves. Also, the Amundsen Sea in front of Pine Island Glacier and Thwaites Glacier is cold.

The subsurface ocean-temperature warms vigorously along sections of the Antarctic Circumpolar Current (ACC) and in the western Weddell Sea at the center of the ocean gyre. For instance, the warm spot in the western Weddell Sea emerges in all CMIP5 models (Figure 2f, i, l). In the coastal strip surrounding Antarctica, the warming is of medium strength and heterogeneous. There, most robust warming appears in the Amundsen Sea and along the coast of the EAIS (between Wilkens Land and Terre Adélie) opposite of Australia. The least warming occurs in front of both the western Ross Ice and Filchner-Ronne Ice Shelves and the neighboring Antarctic Peninsula, where the ocean temperatures are lowest in the control climate (Figure 2c).

### 3.1.3 CMIP5 Data Set as Ice Sheet Model Forcing

The spatial structure of the anomalies discussed above is in general independent of the applied forcing scenario, while the scenarios determine, however, the strength of the anomalies. Regardless of the applied scenario, the discussion of the atmospheric climate anomalies indicates already that both precipitation and air temperature do not necessarily correlate. Instead, regional differences are evident, and a simple scaling of the precipitation with temperature appears to be inadequate.

In front of the Filchner-Ronne, Amery, and Ross Ice Shelves as well as in the Amundsen Sea, the climatological ocean temperature distribution might be too cold because it does not replicate the confined flow of warm water masses through glacier-scoured troughs towards ice shelves (see Figure in Appendix A). To overcome this limitation, we apply a spatially restricted melting correction. The correction increases the melting by $50\,\%$ for the Ronne Ice Shelf region, and it quadruples melting for coastal parts of the West Antarctic Ice Sheet between the Antarctic Peninsula and the Getz Ice Shelf (east of the Ross Ice Shelf).

### 3.2 Precipitation Scaling Across Antarctica

Ice-sheet simulations bridging several millennia often rely on climate anomalies deduced from ice cores, for instance. Based on isotopic signatures in ice cores, temperature anomalies are deduced. Inferred accumulation anomalies from these cores are converted into precipitation anomalies. The scaling deduced from ice cores varies in Antarctica between $5\,\%\,K^{-1}$ and $7\,\%\,K^{-1}$, with a 2-sigma uncertainty of about $1\,\%\,K^{-1} - 3\,\%\,K^{-1}$ (Figure 4, Table 4).

### 3.2.1 Spatial Pattern of Precipitation Scaling in the CMIP5 Data Set

The corresponding average CMIP5 scaling is generally larger than observational estimates at these ice core locations (Table 4). At the Vostok ice core location, the difference is most conspicuous, and the simulated is more than twice as large as the observed scaling. For EDML and EDC locations (Figure 4), there are also substantial differences of around a factor of two. In contrast, the scaling of the Law Dome, Talos Dome, and WAIS ice cores are indistinguishable from the corresponding CMIP5 average within the uncertainties. Here, we have computed the scaling by averaging the precipitation of the piControl run (first 50 years) to obtain the reference data (baseline) and the last 50 years of the RCP8.5 scenario from 2051 until 2100 to get the anomalies.

To test the result's robustness, we exchange the baseline: beginning of the historical period instead of the first 50 years of piControl. Both the first 50 years of piControl and the historical forcing (1850–1899) start from the same state but are subject to diverging forcing, e.g., in atmospheric greenhouse gases and volcanic events (such as Krakatau in 1883; Henderson and Henderson, 2009). Despite replacing the baseline, the values change only slightly. Since the results are very similar when exchanging the baseline, we restrict the analysis to anomalies relative to the first 50 years of the piControl climate and consider the results robust.

The spatial distribution of the scaling derived from our CMIP5 data set is heterogeneous and varies stronger than the ice core data suggest. Values in the range between $4\,\%\,K^{-1}$ and $6\,\%\,K^{-1}$ occur at the Filchner-Ronne Ice Shelf and in the coastal

Terre Adélie region (see Map in Figure 1 for place names). On the WAIS, these values are also present in the coastal strip from the Antarctic Peninsula to the Ross Ice Shelf and along the Transantarctic Mountain Range's eastern flank (Figure 4).

The highest scaling factor emerges on the EAIS, where a c-shaped area as part of the high plateau has factors exceeding $12\,\%\,K^{-1}$. This area reaches out to the Dronning Maud Land with very high scaling factors too. The West Antarctic Ice Sheet has scaling factors generally lower than $8\,\%\,K^{-1}$ and only on the elevated interior values up to $10\,\%\,K^{-1}$ are detected. Over the Ross Ice Shelf and the eastward adjacent Siple Coast, scaling factors are the lowest (Figure 4). Since we detect heightened scaling factors at some places with high elevation, we aimed at determining whether we could find a relationship between

elevation and scaling. However, neither for the entire Antarctic continent nor for defined subregions (see below), could we identify any robust relationship (not shown).

### 3.2.2  Precipitation Scaling Across Regions in Antarctica

Our analysis focuses now on the scaling factors of all grounded ice, which, if lost, contributes to a rising potential sea level. Additionally, we analyze the scaling factors for the entire continent (label "glaciered"), and four glaciated regions labeled

"EAIS Atl", "EAIS Ind", "WAIS", and "Siple Coast" (Figure 5 and Table 5). We detect a slight trend towards higher values if we restrict the analysis to ground ice ($87.5\,\%$ of the glaciated area, see Table 5). However, the scenario selection is decisive, while the choice between "glaciered" and "grounded" is unessential for the CMIP5 mean. In general, individual CMIP5 models show the same result. The sensitivity of many CMIP5 models to the range of applied scenario is within their variability (e.g., CSIRO-Mk3-6-0, CNRM-CM5, MIROC-ESM, MRI-CGCM3) or may hint at an enlarged scaling for weaker scenarios (e.g.,

MPI-ESM-LR). Frieler et al. (2015) found a low dependence of the scaling factors to four RCP scenarios for the whole Antarctic continent. Anomalies are not as distinctly pronounced in RCP2.6 as in the other scenarios due to the weaker forcing scenario. Please note that for CCSM4 RCP2.6 is missing (hence we have hatched the corresponding bar).

The boundaries of the three regions "EAIS Atl", "EAIS Ind", and "WAIS" resemble different oceanographic zones (Whitworth III et al., 2013; Orsi et al., 1999; Foldvik and Gammelsrød, 1988) under the consideration of Antarctica's large-scale

drainage basins (Zwally et al., 2015). This chosen division of Antarctica does not produce surface area of equal size. As already indicated by the spatial distribution (Figure 4), the ordering from high to low scaling factors would be "EAIS Atl", "EAIS Ind", and "WAIS". The difference between both "EAIS" regions is minor, with a tendency towards higher values in "EAIS Atl" in the CMIP5 mean and some individual CMIP5 models. Some models do not show a clear trend between the scenario strength and scaling factor. For example, for MRI-CGCM3 the scaling decreases in "EAIS Atl" from RCP4.5 over RCP8.5 to RCP2.6, while

in "EAIS Ind" the order is different from RCP8.5, RCP2.6, to RCP4.5 (Figure 5). It indicates again, that regional differences matter.

The region "WAIS" has significantly lower scaling factors than both "EAIS" regions. This difference exists for the CMIP5 model average regardless of the applied scenarios and for almost all individual CMIP5 models (Figure A3). Exceptions are MIROC-ESM and MPI-ESM-LR under the RCP2.6 scenario and HadGEM2-ES under all scenarios.

## 3.3 Sea Level Impact of Precipitation Scaling by Air Temperature

To understand how the precipitation boundary condition impacts Antarctica's contribution to the global sea level, we inspect the precipitation fallen on Antarctica (Figure 6). Therefore, the precipitation is integrated over time since 1850 and across the dark-blue masked region representing grounded ice (map on Figure 6). This analysis is restricted to all CMIP5 models driven by RCP8.5 and anomalies computed relative to the first 50 years of the control run. Since accumulated precipitation integrated over Antarctica lowers the global sea level under the assumption that ice loss (basal melting or calving) does not occur, the temporally accumulated potential sea-level impact curves have a negative slope (Figure 6a, b). Further on, this quantity is labeled "integrated precipitation."

In this manuscript, we distinguish between potential/diagnosed sea level and simulated sea level. The potential sea level is the transformation of an ice mass or freshwater volume into a global sea level by applying a global ocean area of $3.61 \cdot 10^{14} \mathrm{m}^2$ (Gill, 1982). In contrast, the simulated sea level is a diagnostic of the ice sheet model, which takes into account the released total mass above flotation and the global ocean area.

The integrated precipitation declines more forcefully since the beginning of the 21st century, which is driven by the concurrent increase of precipitation over Antarctica (Figure 3b). The integrated precipitation shows a more pronounced temporal change than the mean precipitation (Figure 3b) because the vast interior, characterized by light precipitation, dominates the integral. After the year 2100, the integrated precipitation declines linearly (Figure 6b), as we adopt the forcing of the years 2071–2100 recurrently. By applying the actual precipitation anomalies (solid lines, Figure 6a, b), the potential sea-level drop is stronger than using a scaling of $5\,\%\,\mathrm{K}^{-1}$ (dashed lines, Figure 6b) because the models' internal scaling exceeds $5\,\%\,\mathrm{K}^{-1}$ (Figure 5). In the year 5000, the sea level drop ranges from $5\,\mathrm{m}$ to $11\,\mathrm{m}$ when applying simulated precipitation anomalies and only from $3\,\mathrm{m}$ to $6\,\mathrm{m}$ when using the $5\,\%\,\mathrm{K}^{-1}$ scaling.

The difference of the integrated precipitation between $5\,\%\,\mathrm{K}^{-1}$ scaled and directly-applied precipitation anomalies is always positive (solid lines in Figure 6 c, d). This difference ranges approximately from $1\,\mathrm{cm}$ (CISRO-Mk3-6-0) to $15\,\mathrm{cm}$ (CCSM4) in the year 2100 and from $60\,\mathrm{cm}$ (MPI-ESM-LR) to $550\,\mathrm{cm}$ (CCSM4) in the year 5000.

A lower scaling of $2\,\%\,\mathrm{K}^{-1}$ causes a magnified difference (dotted lines in Figure 6 c, d). Ultimately, it corresponds to a reduced potential sea-level contribution. It leads to differences ranging from $5\,\mathrm{cm}$ (MPI-ESM-LR) to $21\,\mathrm{cm}$ (CNRM-CM5) in 2100 and from $150\,\mathrm{cm}$ (MPI-ESM-LR) to $850\,\mathrm{cm}$ (CCSM4) in 5000.

A higher scaling of $8\,\%\,\mathrm{K}^{-1}$ (dashed line in Figure 6c, d) exceeds ice core-based estimates (Table 4, Figure 4), while it corresponds approximately to the CMIP5 data set average (RCP8.5 $\approx 8.2\,\%\,\mathrm{K}^{-1}$, RCP4.5 $\approx 7.8\,\%\,\mathrm{K}^{-1}$, Figure 5). Now, only the CCSM4 model exhibits a positive difference because its scaling reaches $11\,\%\,\mathrm{K}^{-1}$ (Figure 5). Four models are nearly balanced (CNRM-CM5, MRI-CGCM3, HadGEM2-ES, NorESM1-M), while the remaining four feature negative differences (CISRO-Mk3-6-0, CanESM2, MIROC-ESM, MPI-ESM-LR). Hence, the difference range is subject to a change of sign, and the individual differences range from $-5\,\mathrm{cm}$ (CISRO-Mk3-6-0) to $7\,\mathrm{cm}$ (CCSM4) in 2100 and from $-170\,\mathrm{cm}$ (CISRO-Mk3-6-0) to $280\,\mathrm{cm}$ (CCSM4) in 5000.

## 3.4 Relation between Precipitation Boundary Condition and Ice Sheet Thickness

For the diagnostic of the relation between precipitation and ice sheet thickness, we inspect the ensemble mean (average across all ice-sheet simulations) and also the maximum and minimum thickness at each grid point across all ensemble members. Therefore, the field of joined extreme values could come from a diverse set of ice-sheet ensemble members and, hence, does not necessarily lead to dynamically consistent distribution.

Some ice sheet simulations ("control 3", Table 3) are driven solely by the reference forcing fields (Figure 2a-c), e.g., they neglect any anomaly. In these simulations, the detected trend of about $2\,\mathrm{mm}\,\mathrm{decade}^{-1}$ (sea-level equivalent) fades within the first 400 years and differs slightly between the two initial states (PISM1Eq and PISM2Eq). Even if we apply anomalies on top of the reference background fields, we can not exclude a shock-like behavior of the simulations entirely directly following the decades after the year 1850. Since we compute the anomalies relative to the average over the first or the last 50 years, respectively, of the control run for each climate model, these anomalies are not necessarily zero at the beginning of the year 1850. Hence, the ice-sheet model may experience a small jump, which causes an artificial trend initially.

In the year 2100, the ice thickness for both precipitation boundary conditions (precipitation anomaly deduced from the applied climate models versus scaled precipitation) increase over large parts of the Antarctic continent (Figure 7b-e). The thickness for the simulations driven by scaled precipitation grows less over substantial parts of the interior than in the simulations forced by the precipitation anomalies (Figure 7a), as the difference between scaled precipitation and applied precipitation anomaly is mostly negative. This pattern explains the diagnostic, where the sea-level drop is weak for temperature-scaled precipitation with a scaling of $5\,\%\,\mathrm{K}^{-1}$ (Figure 6).

A ring of a pronounced negative thickness difference exists along the coast. This ring emerges for a significant part of the coastal East Antarctic Ice Sheet (EAIS) and West Antarctic Ice Sheet (WAIS). For the latter ice sheet, the negative area is shifted away from the coast towards the interior (Figure 7a). A negative strip of the thickness difference appears at the south side of the Transantarctic Mountain Range, and some grounded ice streams flowing into the Filchner-Ronne Ice Shelf.

Regions of positive differences coincide with thicker ice for simulations driven by scaled precipitation. These are located south of the Transantarctic Mountain Range at the northern edge of the Ross Ice Shelf, along the coastline of the WAIS, and in the coastal Terre Adélie region. There, the scaling is generally lower or falls behind the constant scaling of $5\,\%\,\mathrm{K}^{-1}$. However, this does not explain exclusively positive areas.

For both precipitation boundary conditions, the mean ice thickness of each of the respective sub-ensemble reveals a widespread weakening of the floating ice shelves, such as Filchner-Ronne, Ross, and Amery Ice Shelves (Figure 7b, d). In the WAIS, both Pine Island Glacier and Ferrigno Ice Stream (an ice stream that flows into the Filchner Ice Shelf) thin drastically. Along the Antarctic Peninsula, general shrinking occurs along the coasts. Ice thins also along the coasts of EAIS.

For some places, the ice thickness thins for both precipitation boundary conditions across all ensemble members as the reduction of the maximal ice thickness highlights (Figure 9c, e). This reduction marks those outlet glaciers and ice shelves that are extremely vulnerable. These are around the Rutford Ice Stream, Foundation Ice Stream, Ronne Ice Shelf, Amery Ice Shelf,

three outlet glaciers (in "EAIS Ind" as part of Wilkens Land, Terre Adélie, and George V Land), northwestern Ross Ice Shelf (Ross Island), and Pine Island together with Thwaites Glacier in the Amundsen Sea (Figure 9c, e).

## 3.5 Precipitation Boundary Condition and Sea Level

In the following, we consider the entire ensemble (Table 3). Ensemble members start from both initial states PISM1Eq and
PISM2Eq, they are driven by all climate scenarios (historical followed by RCP2.6, RPC4.5, or RCP8.5; Table 2), and the anomalies are computed relative to the first or last 50 years of the related control run (piControl). The simulated sea-level curves are shifted so that the simulated sea-level contribution is $0\,\mathrm{m}$ in the year 2000 (Figure 8). Since the spread of individual ensemble members may not follow a normal distribution, we present beside the mean also the median sea-level contribution. For the RCP8.5 scenario, we highlight the spreading among models by depicting the standard deviation ($1\sigma$).

For the period 1850 until 2000, the simulated sea-level contribution of Antarctica fluctuates slightly. Hence, the accumulation balances nearly the ice loss at the margin while the basal melting rates of grounded ice is steady (Figure A9). Please note that there is no drift involved, as we have subtracted the trend from the continued ice-sheet simulations under the reference climate (Table 2). We also detect an amplified signal for the simulations driven by the precipitation anomalies compared to those forced by temperature-scaled precipitation anomalies, which corresponds to the above diagnosed sea-level impact of the precipitation
(Figure 6).

    After the year 2000, all our ensemble members, regardless of the forcing scenario, gain mass causing a falling simulated sea level (Figure 8). The basal melting of grounded ice does not affect the sea-level evolution, because this basal melting rate is nearly constant and negligible. Hence, the corresponding integrated sea-level equivalent grows linearly for all scenarios from 1850 until 2100, and only after the year 2500 these curves diverge (Figure A9). Also, the combined loss of iceberg calving
and basal melting of floating ice shelves does not vary considerably over the considered period. Consequently, the growth of simulated accumulation explains the net mass gains and, hence, the negative sea-level contributions from Antarctica after the year 2000 (Figure 8). Depending on the applied forcing and precipitation boundary condition, the global simulated sea-level drop ranges from $2\,\mathrm{cm}$ to $11\,\mathrm{cm}$ until 2100 (Figure 8). This result is in contrast to various publications, and we discuss it below.

    If we continue our ensemble with the last 30 years of forcing until the year 5000, the simulated sea-level contribution of
those ensemble members driven by the temperature-scaled precipitation starts to stabilize and reaches a minimum around the year 2500 (Figure 8). Afterward, they begin to lose more ice at the margins than they gain in the interior. As a consequence, these simulations produce on average a positive contribution to the global simulated sea level after the year 3200 (RCP8.5) and 3900 (RCP2.6), which compensates for the negative contributions since 1850. In the year 5000 at the end of our simulations, these simulations show a trend towards a continuously growing ice loss rate, because the curves have still an upward-directed
tendency. Hence a quasi-equilibrium is not established. In contrast, the simulations driven by the precipitation anomalies continue to show a falling simulated sea level. They always contribute negatively to the global simulated sea level until the year 5000. Their ensemble mean and median sea levels tend to converge towards a new equilibrium at the end of the simulations (Figure 8).

## 4 Discussion

In all CMIP5 models, the 2m-air temperature warms across the entire Antarctic continent without any exception (Figure 2d, g, j, and 3a), because even the minimum 2m-air temperature anomaly is positive everywhere (Appendix Figure A2d, g, j). The warming enhances the hydrological cycle, which causes generally heavier precipitation (Figure 3b) in particular along the coast of Antarctica (Figure 2e, h, k). However, the changing precipitation does not increase at the same rate with increasing air temperature because it is not only thermodynamically influenced but also dynamically controlled. Given that the ensemble mean

temperature scaling is different for the West and East Antarctic Ice Sheet (Figure 5) and has a considerable spatial dependence, the dynamical component is not negligible. Instead, the region of reduced precipitation under rising air temperatures, which we have identified along the Siple Coast, highlights that the dynamics could compensate or even overwhelm the impact of thermodynamics. The continent-wide scaling is per se problematic, even if we would adjust the scaling factor to reproduce the continental-wide average scaling. In this case, the integrated precipitation would be identical, but the spatial structure is still

entirely different (Figure 4). Hence for a realistic projection of Antarctica's sea-level contribution, it is imperative to consider the dynamical effect and the resulting spatial pattern of the future accumulation of precipitation.

The detected downward trend in snow accumulation in the Siple Coast area occurs also in the observations over the last decades (Wang et al., 2017) while the wider West Antarctic Ice Sheet region belongs to the most rapidly warming regions globally (Bromwich et al., 2012). It underpins that less accumulation can befall under a warming climate. Around Antarctica,

CMIP5 models generally simulate a shrinking sea ice extent that modifies the evaporation from the ocean (Turner et al., 2013; Bracegirdle et al., 2008). This sea ice reduction impacts the atmospheric circulation, which controls the flow of humid air masses, delivering precipitation to the Siple Coast. In contrast, observations feature a slightly increasing trend in total Antarctic sea ice extent resulting from larger opposing trends in different sectors (Eayrs et al., 2019; Parkinson, 2019). For example, sea ice has expanded in the Ross Sea (Haumann et al., 2016; Liu, 2004). In general, CMIP5 models neither represent this overall

nor these regional trends correctly (Eayrs et al., 2019; Parkinson, 2019; Bracegirdle et al., 2008). It is an open question if improvements of the simulated sea ice extent significantly reduce precipitation biases.

Although some models simulate decreasing precipitation around the Siple Coast, they have deficits: Even if NorESM1-M reproduces the overall seasonal sea ice extent cycle better than most CMIP5 models (Turner et al., 2013), it shows an unrealistically declining February sea ice trend in the Ross Sea over 1979–2005 (Turner et al., 2013). MPI-ESM-LR has large

negative errors in sea ice extent over the year (Turner et al., 2013).

The ocean (Etourneau et al., 2019) and atmosphere (Mulvaney et al., 2012; Thomas et al., 2009; Morris and Vaughan, 2003) is already warming along the Antarctic Peninsula. This results in a southward progressing of the annual mean 2-m air temperature isotherms of $-9\,°C$ or $-5\,°C$, which is regarded as the range of thresholds for the stability of ice shelves (Morris and Vaughan (2003, -9°C) and Doake (2001, -5°C)). It may also enable the formation of meltwater ponds on ice shelves

(Kingslake et al., 2017) that precedes (van den Broeke, 2005) or even triggers ice-shelf disintegration (Banwell et al., 2013, 2019). After an ice shelf has decayed, the feeding ice streams are losing more ice, as seen for Larsen-B (Rott et al., 2011), which lowers the thickness of grounded ice. Nevertheless, ice shelves along the Antarctic Peninsula have collapsed or are

retreating (Cook and Vaughan, 2010; Rott et al., 1996). In our simulations under the RCP8.5 scenario, this observed retreat and the related ice loss will continue.

For part of the EAIS, simulations show that grounded ice of the Wilkens Basin in the hinterland of George V Land may be prone to a massive ice loss if the ice front loses its buttressing effect (Mengel and Levermann, 2014). Our ensemble shows, on average, a stable situation here. However, ice in deep troughs that are in contact with the warming ocean thins at some spots further to the west. It happens in front of the Astrolabe Trench (in Terre Adélie) and on the coast of the Wilkens Land, for example near the Totten Glacier. Ice also thins in the deep trench leading to the Amery Ice Shelf.

Both the Pine Island and Thwaites Glaciers in the Amundsen Sea as part of the marginal West Antarctic Ice Sheet lose ice (Jeong et al., 2016; Milillo et al., 2019; Rignot et al., 2014; Scambos et al., 2017). According to the ensemble projecting the future, for them, continuous ice loss is inevitable. It also shows that the Ferringo Ice Stream flowing into the Bellingshausen Sea will thin in the future.

Our results do not support the commonly used method to compute precipitation changes via a temporal evolving air temperature in concert with a universal constant. This scaling has a clear spatial structure (Figure 4 and 5). In all large regions ("glaciered", "grounded", "EAIS Atl", "EAIS Ind", and "WAIS"; Figure 5), we see a trend towards lower scaling factors for weaker forcing scenarios in the CMIP5 data set mean, except for "EAIS Ind", where the factors for RCP8.5 and RPC4.5 are indistinguishable. Frieler et al. (2015) found only a low dependence of the scaling factors to the RCP scenario in comparison with the dependence on the specific climate model. Here, the region "WAIS" has on average a lower precipitation scaling than both regions of the East Antarctic Ice Sheet ("EAIS Atl" and "EAIS Ind"), which is also reflected by the scaling factor maxima in these regions (Figure 4). As before, the Ross Ice Shelf and the adjacent Siple Coast feature have, on average, the lowest scaling factors across the entire ice sheet (Figures 4 and 5). Some individual CMIP5 models project even negative scaling: precipitation deficit for rising air temperatures (Figures 4 and 5).

The Siple Coast highlights definitely that it is not adequate to describe the spatial evolution of the precipitation by a fixed air temperature scaling at a continental scale. Since the scaling exceeds mostly the commonly utilized value of $5\,\%\,\mathrm{K}^{-1}$, for instance, we diagnose the potential sea-level impact of applying the actual scaling distribution (e.g., Figure 4) versus a spatially and temporally constant scaling of $2\,\%\,\mathrm{K}^{-1}$, $5\,\%\,\mathrm{K}^{-1}$, or $8\,\%\,\mathrm{K}^{-1}$ across Antarctica. It also highlights that simulations driven by temperature-scaled precipitation could be misleading because they do not reproduce declining precipitation under raising air temperatures.

## 4.1 Attribution of the Driving Model

Above, we discuss how in the entire ensemble the ice sheet thickness changes on average (Figure 7). In contrast, the maximum and minimum thickness at a given grid location is determined by climate forcing from one particular climate model. We inspect which climate model may lead to ice thickness growth or shrinking and restrict ourselves first to the model year 2100, when the transient forcing of period 1850–2100 excites changing ice sheet thicknesses.

### 4.1.1 Ice Sheet Simulations Driven by Precipitation Anomalies

Directly at margins apart from the vast ice shelves, the attributed model that drives either the maximum or minimum ice thickness shows a noisy small-scale pattern (Figure 9d, e). Hence, the marginal regions cannot be associated with a particular climate model. In contrast, the mean and minimum thicknesses of the Filchner-Ronne and Ross Ice Shelves, and also to some extent the Amery Ice Shelf, are highlighted by a nearly unique color patch indicating a reduced thickness. These patches are separated from the surroundings showing either a reduced thinning or even thickening. Intriguingly, the MIROC-ESM model forcing, for instance, thickens grounded ice east and west of the Ross Ice Shelf (Figure 9d), while it also predominantly thins the Ross Ice Shelf (Figure 9e). Hence, the ocean forcing drives the ice-shelf thinning. Since the spatial pattern of extreme atmospheric and ocean forcing that promotes or undermines the ice thickness is not necessarily aligned, this may explain the small scale noisy pattern along the coast. Also (nonlinear) dynamical changes on the considered time scales may occur in response to both ocean and atmospheric forcing.

Beyond the direct coast strip, larger areas appear where the forcing from one climate model determines the maximum or minimum thickness, respectively. However, these extended continuous regions are often interrupted by spots controlled by the climate from other models. The pattern is also changing during the transient simulation because the temporal evolution of the 2m-air temperature and precipitation anomalies are different for each climate model, as the integrated precipitation highlights (Figure 6a, b). Furthermore, after the year 2100, where the same 30 years forcing period (2071–2100) drives the ice-sheet model recurrently, the pattern evolves further (Figure 10). This pattern alteration occurs because the ice sheet has not reached the quasi-equilibrium to the last 30 years forcing.

For grounded ice, three models (CCSM4, CNRM-CM5, MIROC-ESM) determine predominantly the growing ice until the year 2100 (Figure 9d), which is in-line with the diagnosed sea-level contribution (solid line, Figure 6a, b). CCSM4 dominates the "EAIS Atl" sector, while CNRM-CM5 dominates a band from the "EAIS Ind" sector clockwise to the Antarctica Peninsula, which is interrupted by regional-scale patches of the MIROC-ESM. A spatial dominance is not apparent for the minimum ice thickness, because the patchwork of five models (CSIRO-Mk3-6-0, HadGEM2-ES, MPI-ESM-LR, MRI-CGCM3, NorESM1-M) dominates the year 2100. NorESM1-M influences the WAIS, which is supported by its lowest scaling in the Siple Coast region (Figure 5). CSIRO-Mk3-6-0 has an impact around the South Pole, MRI-CGCM3 affects the coastal zone in the EAIS. The control of MPI-ESM-LR and, to a lesser extent, HadGEM2-ES spreads across the entire continent. If we progress into the year 2200, where we have applied the 30 years forcing more than three times, the emerging picture is nearly unchanged for the maximum thicknesses. In contrast, the models' diversity causing minimum thicknesses shrinks and is apparently dominated by three models CSIRO-Mk3-6-0, NorESM1-M, and MPI-ESM-LR (Figure 10).

### 4.1.2 Ice Sheet Simulations Driven by Air Temperature-Scaled Precipitation

We now turn towards those model simulations, in which the air temperature-scaled precipitation forcing has been applied. In those the mean, maximum, and minimum ice thickness distribution (Figure 11) are similar to the ones driven by the precipitation anomalies as discussed above (Figure 7). Also, the same models determine the ice-shelf thickness of the Filchner-Ronne

and Ross Ice Shelves. The latter shows that primarily the ocean controls ice-shelf thickness changes in our simulations. How-
ever, we detect a stark contrast of the model determining the maximum and minimum ice thickness. For the maximum, we
still have the same three models (CCSM4, CNRM-CM5, MIROC-ESM). However, the pattern has changed. CCSM4 controls
a smaller area in the interior around the South Pole, and MIROC-ESM some coastal regions of the East Antarctic Continent.
The remaining majority of the grounded ice is under the control of CNRM-CM5. The most striking changes occur for the
minimum. Now, NorESM1-M determines the entire WAIS and also some parts of "EAIS Ind". MRI-CGCM3 dominates the
remaining East Antarctic Ice Sheet.

In the latter case, air temperature variations force the precipitation-driven ice sheet thickness evolution exclusively (see
Equation 1). Dynamical changes influencing the precipitation are not considered. Hence, the applied scaling or precipitation
boundary condition impacts the temporal evolution of the Antarctic Ice Sheet geometry, which ultimately shapes Antarctica's
contribution to the global sea level.

## 4.2  Ice Sheet Losses

After the spin-up, the simulations have reached a quasi-equilibrium. For the discussion of the ice losses, we concentrate on the
transient period 1850–2100. For all climate scenarios, the calving rate hardly changes (Figure A12), whereas the total ice-shelf
area is nearly constant until 2000 and declines afterward (Figure A15). The ocean-driven basal melting is proportional to the
squared ocean temperature difference between the pressure-dependent melting temperature and the actual ocean temperature.
Since the ocean temperature increases in general (Figure 2f, i, l and Figure 3c), also the mass loss by basal melting increases,
while the total shelf ice area remains quasi-constant until 2000 and declines afterwards (Figure A15). For RCP8.5, the basal
melting increases at the end of the 21st century quadratically. To conclude, the calving rate is nearly constant, while the basal
melting increases by approximately $33\%$ between the years 2000 and 2100.

The mean calving rate is about $8000\,\mathrm{Gt\,year^{-1}}$ and $5000\,\mathrm{Gt\,year^{-1}}$ for the ensemble member utilizing the parameters and
the initial state of PISM1Eq and PISM2Eq, respectively (Figure A12). The basal melting rates for PISM1Eq and PISM2Eq
are similar, however, the loss rates for PISM1Eq are slightly larger than PISM2Eq (Figure A13). The ensemble mean starts at
about $550\,\mathrm{Gt\,year^{-1}}$ in 1850 and reaches $900\,\mathrm{Gt\,year^{-1}}$ in 2100.

Since floating ice shelves nourish ice losses by basal ice shelf melting and iceberg calving, these ice losses do not directly
impact the sea-level. Under the assumption that the inflow of former grounded ice compensates any shelf mass loss, the reported
ice losses of $8500\,\mathrm{Gt\,year^{-1}}$–$9000\,\mathrm{Gt\,year^{-1}}$ ($5500$–$6000\,\mathrm{Gt\,year^{-1}}$) would correspond to a sea-level rise of $2.58\,\mathrm{cm\,year^{1}}$–
$2.74\,\mathrm{cm\,year^{1}}$ ($1.67\,\mathrm{cm\,year^{1}}$–$1.83\,\mathrm{cm\,year^{1}}$). The integration over 250 years to match the period from 1850 to 2100 would
generate a potential sea-level equivalent of $6.47\,\mathrm{m}-6.85\,\mathrm{m}$ ($4.19\,\mathrm{m}-4.57\,\mathrm{m}$). However, the actual ratio between total ice
mass change and the corresponding potential sea level response is obviously not a 1:1 relation. Instead, on average less than
$5\%$ of the total mass lost diminishes grounded ice that raises the sea level (Figure A8). Considering this ratio of $5\%$, the sea
level impact reduces to $0.32\,\mathrm{m}-0.34\,\mathrm{m}$ ($0,21\,\mathrm{m}-0.23\,\mathrm{m}$) by 2100. It is less than integrated precipitation anomalies across
the Antarctic continent (Figure 6a), which explains the total mass gains.

Nevertheless, the integrated basal melting rates are too low (Figure A13) and the calving rates are too high (Figure A12) compared to observational estimates in our ensemble of ice-sheet model simulations. Besides the fact that the total loss exceeds recent observational estimates, our ice sheet is in a quasi-equilibrium after the spin-up. All this may indicate that the integrated precipitation driven accumulation resulting from the RACMO precipitation reference field might be too large. However, the surface mass balance of RACMO agrees well with observational estimates (Wang et al., 2016), while the uncertainty of the surface mass balance (sea-level equivalent of $\sim 0.25 \, \mathrm{mm \, year^{-1}}$ (Van Wessem et al., 2014)) is of almost the same size as Antarctica's observational-based sea-level contribution ($\sim 0.2 \, \mathrm{mm \, year^{-1}}$ between 1992 and 2011 (Shepherd et al., 2012; Wang et al., 2016)). Additionally, recent satellite-based estimates indicate clearly that the Antarctic Ice Sheet has lost mass (sea-level equivalent: $0.4 \, \mathrm{mm \, year^{-1}}$) in the period 2011–2017 (Sasgen et al., 2019).

Beyond the year 2100 (Figure A14), the calving rates decrease and reach a minimum in the period 3000–4000. Afterward, calving increases again slightly. Basal melting rates are subject to a slight decreasing trend (RCP2.6), nearly constant values (RCP4.5), or a negligible upward trend after the year 4000 (RCP8.5).

### 4.2.1 Sea Level Contribution of Corrected Basal Melting

Since the simulated sea level contribution of Antarctica disagrees with the currently observed state showing mass loss, we apply a corrected time series emulating the observational-based ocean-driven basal melting. This analysis shall reveal if a more vibrant basal melting rate in concert with the simulated ice sheet mass evolution leads to a less pronounced ice sheet growth or drives even ice loss. Ultimately, does a more vigorous melting of ice shelves raise the simulated sea level of all ensemble members?

By construction, the corrected time series preserve the fluxes' amplification over time, which is essentially the ratio of the higher end value to the lower start value. Hence, the corrected basal melt flux replicates the original simulated amplification while the flux is identical to the observed reference value ($F_{\mathrm{ref}}(t_{\mathrm{ref}})$) at the reference time ($t_{\mathrm{ref}}$). Under the assumption that only a fraction of the adjusted basal mass contributes to the global sea level, we apply the simulated ratio of the sea level change to the total ice mass change. This ratio renders the so-called dynamic ice loss, and for each ensemble member, it is the median ratio over its entire time series (for details see Section D "Figures" on page 53 in the appendix). Since we examine enhanced mass loss, we do not adjust the iceberg calving rates that are already higher than observed.

By adjusting the basal melting flux, the determined temporal evolution of the sea level correction (Figure A5, Equation D8) does impact the global simulated sea level. Still, it does not change the sign of the contemporary sea-level evolution. Consequently, the impact on the simulated sea level is very small (Figure A6). If we assume instead that all of the additional mass loss of floating ice shelves rises the simulated sea level immediately, we would obtain too extensive corrections of $30 \, \mathrm{cm}$ between 1850 and 2000. This corresponding sea-level rise would be larger than the observed integrated sea level rise of about $20 \, \mathrm{cm}$ since 1850 (Church and White, 2011), which has been driven by world-wide land-water storage changes, shrinking glaciers around the globe, enhanced melting from Greenland, and thermal expansion of the ocean (Cazenave and Remy, 2011; Leclercq et al., 2011; Church and White, 2011).

The sea level correction exceeds observations considerably under the unrealistic assumption that additional basal melting of already floating ice shelves would immediately raise the sea level. In contrast, a negligible sea level correction occurs if we apply the inferred ratio of about $5\,\%$ between simulated total ice mass loss and the corresponding sea-level rise. To conclude: A more vigorous but realistic melting of ice shelves does not essentially raise the simulated sea level.

### 4.3 Limitations

Our simulations presented here are in contrast to others that project a sea-level rise from a shrinking Antarctic Ice Sheet. Some previous studies simulate Antarctica with a finer spatial resolution (Golledge et al., 2015; Pollard et al., 2015), which could improve the representation of ice streams. These streams channelize the flow of grounded ice from the interior to the margins, where they feed the attached ice shelves and discharge directly into the ocean. Despite our coarser resolution than the ones used in recent studies, our simulated surface velocity distribution reasonably reproduces satellite-based estimates (Appendix Figure A10 and Figure A11). Others used the cliff failure parameterization supporting ice loss together with a constant ocean temperature offset of $+2\,°C$ (Pollard et al., 2015), twice as large as the amount found in our data set of nine CMIP5 models (Figure 3), or utilized continuously raising atmospheric and oceanographic temperature forcing (Golledge et al., 2015; Mengel et al., 2015; Winkelmann et al., 2012, 2015) beyond the year 2100. These stronger forcings alone explain a large part of the difference because we apply recurrently the forcing of the years 2071–2100 after 2100.

As already discussed, the application of anomalies may trigger a small shock at the beginning of each simulation. This shock creates an artificial trend in the simulated sea-level time series initially. Nevertheless, the long-term positive and negative sea-level contribution of Antarctica for simulations driven by temperature-scaled and directly-applied precipitation anomalies, respectively, are robust.

An issue could be the parameterization of the grounding line migration because a high resolution of about $1-2\,km$ is needed, according to Gladstone et al. (2012). However, PISM's grounding line parameterizations at medium to lower resolution is consistent with higher-order models (Feldmann et al., 2014). It explains that the present-day grounding line position resembles the current state reasonably, and the simulated grounding line retreat follows the bulk of simulations in the last model intercomparison (Seroussi et al., 2019a); hence, we consider our grounding line migration as reasonable. The apparent stability of ice shelves in the runs driven by the precipitation anomalies seems to comply with the safety band of ice shelves (Fürst et al., 2016), so the calving does stay outside of ice-shelf regions essential for providing buttressing for the inflowing grounded ice streams.

The ocean boundary condition, where ocean conditions are extrapolated into the ice-shelf cavities, drive basal ablation of ice shelves. Here, we could undoubtedly improve simulations if the ice shelves would be coupled to the driving ocean model, so that basal melting impacts the thermal structure of the ocean and, ultimately, the melt patterns. CMIP5 models neglect the ocean-ice-shelf interaction (Meijers, 2014), and their coarse resolution around Antarctica does not allow the representation of the regional conditions (Heuzé et al., 2013; Sallée et al., 2013b). They are subject to unrealistic open-ocean convection (Heuzé et al., 2013; Meijers, 2014; Sallée et al., 2013a) instead of convection on or near the continental shelf (Årthun et al., 2013; Nicholls et al., 2009). All these taint the hydrographic structure along Antarctica's coasts. Hence, any improved parameterization can not

rectify the existing biases in the ocean forcing. These biases are reduced if we apply ocean temperature anomalies on top of an observational-based climatological data set as performed in our study.

Since we extrapolate coastal ocean temperatures laterally into the ice-shelf cavities, the obtained ocean warming might be higher if it would include the amplified warming of the gyre centers. If this may have been incorporated in the forcing of other groups obtaining a higher ice loss, depends on the setup details. However, it may help to bridge the gap between other studies and our simulations.

Nevertheless, the simulated sea-level decrease for the used precipitation anomaly forcing is in agreement with a growing surface mass balance since 1800 AD, driven mainly by the Antarctic Peninsula region (Thomas et al., 2017). During intensive El Nino years, the accumulation-driven ice height increase between Dotson Ice Shelf and Ross Ice Shelf exceeds the height reduction by basal melting processes (Paolo et al., 2018), but the ice mass is still decreasing since the low-density snowfall replaces ice with a higher density. The stability arguments of Ritz et al. (2015) confirm the apparent stability of Antarctica in our simulations. Furthermore, various recent ice-sheet model simulations, driven by selected CMIP5 climate model fields in the framework of the ISMIP6 exercise, are subject to a negative sea-level contribution under a warming climate (Seroussi et al., 2019b).

## 5 Conclusions

How precipitation is specified in ice-sheet simulations is crucial to the outcome of numerical simulations of Antarctica's sea-level contribution. The commonly used method of scaling the precipitation changes with the simulated air temperature changes from ice cores or global climate models leads to a positive Antarctic simulated sea-level contribution, i.e., a simulated sea-level rise. However, when considering the simulated precipitation changes from the global climate models, the situation changes. In this case, our numerical projections simulate a negative sea-level contribution. Nevertheless, independent of the applied precipitation boundary condition, we detect regions where the ice sheet thickness thins for all ensemble members. These regions are the Amundsen Sea Embayment including Pine Island and Thwaites Glaciers, some outlet glaciers of the East Antarctic Ice Sheet (EAIS) between George V and Wilkens Land, the Amery Ice Shelf, and along the Northern Antarctic Peninsula (Figure 7c and e). These regions correspond to those, which have been identified across sixteen models within a recent model intercomparison exercise, where marginal ice wanes due to ocean warming (Seroussi et al., 2019a).

Precipitation and air temperature, on average over the entire Antarctic continent, grow simultaneously in climate model simulations of the future (Figure 3). In concert with estimates of accumulation changes and air temperature anomalies obtained from ice cores, it may (mis)lead us to scale the precipitation by the temporally evolving air temperature. Therefore, fixed scaling factors are common. However, a tendency towards higher scaling exists under more vigorous climate trends (Figure 5), and the scaling has a clear spatial dependence (Figure 4 and 5). As a consequence, the accumulated snowfall on Antarctica for future climate projections differs between the methods, which ultimately leads to biased estimates of Antarctica's contribution to the global potential sea level (Figure 6). To assess the introduced bias, we analyze simulations of the Parallel Ice Sheet

Model driven with numerous variants of the above-discussed climate conditions and a diverse set of implemented boundary conditions.

The region "Siple Coast" (area $0.69 \cdot 10^6$ km$^2$, see Table 5) as a part of the "WAIS" region (area $4.26 \cdot 10^6$ km$^2$) is different in many aspects. It has the smallest area compared to the other regions (Table 5), and it shows the lowest mean scaling factors for all scenarios. Also, as before, no clear trend exists between different scenarios across the entire CMIP5 data set, while the

685 spread of trends among individual CMIP5 models is substantial (Figure 5). Furthermore, some members exhibit a negative scaling, where precipitation decreases for rising air temperatures: MPI-ESM-LR under the RCP8.5 scenario and NorESM1-M under all scenarios (RCP8.5, RCP4.5, and RCP2.6). Observations over the last decades feature a downward trend in snow accumulation in this region (Wang et al., 2017). The inverted sign of the scaling is in stark contrast to the CMIP6 data set average.

Major uncertainties affect these simulations, such as the partitioning of ice losses into calving and basal melt — which is quite different from observational estimates due to very crude representations in the ice-sheet model — or the omission of important processes, such as the interaction between ocean, ice shelves, and ice sheet. While we could improve some aspects of the involved process descriptions, our simulations are state-of-the-art and suffer, thence, the same limitations as others.

Since the precipitation boundary condition determines if Antarctica raises or lowers the global sea level (Figure 8), it may

be appropriate to utilize a more sophisticated surface mass balance (SMB) model. The recent publication that indicates a Greenlandification of Antarctica's margin at the end of the century (Bell et al., 2018) supports this approach, but the required atmospheric inputs fields are not available at sufficient temporal resolution. Hence, this will be an option for simulations driven by the forthcoming CMIP6 model output.

To evaluate the impact of the precipitation boundary condition, fully coupled simulations between a dynamic ice-sheet/shelf

model and a global climate model are inevitable. The system would include the ice-shelf-ocean interaction of coupled ocean-ice shelves at a sufficiently high spatial resolution around Antarctica. In addition, it would contain a sophisticated surface mass balance computation. We hope these coupled atmosphere-ocean/sea-ice–ice-sheet/shelf models will overcome the discussed limitations. The discrepancy of the simulation results between both methods describing the precipitation illustrates the uncertainty of the possible range of future precipitation growth in a warming world.

*Code and data availability.* The Code of the Parallel Ice Sheet Model is freely available from https://github.com/pism/pism. Modifications of the PISM's code are available from TK upon reasonable request. The data is available from the corresponding author or TS upon reasonable request.

*Author contributions.* MP and CR prepared the CMIP5 data. CR performed the simulations and wrote the manuscript. All authors contributed to the interpretation of the results and proofreading of the manuscript.

*Competing interests.* The authors declare that they have no conflict of interest.

*Acknowledgements.* We thank all reviewers for their engagement and their excellent suggestions leading to an improved manuscript. This work has been financed through the German Federal Ministry of Education and Research (Bundesministerium für Bildung und Forschung: BMBF) project ZUWEISS (grant agreement 01LS1612A). Parts of this work are supported by BMBF grant 01LP1503B (project PalMod1.2). CR acknowledges funding via the Alfred Wegener Institute's research program PACES2. The Deutsche Klima Rechenzentrum (DKRZ) supplied computer resources on the cluster "mistral". CR wants to thank the AWI's HPC administrators for their proactive and generous support enabling this work during the development phase. The development of PISM is supported by NASA grant NNX17AG65G and NSF grants PLR-1603799 and PLR-1644277. The data analyzes and the production of figures have been predominantly performed with the help of the following software products (alphabetic order): Climate Data Operators (CDO: https://code.mpimet.mpg.de/projects/cdo), Generic Mapping Tools (GMT: https://www.generic-mapping-tools.org), Ncview (http://meteora.ucsd.edu/~pierce/ncview_home_page.html), netCDF Operator (NCO, http://nco.sourceforge.net), PyFerret (https://ferret.pmel.noaa.gov/Ferret/documentation/pyferret), python (python3, https://www.python.org, including the following packages NumpPy: https://numpy.org, matplotlib: https://matplotlib.org, and xarray: https://xarray.pydata.org). We thank the numerous authors and their financial supporters of these software products.

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

## Tables

**Table 1.** List of CMIP5 models and the used RCP climate projections covering the period 2005–2100 (Moss et al., 2010) beside the historical ("hist", period 1850–2004) and the piControl ("piCtrl") scenarios. The third column lists the length of the piControl simulation ("piCtrl"). Note, we do not use the RCP2.6 scenario of the CCSM4 model. See also Table 3.

| Model Name | RCP Projections | Scenarios | Length of piCtrl |
|---|---|---|---|
| CanESM2 | RCP2.6, RCP4.5, RCP8.5 | hist, piCtrl | 996 years |
| CCSM4 | RCP4.5, RCP8.5 | hist, piCtrl | 1051 years |
| CNRM-CM5 | RCP2.6, RCP4.5, RCP8.5 | hist, piCtrl | 850 years |
| CSIRO-Mk3.6.0 | RCP2.6, RCP4.5, RCP8.5 | hist, piCtrl | 500 years |
| HadGEM2-ES | RCP2.6, RCP4.5, RCP8.5 | hist, piCtrl | 575 years |
| MIROC-ESM | RCP2.6, RCP4.5, RCP8.5 | hist, piCtrl | 630 years |
| MPI-ESM-LR | RCP2.6, RCP4.5, RCP8.5 | hist, piCtrl | 1020 years |
| MRI-CGCM3 | RCP2.6, RCP4.5, RCP8.5 | hist, piCtrl | 500 years |
| NorESM1-M | RCP2.6, RCP4.5, RCP8.5 | hist, piCtrl | 501 years |

**Table 2.** Forcing used for ice-sheet model spin-up and as reference fields for the anomaly forcing.

| Forcing | Period | Label | Reference Fields | Reference |
|---------|--------|-------|------------------|-----------|
| Atmosphere | 1979–2011 | RACMO 2.3/ANT | 2m-air temperature, Total precipitation | Van Wessem et al. (2014) |
| Ocean | Climatological mean | World Ocean Atlas 2009 (WOA09) | Potential ocean temperature | Locarnini et al. (2010) |

**Table 3.** Members of the Ice Sheet Model Ensemble. The second column "D/A" indicates if forcing has been applied directly "D" or as an anomaly "A". The column "SMB" represents the surface mass balance, where "PDD" is the positive degree day approach, and "LR" indicates the use of a air temperature correction due a local ice surface height difference utilizing a constant lapse rate. If the entry is "precipitation" than the precipitation of the forcing data set equals the surface mass balance. The simulations "control 1" to "control 3" are driven by the reference data sets (Table 2). Other ice sheet simulations are forced by anomalies on top of the reference data sets. These anomalies are computed relative to the the first and last 50 years of the available piControl simulations. We utilize the following CMIP5 scenarios: "piCtrl"="piControl", "hist"="historical", "RCP2.6", "RCP4.5", and "RCP8.5" (Table 1). The CMIP5 scenario "hist" represents the historical period form 1850 until 2004 and the three projections "RCP2.6", "RCP4.5", and "RCP8.5" cover the period 2005–2100. Beyond the year 2100, the forcing of the last 30 years (2071–2100) are recurrently applied until the model year 5000. The data set comprises 26 anomaly forcing scenarios. Each scenario starts from the initial condition PISM1Eq (Figure A10) and PISM2Eq (Figure A11) and is driven by two precipitation conditions (see main text for details, e.g. sections 2.3: "Surface Mass Balance" and 2.4: "Precipitation Scaling"). Hence, the ensemble of anomaly ice sheet simulations has 208 members plus "control" runs.

| Name | D/A | SMB | Years | Remark |
|------|-----|-----|-------|--------|
| Spin-up | D | precipitation | – | (for reference: not used here) |
| control 1 | D | precipitation | 1850–5000 | SMB see Figure A16a |
| control 2 | D | PDD | 1850–5000 | |
| control 3 | D | PDD + LR | 1850–5000 | SMB see Figure A16b |
| hist+RCP2.6: first 50 yr | A | PDD + LR | 1850–5000 | recurring 2071–2100 beyond 2100, |
| hist+RCP4.5: first 50 yr | A | PDD + LR | 1850–5000 | anomaly computed relative to the |
| hist+RCP8.5: first 50 yr | A | PDD + LR | 1850–5000 | *piControl* mean (first 50 years) |
| hist+RCP2.6: last 50 yr | A | PDD + LR | 1850–5000 | recurring 2071–2100 beyond 2100, |
| hist+RCP4.5: last 50 yr | A | PDD + LR | 1850–5000 | anomaly computed relative to the |
| hist+RCP8.5: last 50 yr | A | PDD + LR | 1850–5000 | *piControl* mean (last 50 years) |

**Table 4.** Air temperature scaling of the precipitation for six ice core locations in Antarctica. The second column lists the ensemble mean scaling (RCP8.5, first 50 years, both initial states PISM1Eq and PISM2Eq) and standard deviation (2-sigma) across all ensemble members. The third column provides scaling factors deduced from ice cores (Frieler et al., 2015), including the provided error margins (2-sigma). Please inspect Figure 4 for the ice core locations.

| Core Name Location | Scaling of Ensemble Mean | Scaling Ice Core |
|---|---|---|
| EDML | 11.0±6.6 | 5.0±2.8 |
| Vostok | 14.0±5.6 | 6.1±2.5 |
| Law Dome | 5.8±6.3 | 5.2±2.3 |
| EDC | 11.0±5.0 | 5.9±2.2 |
| Talos Dome | 8.4±5.2 | 6.8±2.8 |
| WAIS | 6.8±5.4 | 5.5±1.2 |

**Table 5.** Defined areas as part of our diagnostic. The fraction is computed relative to "glaciered." The Figures 1, 4, and 5 depict these areas.

| Region Label | Area ($10^6$ km$^2$) | Fraction (%) | Longitude Range | Comment |
|---|---|---|---|---|
| glaciered | 13.6 | 100.0 | $[-180°\text{E}, +180°\text{E}[$ | Antarctica incl. ice shelves |
| grounded | 11.9 | 87.5 | $[-180°\text{E}, +180°\text{E}[$ | Without ice shelves |
| EAIS Atl | 3.77 | 27.6 | $[-45°\text{E}, +55°\text{E}]$ | |
| EAIS Ind | 5.66 | 41.1 | $[+55°\text{E}, +155°\text{E}]$ | Including floating ice shelves |
| WAIS | 4.26 | 31.3 | $[+155°\text{E}, -45°\text{E}]$ | |
| Siple Coast | 0.69 | 5.12 | $[+155°\text{E}, -140°\text{E}]$ | Latitude$> 85°$S |

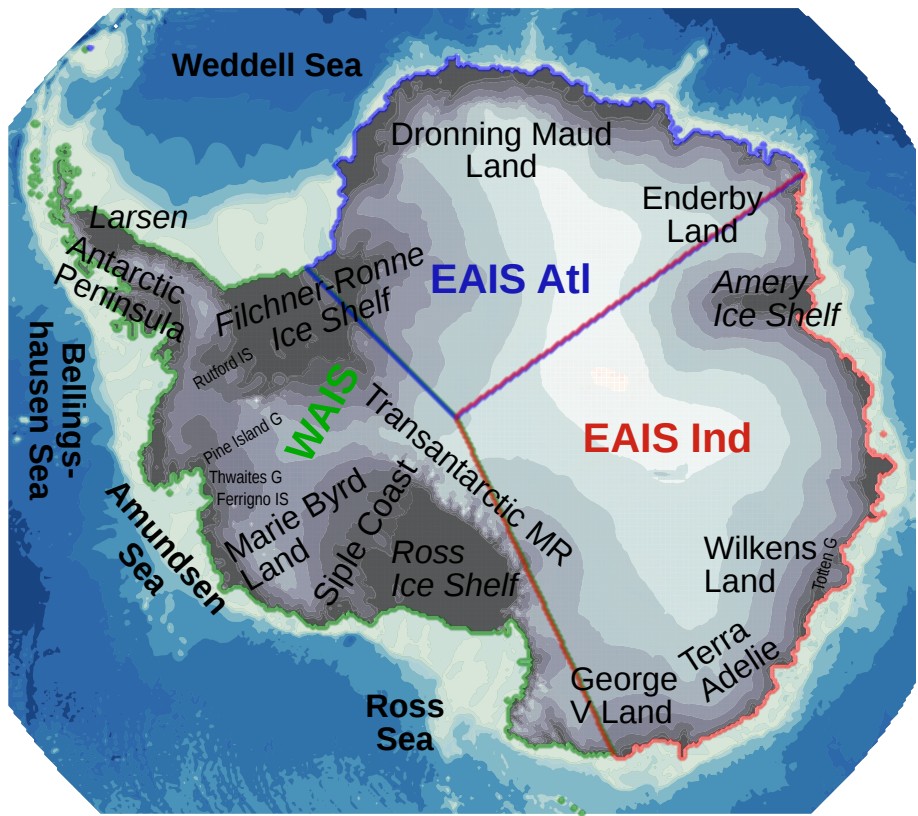

**Figure 1.** Map of Antarctica. The seafloor depth is shown with a blue color scale, while the elevation of Antarctica above sea level is depicted by a colorbar of dark-gray (low elevation) to white colors (high elevation). The font style of ocean labels is in bold and of ice shelves is in italic. The smaller font size tags individual glacier (G) and ice streams (IS). The abbreviation "MR" stands for "Mountain Range". Colored labels define three regions: WAIS: West Antarctic Ice Sheet (green), EAIS Atl: East Antarctic Ice - Sheet Atlantic Sector (blue), EAIS Ind: East Antarctic Ice Sheet - Indian Ocean Sector (red). These regions bound by the coastal areas by their shared boundaries in the interior. Also the Figures 4 and 5 show the boundaries of these regions. The here depicted bedrock topography and surface orography are taken from Fretwell et al. (2013).

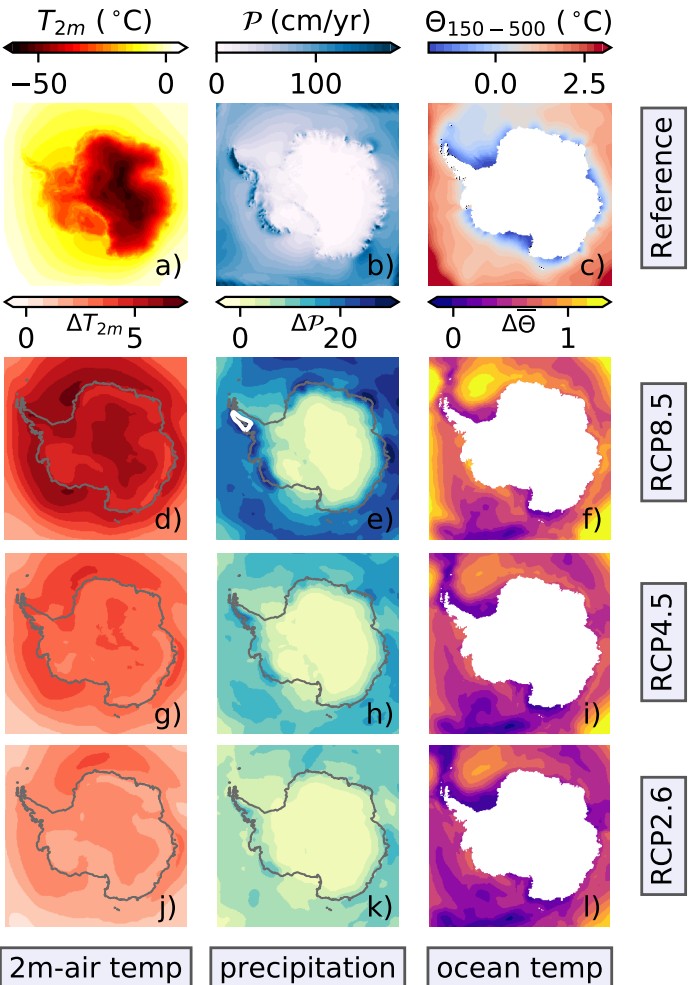

**Figure 2.** CMIP5 data set mean anomalies (d–l) relative to the atmospheric (a, b) and oceanographic (c) reference forcing. The corresponding maximum and minimum fields are depicted in the appendix Figure A1 and Figure A2, respectively. The top row represents the reference fields to spin-up the ice-sheet model (Table 2). The 2m-air temperature (a) and the total precipitation (b) are mean fields from the regional RACMO model, while the ocean temperatures come from the World Ocean Atlas 2009 (c); see Table 2 for more details. Each reference field has its colorbar above its plot. Below each reference field, the related anomalies, including their colorbar, are compiled for the period 2071–2100. Here, the second (third and fourth) row shows the anomalies for RCP8.5 (RCP4.5, RCP2.6). In these atmospheric anomaly plots, the dark-gray line follows the current coastline. All potential ocean temperatures (c, f, i, l) are a vertical mean of the depth interval from 150 m to 500 m. The white contour lines in the anomaly plots highlight the following precipitation threshold (e, h, k): 30 cm/yr. All these anomalies are the CMIP5 model mean of the models listed in Table 1; please note that CCSM4 is not part of RCP2.6. Antarctica's contours are deduced from Fretwell et al. (2013).

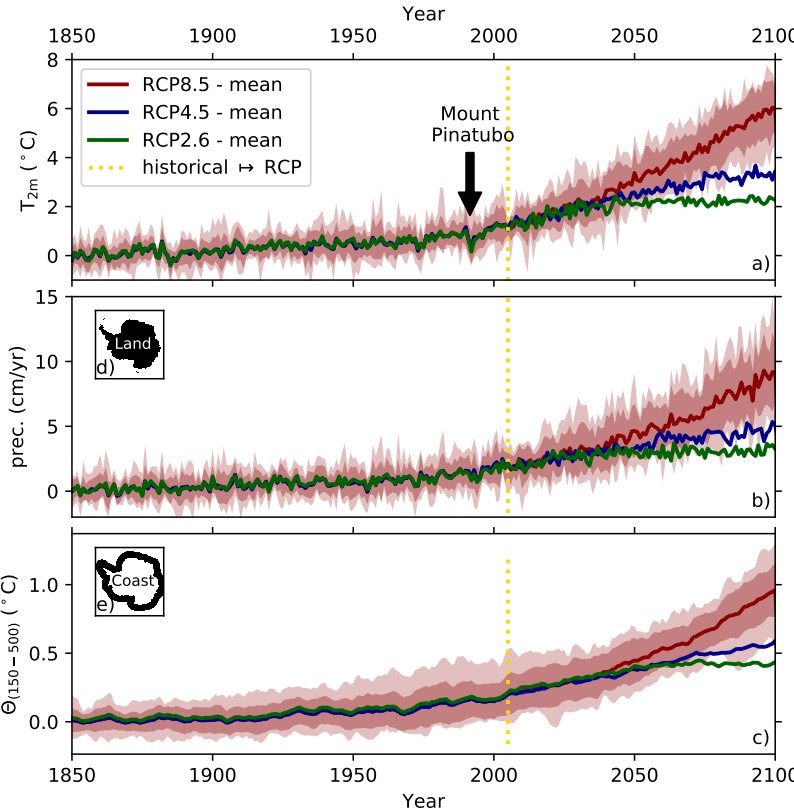

**Figure 3.** Spatial mean of the a) 2m-air temperature and b) total precipitation anomalies on Antarctica (d). Spatial c) potential ocean temperature mean averaged over the depth interval from $150\,\mathrm{m}$ to $500\,\mathrm{m}$ in the coastal zone (e) surrounding Antarctica. The CMIP5 data set mean values are shown for the scenarios according to the legend in a). The dark red band highlights the 1-sigma standard deviation (66 %), while the light red band shows the full range covered by all CMIP5 models for RCP8.5 only. The vertical golden line marks the transition from the historical forcing to the RCP. The distinct air temperature jump during the historical period in 1991 marks the Mount Pinatubo volcano eruption. The contours of the Antarctic continent (d) follow the outer edges defined by the data set of Fretwell et al. (2013), while the coastal strip (e) is an extension into the sea with smoothed northern edges (typical width of about $500\,\mathrm{km}$).

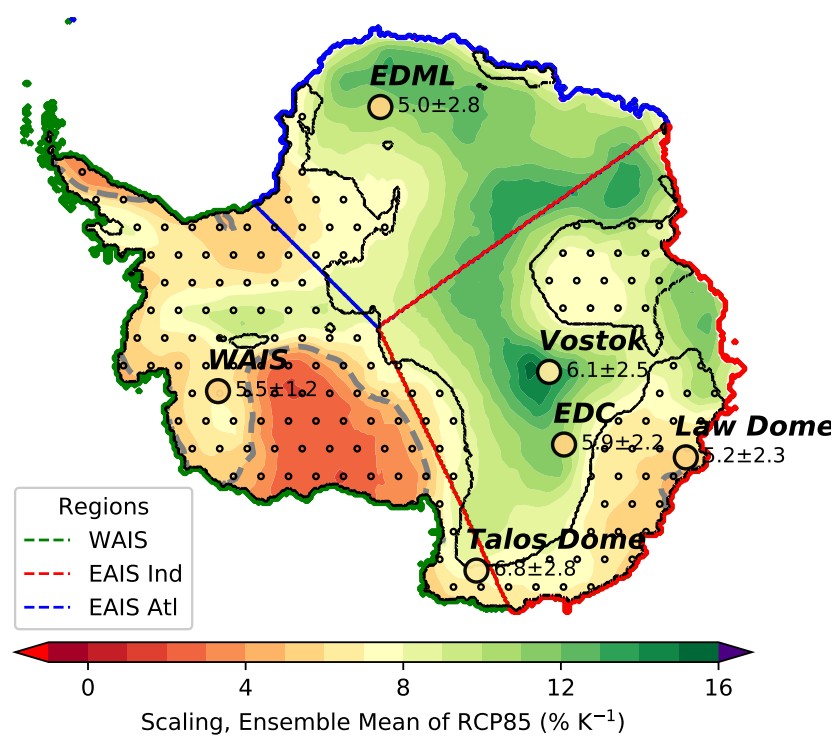

**Figure 4.** CMIP5 data set mean of the temperature-scaled precipitation for the period 2051–2100. This scaling under the RCP8.5 scenario comes from nine CMIP5 models (Table 1), which are driven by anomalies relative to the first 50 years of piControl. In the dotted regions enclosed by black contours, the combined simulated scaling and the standard deviation contains the value of $5\,\%\,\mathrm{K}^{-1}$. Gray dashed lines follow this $5\,\%\,\mathrm{K}^{-1}$ contour. The scaling values deduced from ice cores are shown at their location (Frieler et al., 2015) by using the same colorbar as the spatial distribution within the circle. The neighboring printed values are the mean and the 2-sigma uncertainty. Three defined regions (Table 5) named "WAIS", "EAIS Atl", and "EAIS Ind" are outlined by their green, blue, and red, respectively, boundaries (lower left legend). For further details, inspect the section 3.2 "Precipitation Scaling Across Antarctica", please. Appendix Figure A3 provides corresponding distributions for each CMIP5 model. Antarctica's contour is deduced from Fretwell et al. (2013).

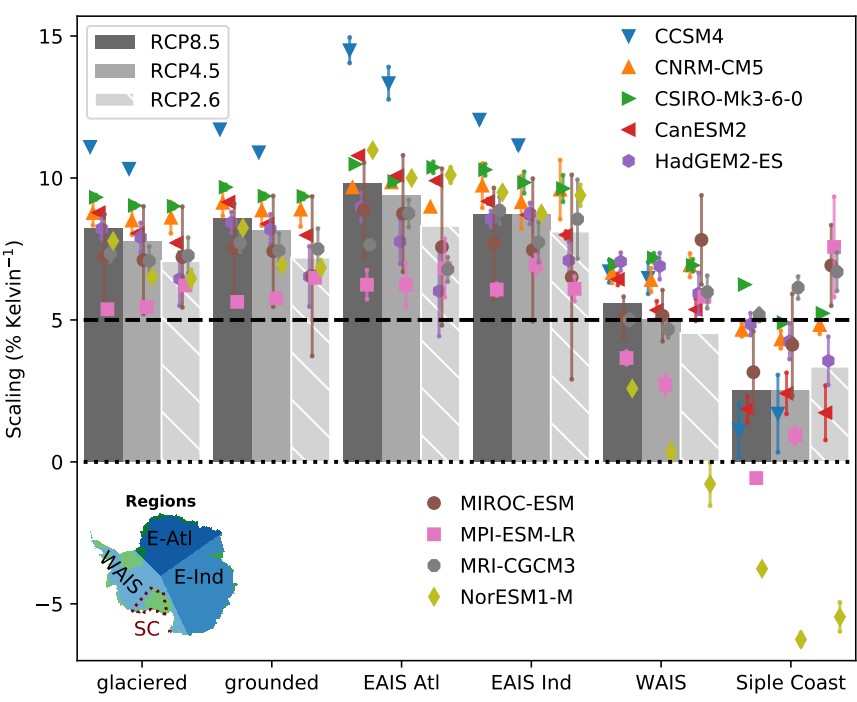

**Figure 5.** Air temperature-precipitation scaling deduced for the nine CMIP5 models (Table 1) and three future scenarios (legend) in six defined regions in Antarctica (see map in the lower-left corner and Table 5. The coastlines and the grounding line positions are deduced from Fretwell et al. (2013)). The gray bars represent the CMIP5 data set average, whereas the individual symbols stand for CMIP5 models. Here the results apply for both reference periods, where the anomalies are computed relative to the first or last 50 years of piControl. Each symbol is the model average of both reference periods, while the attached line indicates the scatter range between the first and last 50 years reference period. Please note, that the RCP2.6 scenario does not include the CCSM4 model; hence, the corresponding bar is hatched.

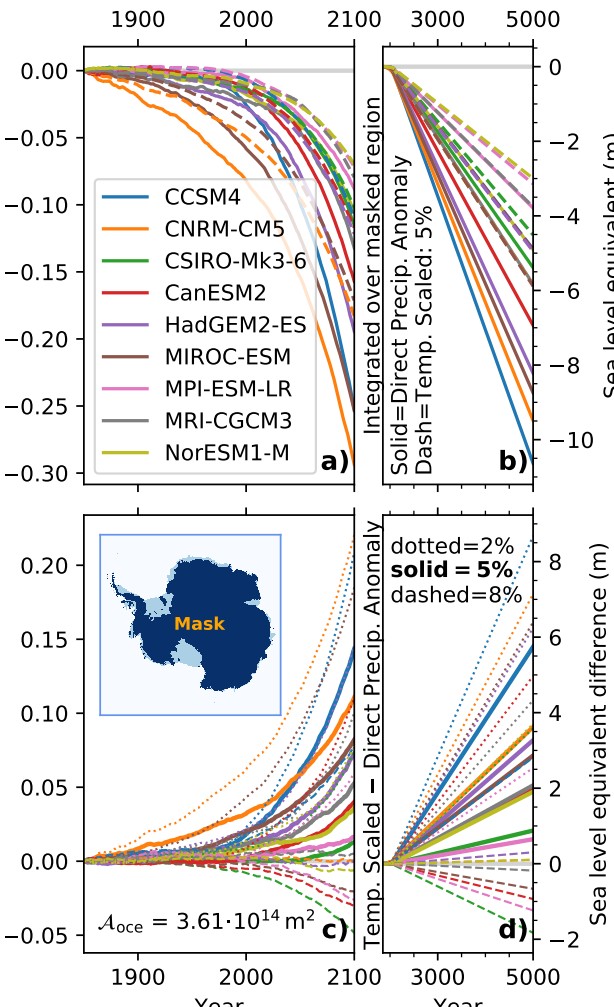

**Figure 6.** The top row (a, b) shows the integrated potential sea-level equivalent of the precipitation falling on grounded ice in Antarctica (see dark-blue mask in the lower left, the light-blue parts highlight ice shelves; The grounded and floating ice areas are derived from Fretwell et al. (2013)) for the anomaly forcing (solid lines) and temperature-scaled precipitation (dashed lines) considering a scaling of $5\,\%\,K^{-1}$. The difference in the potential sea-level impact between the anomalies and the temperature-scaled precipitation is depicted in the lower row (c, d). Here, the solid lines consider scaling of $5\,\%\,K^{-1}$, while the dotted and dashed lines consider a scaling of $2\,\%\,K^{-1}$ and $8\,\%\,K^{-1}$, respectively. The left subfigures a) and c) are restricted to the period 1850–2100, while b) and d) cover the full period from 1850 until 5000. Every single colored line (see legend in the upper left) represents one CMIP5 model (Table 1). The corresponding curves for the scenario RCP4.5 as well as for a different mask that covers the entire continent are available in the Appendix Figure A4.

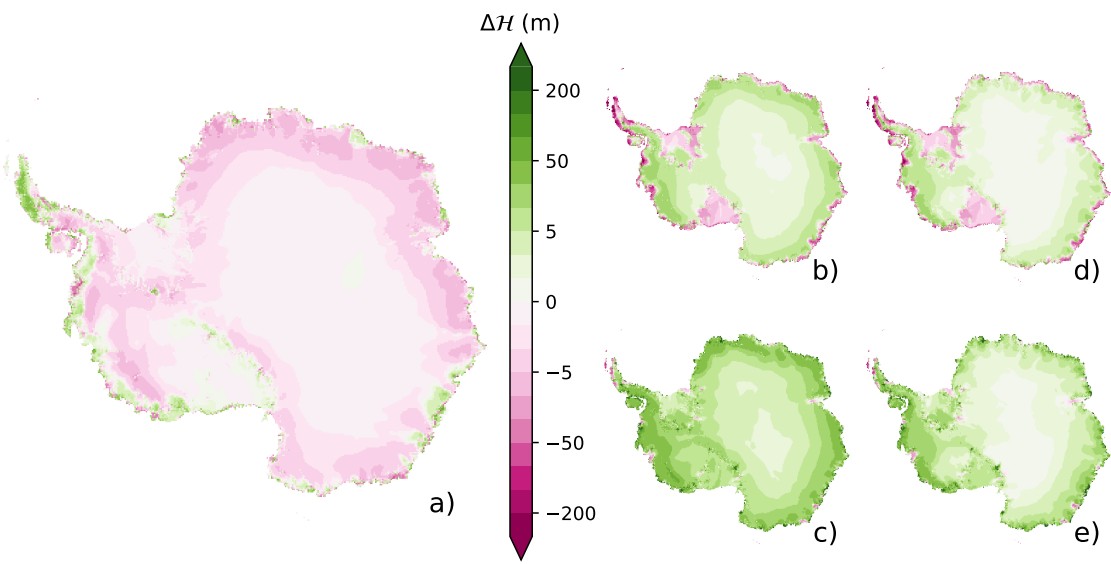

**Figure 7.** Ice thickness changes under the RCP8.5 scenario in the year 2100 since the year 1850. The ensemble mean difference between the runs forced by the scaled precipitation and the precipitation anomalies (a). For each climate model scenario, the anomalies are computed relative to the 50 years of the related piControl scenario. The simulations driven with the precipitation anomaly (b, c) have the mean ice thickness (b), and the maximum ice thickness (c) changes. The temperature-scaled precipitation of $5\,\%\,\mathrm{K}^{-1}$ gives the corresponding ensemble mean (d) and maximum (e). Please note that all subplots share the same colorbar, and subplot (a) equals subplot (d) minus subplot (b).

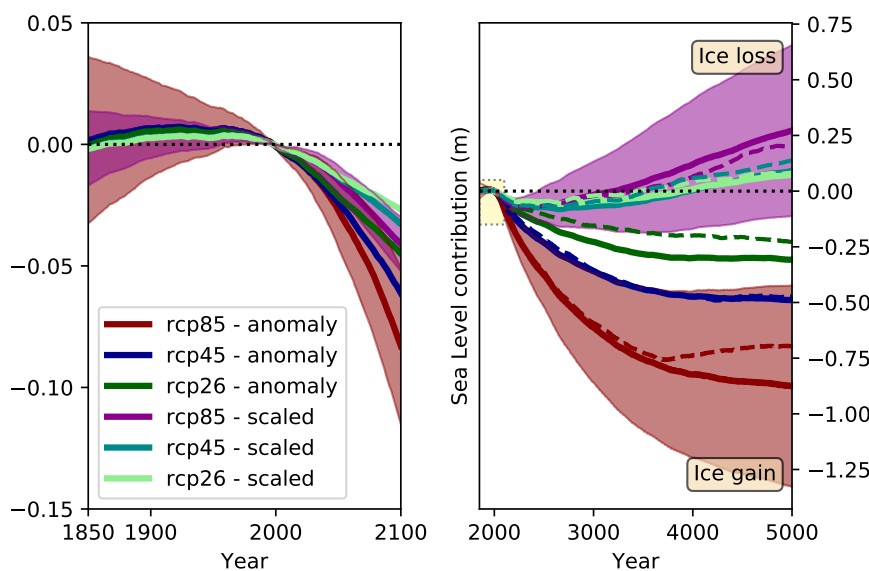

**Figure 8.** Sea level contribution of Antarctica computed by the ensemble of ice-sheet simulations (please see section 3.5 "Precipitation Boundary Condition and Sea Level" for details). The solid lines represent the ensemble averages for the applied precipitation anomalies and the air temperature-scaled precipitation boundary conditions according to the legend (lower left), while the dashed lines are the corresponding medians. For the RCP8.5 scenario, the shading highlights the standard deviation (1-sigma) as a measure of the variability among the ice-sheet ensemble members driven by various climate models (Table 1).

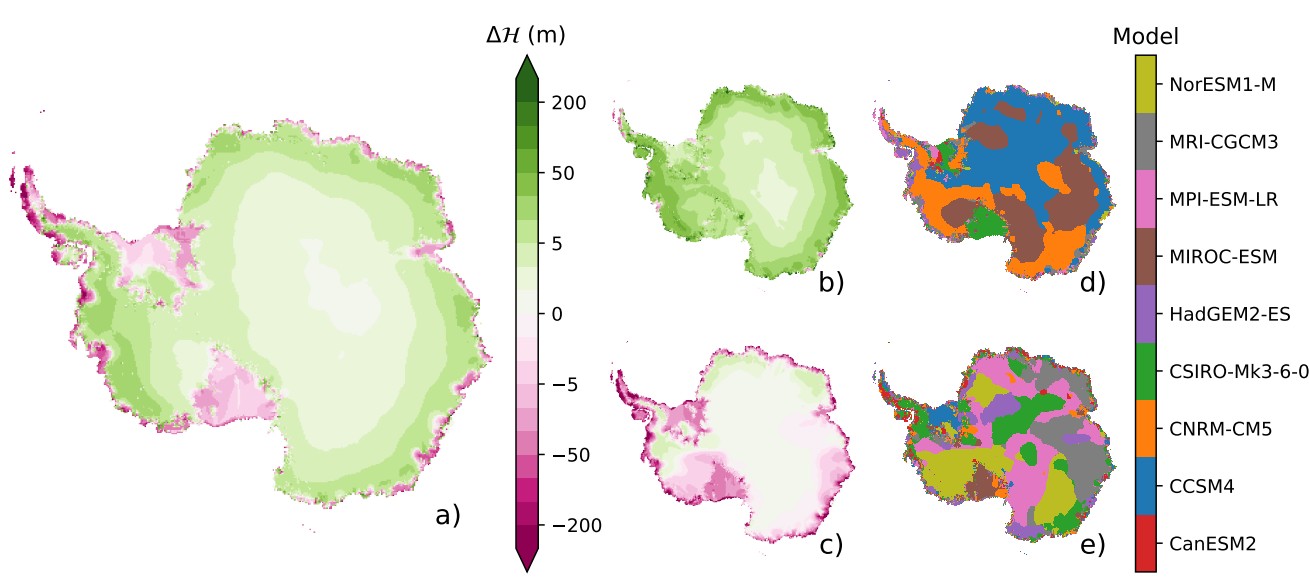

**Figure 9.** Ice thickness changes since 1850 under the RCP8.5 scenario for the actually applied precipitation anomaly in the year 2100. Highlighted are the (a) ensemble mean, maximum (b), and minimum (c). The climate model that is used to drive the ice-sheet model simulation causing the maximum and minimum thickness are shown in (d) and (e), respectively, next to the ensemble maximum (b) and minimum (c).

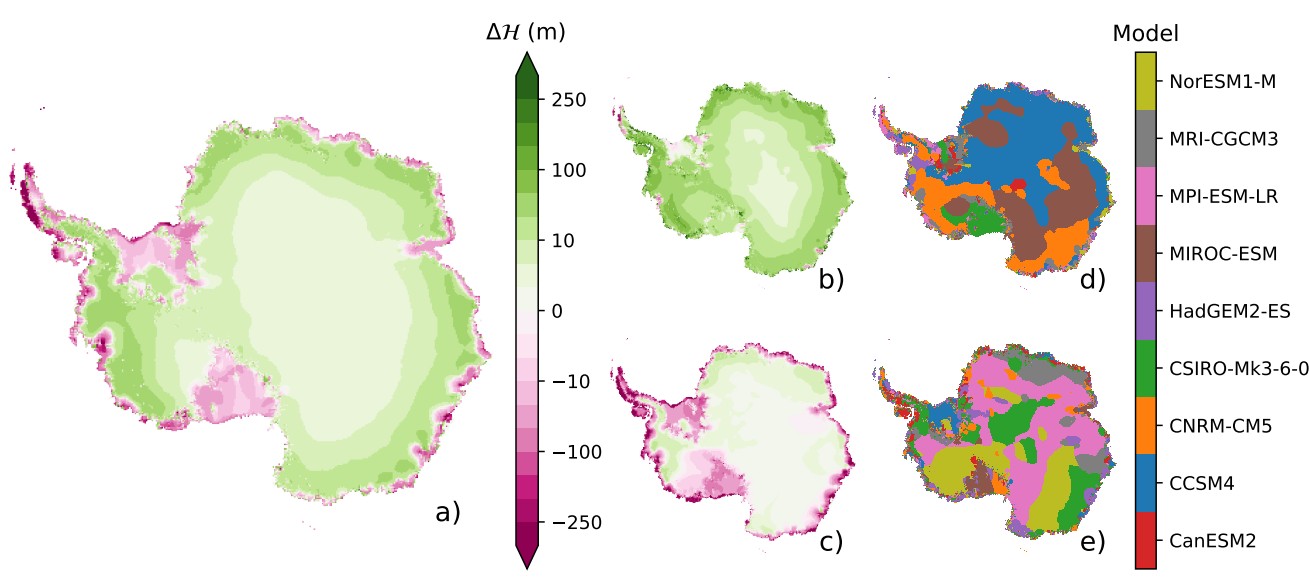

**Figure 10.** Ice thickness changes since 1850 under the RCP8.5 scenario for applied precipitation anomalies in the year 2200. Highlighted are the (a) ensemble mean, maximum (b), and minimum (c). The climate model that is used to drive the ice-sheet model simulation causing the maximum and minimum thickness are shown in (d) and (e), respectively, next to the ensemble maximum (b) and minimum (c).

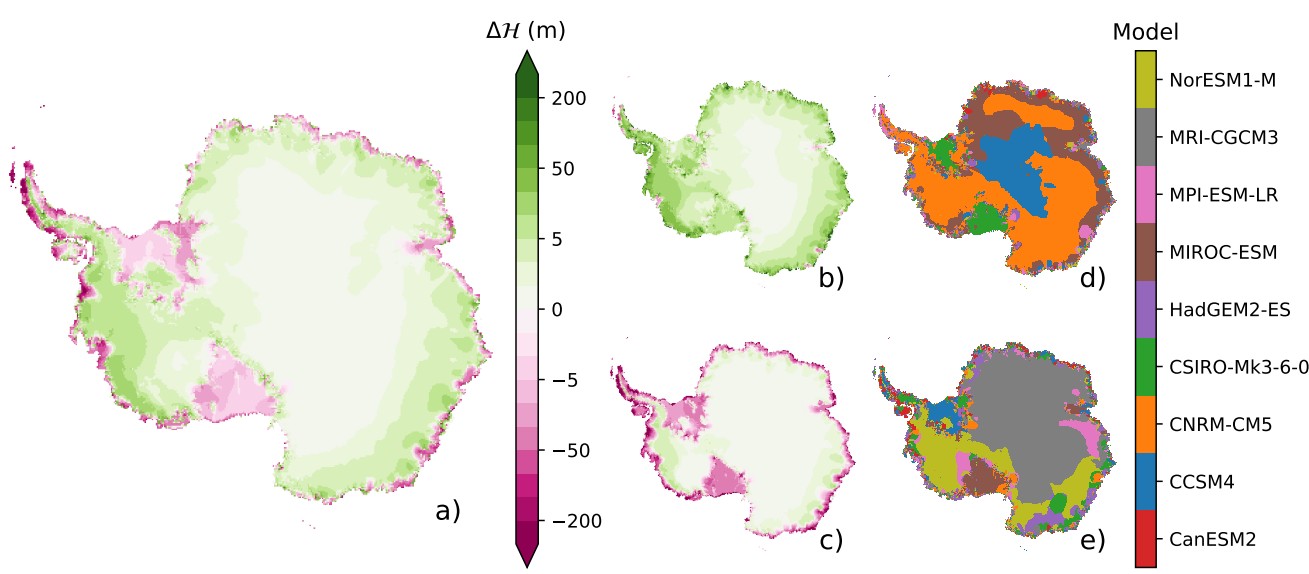

**Figure 11.** Ice thickness changes since the year 1850 under the RCP8.5 scenario in the model year 2100. Here the precipitation is scaled by the air temperature anomaly with a value of $5\,\%\,\mathrm{K}^{-1}$. Depicted are the (a) ensemble mean, maximum (b), and minimum (c). The climate model that is used to drive the ice-sheet model simulation causing the maximum and minimum thickness is shown in (d) and (e), respectively, next to the ensemble maximum (b) and minimum (c). This figure is similar to Figure 9, but there the results under precipitation anomalies are shown.

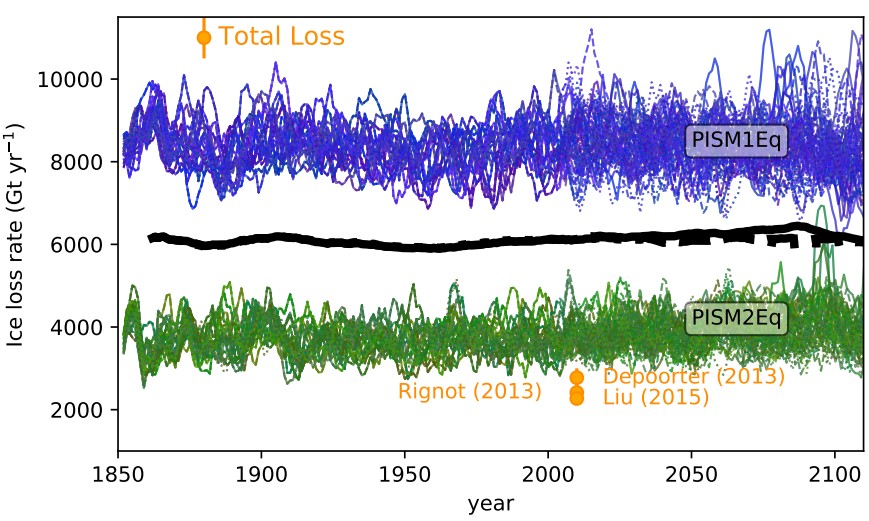

**Figure 12.** Temporal evolution of the ocean-driven ice loss rates of the ice shelves around Antarctica for the period from 1850 to 2100. The ice loss comprises iceberg discharge and basal melting of ice shelves. The thin blue lines are all ensemble members starting from the initial state PISM1Eq, where the Eigen-calving parameter amounts $10^{18}$, while the green lines are the corresponding simulations starting from PISM2Eq (Eigen-calving parameter $10^{17}$). A running mean with a window of 5 years has been applied for the thin lines. All simulations start under historical conditions and continue after 2005 under the RCP8.5 (solid lines), RCP4.5 (dashed lines) or RCP2.6 (dotted lines) scenario. The thick black lines represent the ensemble mean of the three future scenarios with a moving window length of 25 years. Recent estimates of the total loss rates (Top-left legend with the golden circles). Estimated uncertainties are given as vertical lines if the uncertainties are larger than the symbol size.

## Appendix A: Ocean Forcing

Regarding oceanic influence, we focus on the changes of the mean potential ocean temperature under the RCP8.5 scenario in a depth range between $150\,\mathrm{m}$ and $500\,\mathrm{m}$ (Figure 3), because these water masses flow into the ice-sheet cavities and are in contact with the ice shelve bases. Highest ocean temperature increases occur in the Bellingshausen and the Amundsen Seas as part of the West Antarctic Ice Sheet (WAIS) and some spots along the East Antarctic Ice Sheet (EAIS) according to observations (Schmidtko et al., 2014; Jacobs, 2006). In the Bellingshausen and the Amundsen Sea, warm water masses flow into ice-shelf cavities as indicated by observations (Arneborg et al., 2012; Thompson et al., 2018) and model simulations (Nakayama et al., 2018). These water masses drive the highest basal melting rates (Nakayama et al., 2014) that trigger potential the Marine Ice Sheet Instability (MISI) because WAIS has a retrograde bedrock topography. The tremendous ice shelves, Filchner-Ronne, Ross, and Amery, are influenced by moderate ocean temperature increases. However, our setup misses the interaction between the ice-shelf topography and the underlying dynamically evolving ocean. Hence, the setup does not describe related circulation changes that may bring warmer water masses into the ice cavities. For instance, it has been found that warmer water masses could find their way into these ice-shelf cavities and cause a strongly amplified basal mass loss under a changing climate (Hellmer et al., 2012). They have simulated an ocean warming by more then $2\,^{\circ}\mathrm{C}$ in the Filchner Trough (eastern Filchner-Ronne Ice Shelf). At the ice shelves edge of the Filchner-Ronne Ice Shelf, our CMIP5 data set maximum ocean temperature anomaly (Figure A1) of about $1.5\,^{\circ}\mathrm{C}$ generates a much weaker forcing.

## Appendix B: Spatial pattern of the Temperature Scaling of Precipitation for Individual Climate Models

If one calculates temperature scaling factors out of the CMIP5 model simulated air temperature and precipitation changes, it turns out that the temperature scaling factor of the precipitation is different for each model and therefore shows an inhomogeneous spatial pattern (Appendix Figure A3). Furthermore, the details of the scaling factors depend on the time period we chose as a reference, which drive our ice-sheet simulations, relative to the first or last 50 years of the corresponding piControl runs. If we alternatively compute the anomalies relative to averaged first 30 years of the historical period (1850–1879), we obtain also slightly different results. However, these differences do not significantly change the spatial structure. The choice of the baseline (first or last 50 years of piControl or first 30 years of the historical period) to compute the scaling distribution is of minor consequence. However, selecting the forcing data set from the pool of CMIP5 models determines the scaling distribution overwhelmingly. The across Antarctica averaged scaling factors reveal that the scatter range for one model is much smaller than the scaling values' distance among models (Figure 5).

The scaling across all model tends to be highest for the EAIS, where the part facing the Atlantic Ocean exhibits highest scalings (Figure 5). The WAIS has a lower scaling and the embedded region "Siple Coast" has on average the lowest scaling. There is a tendency for a higher scaling under a more vigorously changing climate across all regions, except for the smallest region "Ross." This tendency exists for the CMIP5 data set average and across models characterized by a larger than average scaling. Most models represent the detected precipitation deficit (shrinking precipitation rates), captured by reanalysis data and shallow ice cores in the "Siple Coast" region (Wang et al., 2017). Only NorESM1-M reproduces less precipitation (precipitation

deficit) under rising air temperatures across all future climate scenarios. When considering the whole Antarctica, the difference between the grounded ice sheet only and all glaciated regions (including ice shelves) is small.

The highest scaling spread between the first and last 50 years piControl reference period has MIROC-ESM across all inspected regions and scenarios (see scatter range in Figure 5), which is probably related to the pronounced trend of the global 2m-air temperature ($0.67°C$) between these two reference periods in our CMIP5 data set. Otherwise, the spread is related to enhanced/amplified long-term regional climate variability expressed by differing values in the reference period. For example, CCSM4 or MPI-ESM-LR are subject to a larger spread in the Atlantic sector of the EAIS, while in the neighboring Indian sector the variability is negligible. The higher spread of the smaller subregion Ross within the WAIS sector supports this interpretation (at least for the models CCSM4, CanESM2, HadGEM2-ESM).

There exists a tendency towards a higher scaling of coastal areas that are subject to incoming storm tracks, which potentially deliver heavier precipitation events that are also controlled by the rising topography height towards the interior of Antarctica. In the majority of the simulations, we identify a lower scaling in WAIS and also a low to negative scaling in the area of the Ross Ice Shelf and the adjacent parts of the WAIS.

**Appendix C: Ice Sheet Loss by Basal Melting of Ice Shelves and Iceberg Calving**

We turn our analysis to the individual mass balance terms: Iceberg calving, basal melting in the ice-shelf cavities, and surface mass balance. To recap: the surface mass balance is obtained by applying the individual spatial atmospheric model forcing on top of the reference fields obtained from RACMO, while the basal melting is calculated by adding ocean anomalies on top of the World Ocean Atlas climatology (Table 2). The calving is composed of three processes (thickness calving, Eigen-calving, kill mask calving) as part of the Parallel Ice Sheet Model (PISM) simulations. Here, the analysis focuses predominantly on the period from 1850 to 2100, because after 2100, we reapply the forcing from 2071–2100 recurrently.

Until 2100, the temporal evolution of the iceberg calving rates of individual ensemble members is subject to some variability, which is typical for such event-based mass losses. For some models, we could identify some reduced calving of $20\%$ around 1850 and 1970, and some enhanced calving of $25\%$ around 1920 and 2050. For individual ensemble members the temporal evolution of the calving rate is noisy and independent of the applied forcing scenario RCP2.6, RCP4.5, and RCP8.5 (Figure A12). Overall, the temporal evolution of the calving does not show a clear trend, and the average calving loss rate of the entire ensemble is about $5500 \, \mathrm{Gt \, year^{-1}}$ (Figure A12). Clearly separated are the calving rates of ensemble members starting from the initial state PISM1Eq or PISM2Eq. The members of the group starting from PISM1Eq have on average a calving rate of approximately $7500 \, \mathrm{Gt \, year^{-1}}$, while it amounts to about $3500 \, \mathrm{Gt \, year^{-1}}$ in the PISM2Eq. So a reduction of the Eigen-calving constant by an order of magnitude from $10^{18}$ (PISM1Eq) to $10^{17}$ (PISM2Eq) halves approximately the total calving rate, while in both cases the thickness calving is active for marginal ice-shelf point with a thickness of less than $150 \, \mathrm{m}$.

According to observational estimates control iceberg calving and basal ice-shelf melting the overall mass loss of Antarctica, while the relative contribution is the subject of current research. Depoorter et al. (2013) report a nearly equal share between calving ($1321 \pm 144 \, \mathrm{Gt \, year^{-1}}$) and basal melting ($1454 \pm 174 \, \mathrm{Gt \, year^{-1}}$) in the period 1995–2009, Rignot et al. (2013) detect

a slightly higher contribution from basal melting ($1325 \pm 235\,\mathrm{Gt\,year}^{-1}$ compared to calving with $1089 \pm 139\,\mathrm{Gt\,year}^{-1}$) in 2003–2008, while Liu et al. (2015) find that the basal melting ($1516 \pm 106\,\mathrm{Gt\,year}^{-1}$) contribution is twice as much as the calving ($755 \pm 24\,\mathrm{Gt\,year}^{-1}$) contribution (2005–2011).

     Both ensemble branches starting from PISM1Eq and PISM2Eq overestimate the currently observed calving rates of less than $1500\,\mathrm{Gt\,year}^{-1}$ (Depoorter et al., 2013; Liu et al., 2015; Rignot et al., 2013). Also the combined observed mass loss by calving

and basal melting of ice shelves, which is about $2500\,\mathrm{Gt\,year}^{-1}$ (Depoorter et al., 2013; Liu et al., 2015; Rignot et al., 2013), is on average smaller than the lower simulated calving rate from our ensemble members starting from PISM2Eq. Therefore, our ensemble mean ice loss rate exceeds current estimates, which could lead to an overestimation of the total sea-level rise in our simulations.

     The basal melting rate of floating ice shelves (hereinafter basal melting rates) is the second ocean-driven ice mass loss

process beside iceberg calving. In broad terms, the basal melt rate increases generally by $10\,\%$–$100\,\%$ over the period 1850– 2100 (Figure A13). In the beginning, the melting rises slowly because the additional ocean-temperature forcing remains weak (Figure3). Starting around the year 1970, the raise becomes nonlinear, and basal melting accelerates. The simulated historical trend is nearly independent of the initial state (PISM1Eq and PISM2Eq) as well as to the reference period selected for the computation of the ocean temperature anomaly. For each climate model scenario, the anomalies are computed relative to the

first or last 50 years of the pre-industrial climate (piControl) simulations. However, the reference state only matters for MIROC-ESM (first vs. last 50 years piControl), because this model is subject to a non-negligible trend ($0.08\,\mathrm{m}$) during the piControl phase. For instance, the average of the global absolute 2m-air temperature difference between the first and last 50 years of piControl amount to $0.17\,\mathrm{K}$ (median $0.12\,\mathrm{K}$) for all CMIP5 models considered in our study. In contrast, MIROC-ESM's value is $0.67\,\mathrm{K}$.

In future projections, the basal melting rate increases between $10\,\%$ and more than $100\,\%$ until the year 2100 relative to the 50 years reference period 1951–2000. The latter increase is consistent with results from dedicated ocean simulations. These simulations resolve ice shelves, include the ocean-ice-sheet interaction explicity, are driven by future projection from various climate models (Naughten et al., 2018; Hellmer et al., 2012).

     The basal melting rates increase until 2100, but then suddenly decrease back to 2071 values (Figure A13), since by experi-

mental design, the last 30 years of forcing (2071-2100) is repeated after year 2100. Also, for the basal melting the separation of ensemble members starting from PISM1Eq and PISM2Eq is self-evident. However, both groups are close to the ensemble mean, which is in contrast to the calving rate. The basal melting rates of all ensemble members underestimate the observational basal melting rates.

     Since, in general, the observed calving rate is lower than the basal melting rate, our model ensemble swaps the importance

of basal melting and iceberg calving. Also the sum of the calving rate and basal melting rate exceeds the observed estimates. Hence, our simulations could tend to overestimate ice loss and, ultimately, sea-level rise.

     The ensemble mean calving and basal melting rates stay nearly constant or reach a maximum of around 2100 and scenarios with a higher forcing (RCP8.5 vs. RCP4.5, for instance) cause more ice loss by both calving and basal melting. Beyond 2100, ice loss rates decrease in general (Figure A14). Since the temporal variability remains high also after 2100, our approach works

to construct the forcing beyond the year 2100 (see section 2: "Material and Methods"). To highlight the primary trend in the temporal evolution after 2100, a 250-year running mean is applied after 2100.

The basal melting rates of the stronger forcing scenario RCP8.5 show a minimum of around the year 3500 and increase afterward slightly, while the other scenarios (RCP4.5 and RCP2.6) indicate a tendency for stabilization at the end of our simulation in the year 5000 (Figure A14). Over the entire period, the basal melting rate is higher for the stronger forcing scenarios. This result reflects the dependence of the basal melting on the ocean temperature because a warmer climate scenario induces higher ocean temperature anomalies.

The calving rates before 2100 tend to be slightly higher for the RCP8.5 scenario. However, after 2100, we detect the sharpest fall of the ice loss rates for the scenario RCP8.5 and an intermediate decrement for RCP4.5 and a moderate reduction for RCP2.6 (Figure A14). Around 3000, RCP8.5 calving reaches its minimum, followed by an enhanced increase for 500 years and a moderate increase afterward. Scenarios with reduced radiative forcing reach the minimum later, so that RCP4.5 has its minimum around 3200, while RCP2.6 shows the minimum around 3700. At this time, the ensemble mean calving rates of the RCP4.5 and RCP2.6 are similar (please note that RCP2.6 does not include simulations driven by CCSM4). The trends of all scenarios converge around 4000.

In the long term, the most active basal melting occurs for the stronger forcing scenarios, while the highest calving occurs under scenarios with a lower forcing. The calving rate controls the evolution of the total ice mass loss in our simulations. Before the year 2100, RCP8.5 has the highest calving rates, while these are lowest shortly afterward. The ordering of the scenarios with the highest calving rates (RCP2.6) is those with the lowest forcing (RCP2.6) and vise versa (RCP8.5). The ensemble mean of the basal melting increases by $60\,\%$–$70\,\%$, $70\,\%$–$85\,\%$, and $90\,\%$–$115\,\%$, for RCP2.6, RCP4.5, and RCP8.5, respectively. The fractional calving change of the ensemble mean is between $+2\,\%$– $-4\,\%$, $+2\,\%$– $-10\,\%$, and $+2\,\%$– $-19\,\%$ for RCP2.6, RCP4.5, and RCP8.5, respectively. Across these, we detect that the most substantial ice-shelf area reduction occurs for RCP8.5 and the lowest for RCP2.6. Our simulations suggest that the warmer climate causes a stronger ice-shelf retreat and a stronger drop in the calving rate in the period, where the ice shelf could adjust to the quasi-equilibrium forcing. Based on these results we conclude: warmer climate drives more basal melting and enhances calving so that we obtain smaller ice shelves. The total area of ice shelves is, in general, smaller when a warmer climate scenario impacts these ice shelves (Figure A15) and the degraded total ice shelf area downgrades the calving probability. Ultimately, the integrated calving rate is lower under a warmer climate.

**Appendix D:  Bias-corrected Fluxes of Basal Melting and Calving**

Since the simulated ocean-driven basal melting rates are lower than observational-based estimates (Figure A13), the impact of flux corrected basal melting rates on the model results are discussed in the main text (Section 4.2.1 "Sea Level Contribution of Corrected Basal Melting" on page 19). This section describes the method.

Starting from original simulated ablation flux $F_{\mathrm{org}}$, which could be the basal melting flux $F_{\mathrm{org}}^B(t)$ or the iceberg discharge flux $F_{\mathrm{org}}^D(t)$, and the corresponding reference flux $F_{\mathrm{ref}}(t_{\mathrm{ref}})$ at time $t_{\mathrm{ref}}$, we define the following ratios. The fraction of the

temporal evolving flux ($F_{\mathrm{org}}(t)$) to the original flux at the reference time ($t_{\mathrm{ref}}$):

$$r(t) = \frac{F_{\mathrm{org}}(t)}{F_{\mathrm{org}}(t_{\mathrm{ref}})} \implies r(t_{\mathrm{ref}}) = 1, \tag{D1}$$

and the fraction of the original simulated flux to the reference flux ($F_{\mathrm{ref}}$)

$$q = q(t_{\mathrm{ref}}) = \frac{F_{\mathrm{ref}}(t_{\mathrm{ref}})}{F_{\mathrm{org}}(t_{\mathrm{ref}})}. \tag{D2}$$

The corrected flux $F_{\mathrm{cor}}$ using Equation D1 is defined as

$$F_{\mathrm{cor}}(t) = r(t) \cdot F_{\mathrm{ref}}(t_{\mathrm{ref}}), \tag{D3}$$

so that the flux difference $\Delta F(t)$ is

$$\begin{aligned} \Delta F(t) &= F_{\mathrm{cor}}(t) - F_{\mathrm{org}}(t) \\ &= F_{\mathrm{org}}(t)\left[\frac{F_{\mathrm{ref}}(t_{\mathrm{ref}})}{F_{\mathrm{org}}(t_{\mathrm{ref}})} - 1\right]. \end{aligned}$$

With Equation D2 we obtain

$$\Delta F(t) = F_{\mathrm{org}}(t)\left[q - 1\right]. \tag{D4}$$

To relate the sea-level change to the ice mass evolution, we define the ratio $p(t)$ of the sea level temporal deviation to the ice mass temporal deviation as

$$p(t) = \frac{\dfrac{dz_l(t)}{dt}}{\dfrac{dm_{\mathrm{ice}}(t)}{dt}}, \tag{D5}$$

where $z_l$ is the sea level and $m_{\mathrm{ice}}$ the total ice mass, which includes grounded and floating ice. We use here $p = \mathrm{median}(p(t))$ so that each ensemble member is characterized by one value for its entire time series. If $p = \frac{1}{\rho A_{\mathrm{oce}}}$, $100\,\%$ of flux difference (Equation D4) contributes immediately to the sea level of the global ocean with an area of $A_{\mathrm{oce}}$.

The total ice mass ($m_{\mathrm{ice}}$) changes are driven by four terms

$$\frac{m_{\mathrm{ice}}}{dt} = \underbrace{\left[F^{SMB}(t) + F^G(t)\right]}_{\text{unchanged under correction}} + F^B(t) + F^D(t),$$

where $F^{SMB}(t)$ is the surface mass balance flux and $F^G(t)$ the basal mass flux of grounded ice (Figure A9). We assume that these two terms in the brackets do not change regardless of the applied corrections to the last two terms $F^B$ and $F^D$. Hence the difference in the ice mass change is

$$\Delta\frac{dm_{\mathrm{ice}}}{dt} = \Delta F^B(t) + \Delta F^D(t) \tag{D6}$$

Now we relate the temporal evolution of the sea level to the total ice mass changes by utilizing the Equation D5

$$\frac{z_{l\,\mathrm{cor}}}{dt} = \frac{z_{l\,\mathrm{org}}}{dt} + p(t) \cdot \left[ \Delta F_{\mathrm{cor}}^{B}(t) + \Delta F_{\mathrm{cor}}^{D}(t) \right],$$

so that we obtain:

$$z_{l\,\mathrm{cor}} = z_{l\,\mathrm{org}} + \Delta z_l(t), \tag{D7}$$

where the sea-level difference $\Delta z_l(t)$ is

$$1220 \quad \Delta z_l(t) = \int_{t_0}^{t} p(\hat{t}) \left[ \Delta F_{\mathrm{cor}}^{B}(\hat{t}) + \Delta F_{\mathrm{cor}}^{D}(\hat{t}) \right] d\hat{t}. \tag{D8}$$

The Figures A5 and A7 depict the sea-level difference for two cases. If the additional mass loss contributes immediately to a rising sea level (Figure A7), the corresponding sea level rise of $30\,\mathrm{cm}$ would be larger than the actual sea level rise since 1850 of about $20\,\mathrm{cm}$ (Church and White, 2011).This case is not realistic because a melting floating ice shelf does not impact the sea level. Only the flow of grounded ice across the grounding line, to feed an ice shelf, or the direct loss of grounded ice
contributes to the sea level.

In contrast, the sea level changes hardly (Figure A5) if the deduced ratio $\overline{p(t)}$, which corresponds to the ratio defined in Equation D5. It is computed for each ensemble mean as median of its time series. Whether the ratio between ice loss and sea-level rise is constant under amplified basal melting of ice shelves is an open question. Strongly intensified ocean-driven ice loss will probably cause a retreating grounding line on a longer time scale, which ultimately releases grounded ice into the sea
and increases the sea level.

Figure A8 shows the proportion of the deduced ratio to the $100\,\%$ ratio. Only very few ensemble member lose about 15% of the maximum value of $p = 1/(\rho \cdot A_{\mathrm{oce}})$. In contrast, the mean and median value of this proportion is generally less then $5\,\%$. For all ensemble members driven by the precipitation anomaly, this proportion is on average $4.7\,\%$ with a median of $3.9\,\%$. It is even lower for ensemble members driven by the temperature-scaled precipitation. The median amounts $0.7\,\%$ and the
1235 corresponding mean is $0.9\,\%$. Please note that some ensemble members under the temperature-scaled precipitation are subject to a negative scaling. This result confirms the above presented low positive and negative scaling seen for restricted regions (Figure 5). It highlights also that simulations driven by temperature-scaled precipitation could show unexpected results.

## Appendix E: Appendix Figures

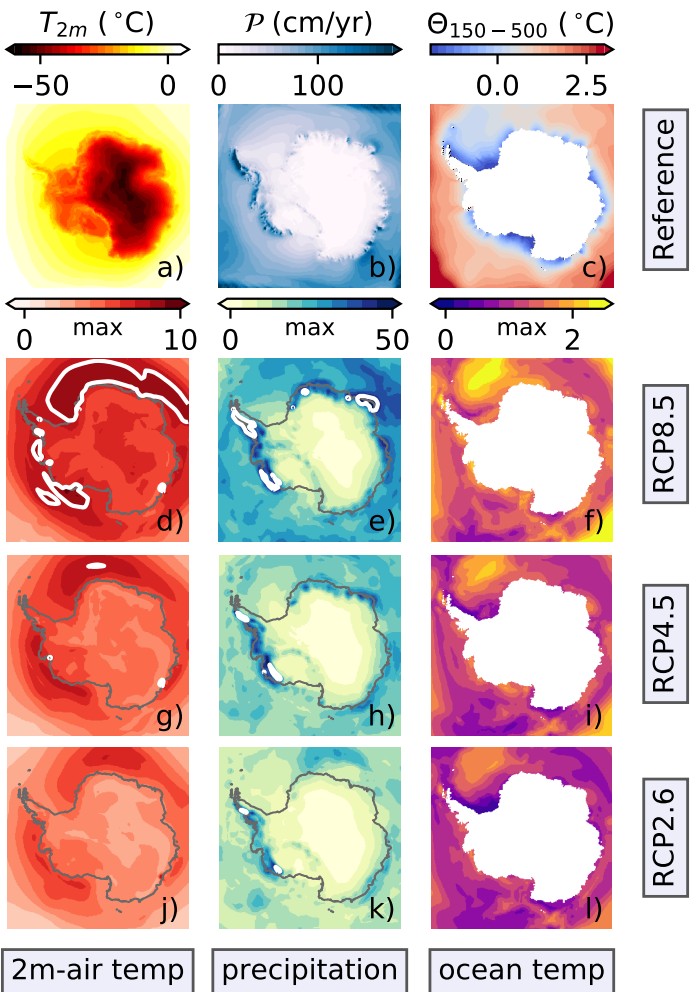

**Figure A1.** CMIP5 data set maximum anomalies (d–l) relative to the atmospheric (a, b) and oceanographic (c) reference forcing. The corresponding mean and minimum fields are depicted in the Figure 2 and Figure A2, respectively. The top row represents the reference fields to spin-up the ice-sheet model (Table 2). The 2m-air temperature (a) and the total precipitation (b) are mean fields from the regional RACMO model, while the ocean temperatures come from the World Ocean Atlas 2009 (c). Below each reference field, the related maximum anomalies are compiled for the period 2071–2100. Here, the second (third and fourth) row shows the anomalies for RCP8.5 (RCP4.5, RCP2.6). The dark-gray line follows the current coastline. All potential ocean temperatures (c, f, i, l) are a vertical mean of the depth interval from 150 m to 500 m. The white contour lines in the anomaly plots highlight the following thresholds. 2m-air temperature (d, g, j): 8°C; total precipitation (e, h, k) 50 cm year$^{-1}$. All these anomalies are the CMIP5 data set maximum of the models listed in Table 1; CCSM4 is not part of RCP2.6.

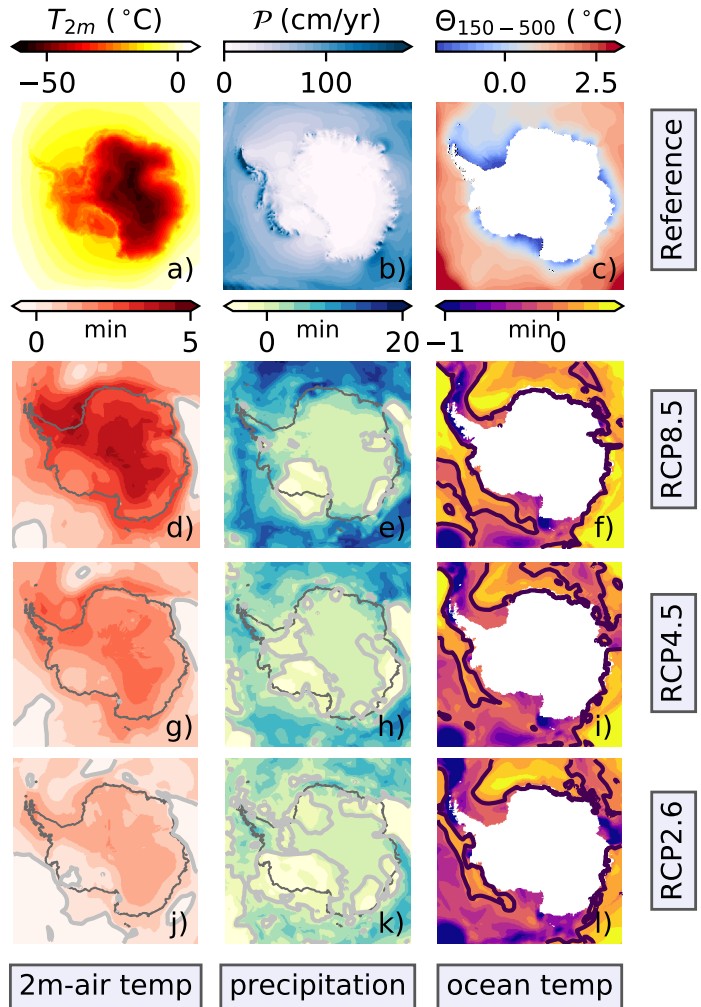

**Figure A2.** CMIP5 data set minimum anomalies (d–l) relative to the atmospheric (a, b) and oceanographic (c) reference forcing. The corresponding mean and maximum fields are depicted in the Figure 2 and Figure A1, respectively. The top row represents the reference fields to spin-up the ice-sheet model (Table 2). The 2m-air temperature (a) and the total precipitation (b) are mean fields from the regional RACMO model, while the ocean temperatures come from the World Ocean Atlas 2009 (c). Below each reference field, the related minimum anomalies are compiled for the period 2071–2100. Here, the second (third and fourth) row shows the anomalies for RCP8.5 (RCP4.5, RCP2.6). The dark-gray line follows the current coastline. All potential ocean temperatures (c, f, i, l) are a vertical mean of the depth interval from 150 m to 500 m. The light-gray lines in the anomaly plots highlight the following thresholds. 2m-air temperature (d, g, j): $0°C$; total precipitation (e, h, k) $0\,\mathrm{cm\,year^{-1}}$; potential ocean temperature $0°C$. All these anomalies are the CMIP5 data set minimum of the models listed in Table 1; please note that CCSM4 is not part of RCP2.6.

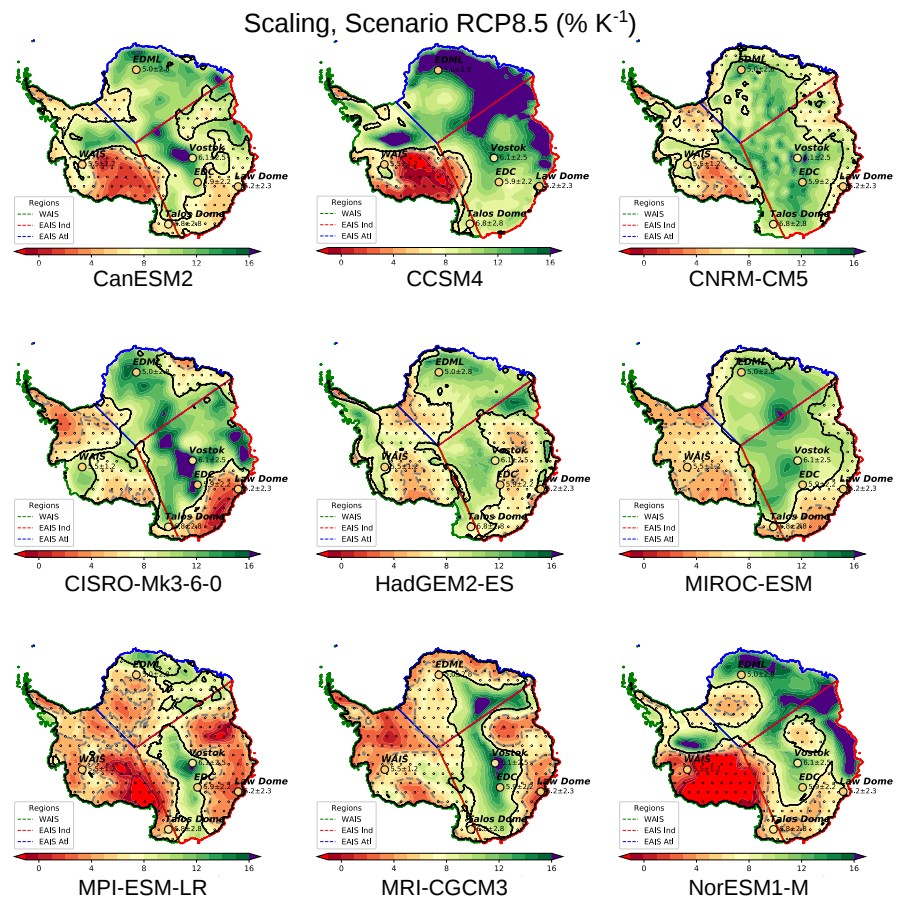

**Figure A3.** Air temperature-scaled precipitation under the RCP8.5 scenario for nine CMIP5 models (Table 1): Period 2051–2100. The ice-sheet simulations are driven by anomalies relative to the first 50 years of the related piControl climate scenario. In the dotted regions enclosed by black contours, the combined simulated scaling and the standard deviation contains the value of $5\,\%\,\mathrm{K}^{-1}$. Gray dashed lines follow this $5\,\%\,\mathrm{K}^{-1}$ contour. The scaling values deduced from ice cores are shown at their location (mean and the 2-sigma uncertainty). The regions named "WAIS," "EAIS Atl," and "EAIS Ind" are outlined by their green, blue, and red, respectively, boundaries (lower left legend). For further details, inspect section 3.2 "Precipitation Scaling Across Antarctica", please. Figure 4 shows the corresponding CMIP5 data set average. The contours of the Antarctic continent are deduced from Fretwell et al. (2013).

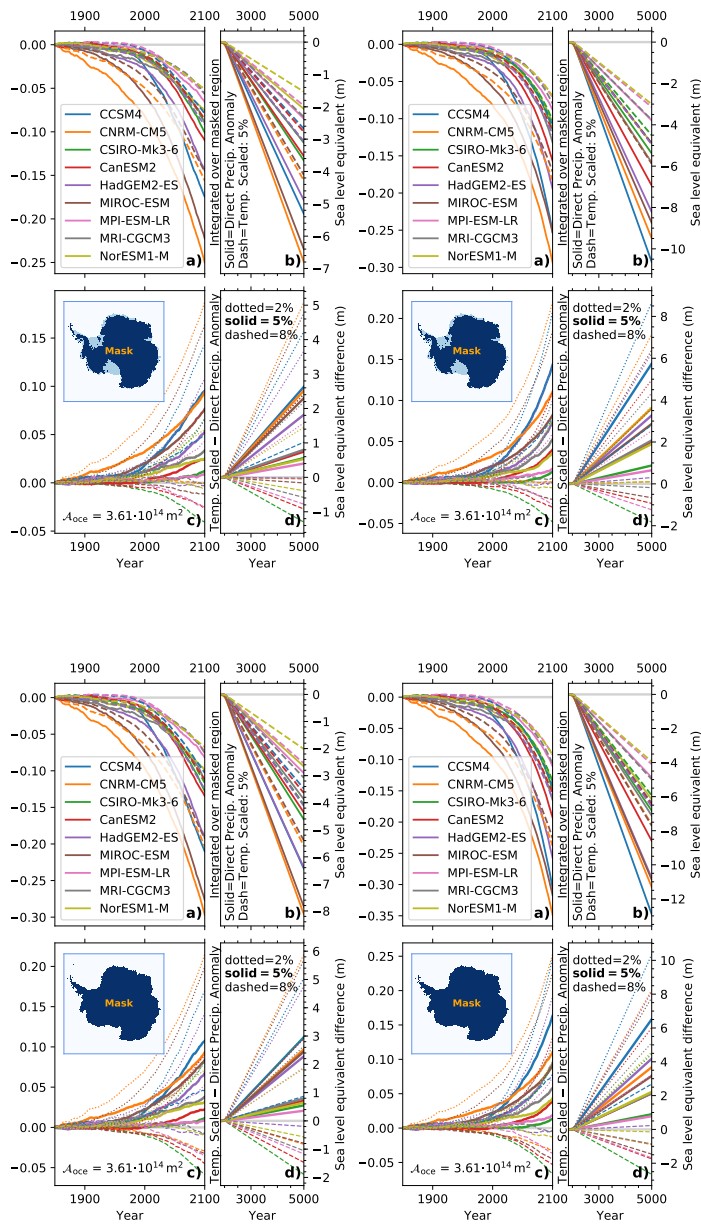

**Figure A4.** Integrated potential sea-level equivalent of the precipitation falling on Antarctica (see the mask in each subpanel) from the anomaly forcing (a, b: solid lines) and temperature-scaled precipitation (a, b: dashed lines). The potential sea-level impact between the anomalies and the temperature-scaled precipitation (c, d) is depicted for each CMIP5 model (legend on the lower left). The subpanels in the left and right columns show the results under the RCP4.5 and RCP8.5 scenarios, respectively. The upper row depicts the scaling for the entire Antarctic continent ("glaciered"), while the lower row is restricted to grounded ice. The lower left subpanel is identical to Figure 6. Please inspect this figure 6 for further details. The grounded and floating ice areas are derived from Fretwell et al. (2013)

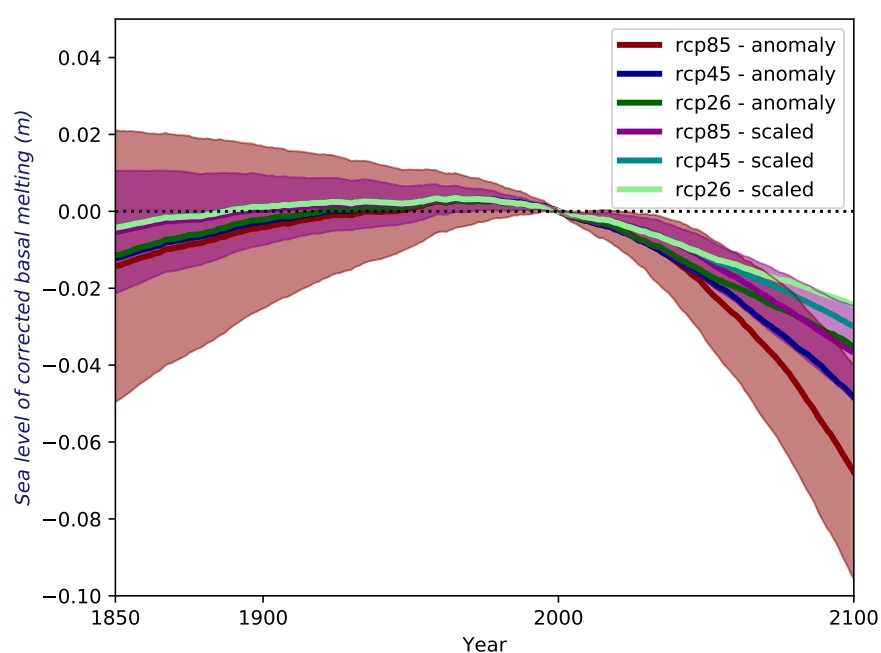

**Figure A5.** The sea level correction as defined by Equation D8 covers the period from 1850 until 2100. Here, the ratio $p(t)$ (Equation D5) is the temporal median for each ensemble member (please see Figure A7 for the corresponding figure assuming that all additional mass loss rises the global simulated sea level). The correction is computed relative to the simulated sea level for each individual simulation at the year 2000 as in Figure 8. The resulting simulated sea level for the entire period from 1850 until 5000 is depicted in the Figure A6. As reference value for the basal melting rate $F_{\mathrm{ref}}^B(t_{\mathrm{ref}})$, we use $1431\,\mathrm{Gt\,year^{-1}}$, which corresponds to the estimate of Depoorter et al. (2013) with $1454 \pm 174\,\mathrm{Gt\,year^{-1}}$, while it falls below the values of Liu et al. (2015) ($1516 \pm 106\,\mathrm{Gt\,year^{-1}}$), but is also exceed the rate of $1325 \pm 235\,\mathrm{Gt\,year^{-1}}$ reported by Rignot et al. (2013); our reference $F_{\mathrm{ref}}^B(t_{\mathrm{ref}})$ corresponds to the mean of all these basal melting estimates: $1431\,\mathrm{Gt\,year^{-1}}$.

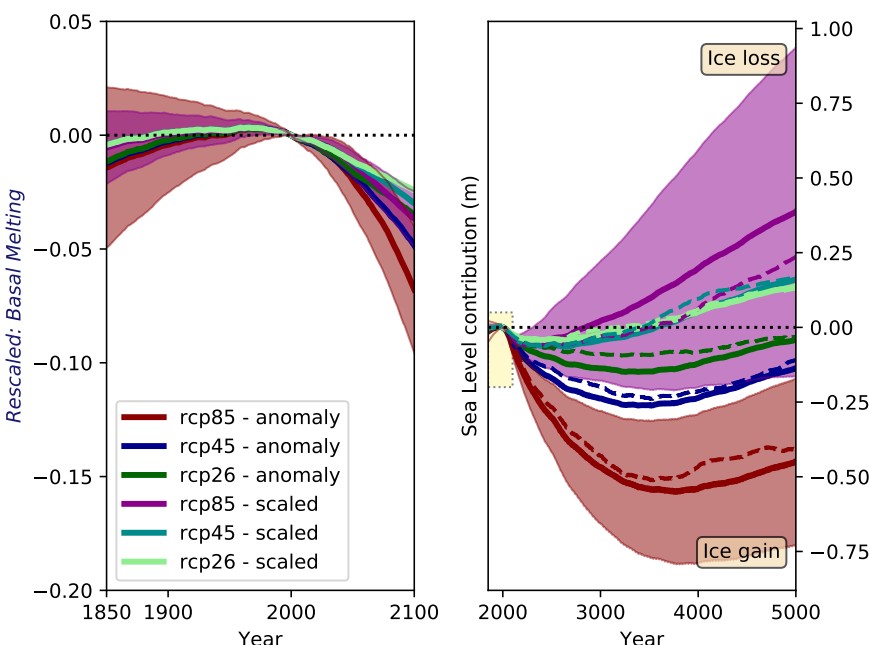

**Figure A6.** The simulated sea level considering the correction as defined by Equation D8 and Figure A5. The sea level (in meter) is computed relative to the simulated sea level for each individual simulation at the year 2000 as in Figure 8. Please inspect the Figure 8 for further details.

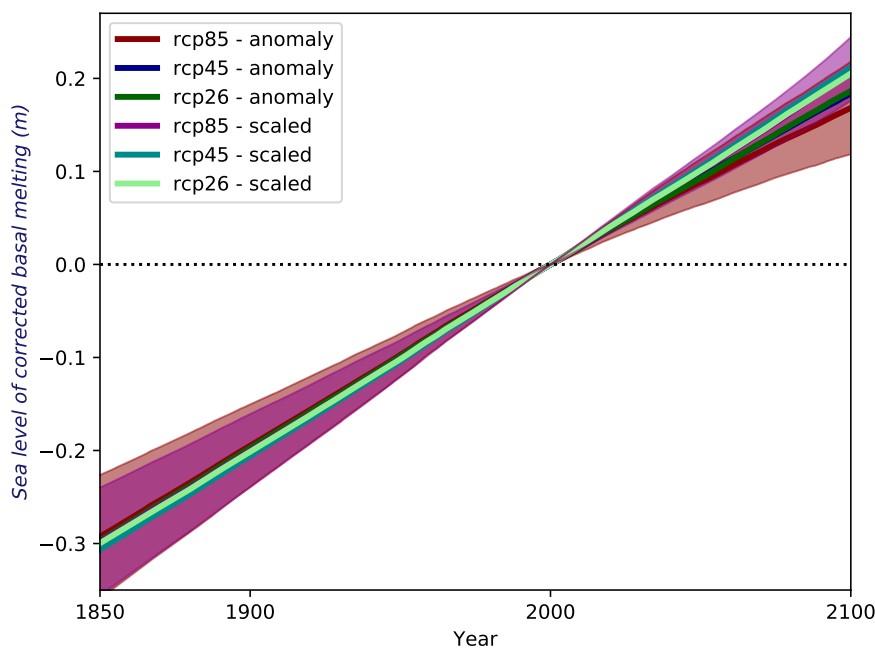

**Figure A7.** The sea level correction as defined by equation D8 covers the period from 1850 until 2100, where $100\,\%$ of the additional mass loss contributes immediately to a rising sea level, hence the ratio $p$ (Equation D5) equals $p = 1/(\rho A_{\mathrm{oce}})$. The correction is computed relative to the simulated sea level for each individual simulation at the year 2000 as in Figure 8. The corresponding Figure A5 depicts the case where the correction considers the actual deduced ratio $p(t)$.

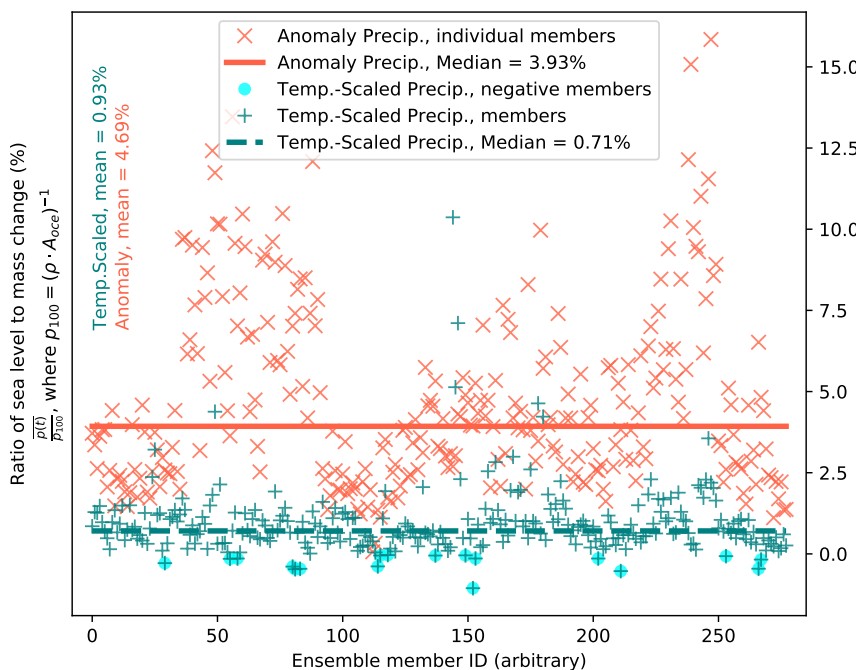

**Figure A8.** The ratio between the actual sea level contribution due to mass loss and the sea level equivalent of corresponding mass. Individual ensemble members are shown as crosses. A red "x" represents a member that is driven by the precipitation anomaly, while a blue-green "+" indicates those driven by the temperature-scaled precipitation. In the latter case, light blue circles highlight members with negative ratios. Vertical lines mark median values for these two groups; see also legend. The corresponding mean values are listed on the left side. The term $\overline{p(t)}$ is defined by the Equation D5 and $p_{100} = \frac{1}{\rho \cdot A_{\mathrm{oce}}}$, where $\rho$ is the density and $A_{\mathrm{oce}}$ represents the global ocean area.

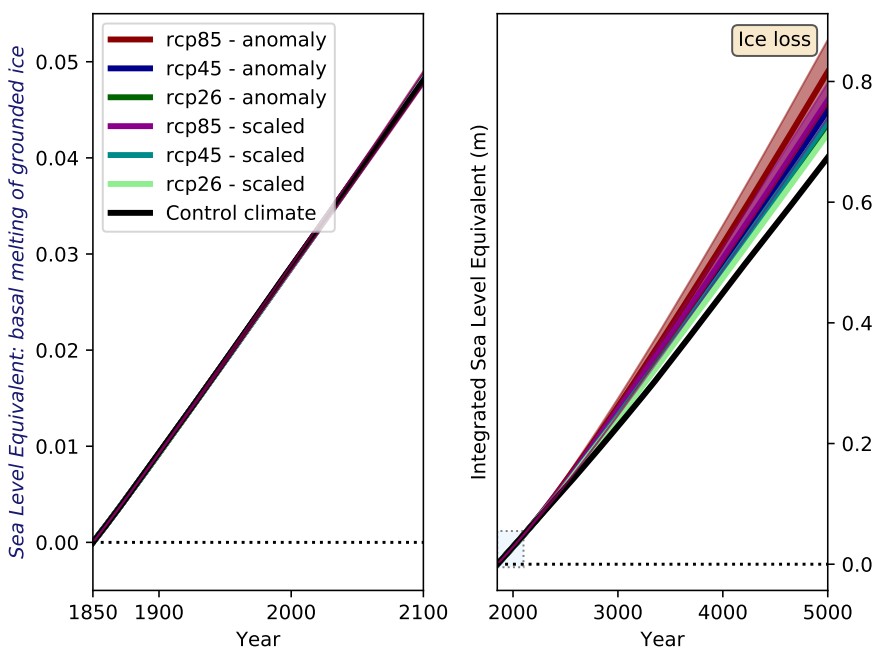

**Figure A9.** Cumulated basal melting of grounded ice as sea level equivalent. Shown are the results of the entire ensemble of ice-sheet simulations. The solid lines represent the ensemble averages for the applied precipitation anomalies and the temperature-scaled precipitation boundary conditions according to the legend (lower left). For the RCP8.5 scenario, the shading highlights the standard deviation (1-sigma) as a measure of the variability among the ice-sheet ensemble members driven by various climate models (Table 1).

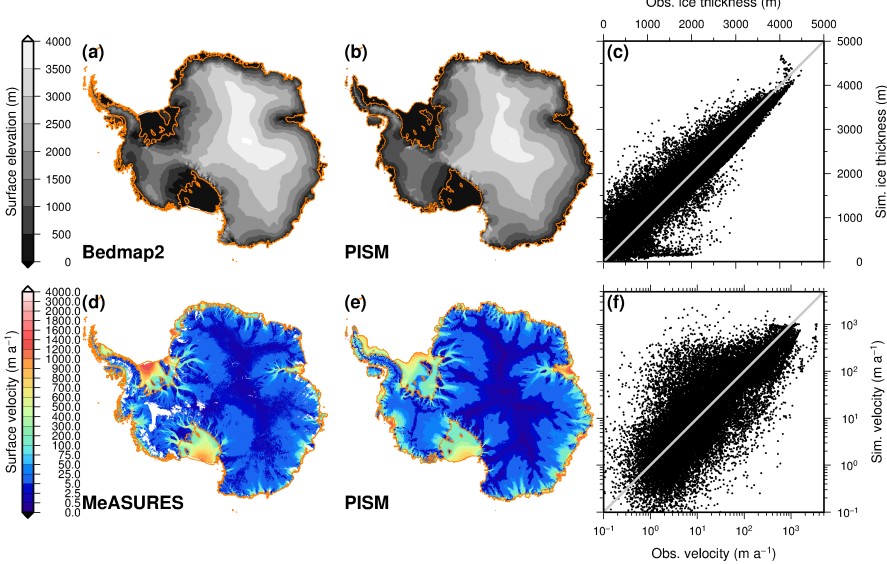

**Figure A10.** Comparison between the initial state PISM1Eq and observational estimates. The top row depicts the surface elevation: a) Bedmap2 data set (Fretwell et al., 2013), b) simulated ice elevation in PISM and c) point-wise comparison. The lower row shows the surface velocity distribution: d) Observations (Rignot et al., 2016), e) simulation and f) point-wise comparison; please note that both axes are logarithmic. For the point-wise comparison, the observations follow the x-axis (abscissa) while simulated values follow the y-axis (ordinate). Essential information about the initial ice-covered area and volume and its comparison with the other initial state PISM1Eq (Figure A11) is listed in table A1.

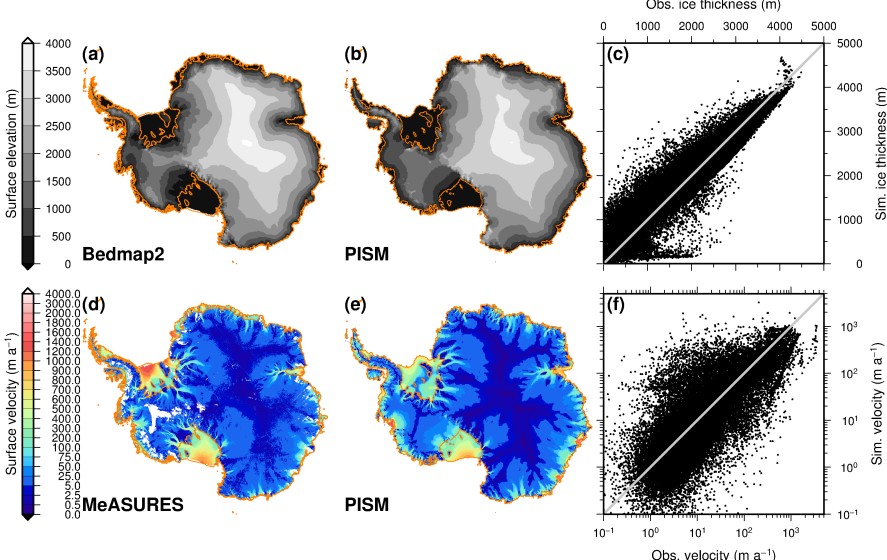

**Figure A11.** Comparison between the initial state PISM2Eq and observational estimates. The top row depicts the surface elevation: a) Bedmap2 data set (Fretwell et al., 2013), b) elevation in PISM and c) point-wise comparison, while the lower row shows the surface velocity: d) Observations (Rignot et al., 2016), e) simulation and f) point-wise comparison. Please see also Figure A10 (PISM1Eq) for further details, and inspect the table A1 for essential information and a comparision between both states.

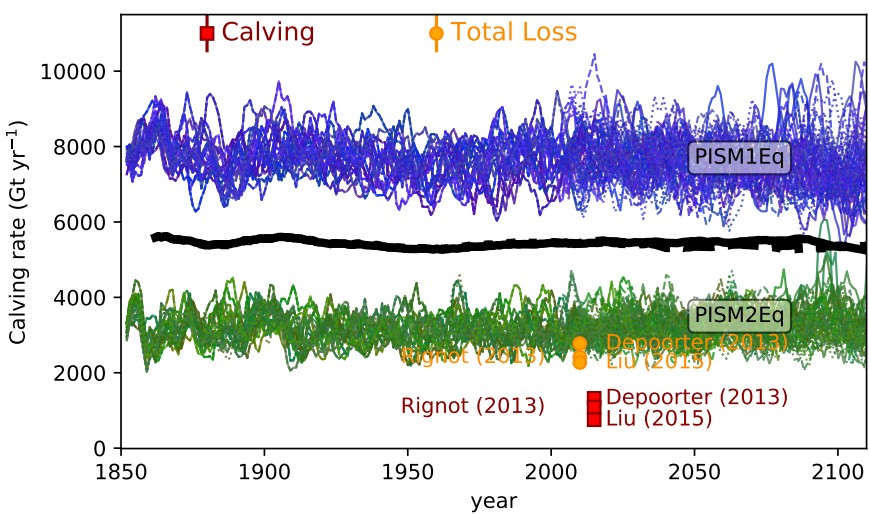

**Figure A12.** Temporal evolution of Antarctic-wide calving rates for the period from 1850 to 2100. All simulations start under historical conditions and continue after 2005 under the RCP8.5 (solid lines), RCP4.5 (dashed lines) or RCP2.6 (dotted lines) scenario. After the year 2100, the forcing of the last thirty years until 2100 drives the model recurrently. The thin blue lines are all ensemble members starting from the initial state PISM1Eq, where the Eigen-calving parameter amounts $10^{18}$, while the green lines are the corresponding simulations starting from PISM2Eq (Eigen-calving parameter $10^{17}$). A running mean with a window of 5 years has been applied for the thin lines. The thick black lines represent the ensemble mean of the three future scenarios with a moving window of 25 years. The applied running means shift the apparent maximum backward in time so that it occurs visually before the year 2100. Recent estimates of the total (orange circles) and the calving (red squares) ice mass loss are given for three studies (legend at the top). Vertical bars depict the reported uncertainties of the shown estimates by Liu et al. (2015); Depoorter et al. (2013); Rignot et al. (2013).

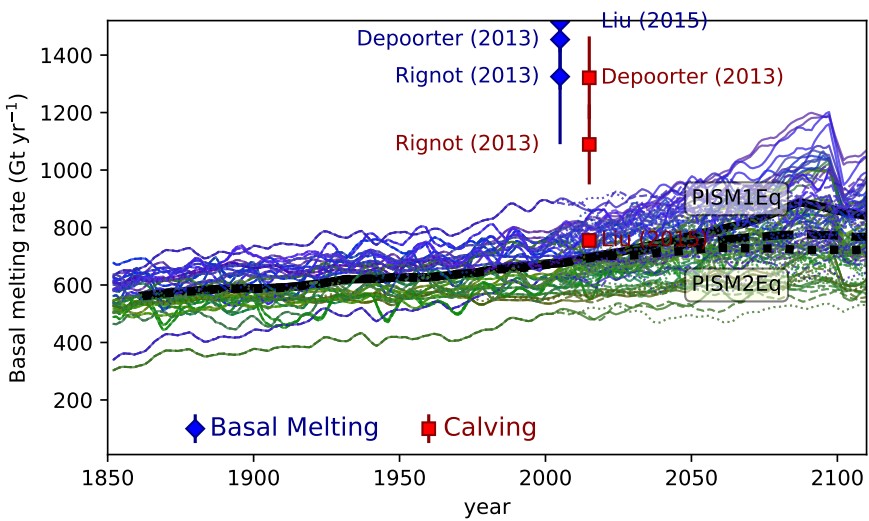

**Figure A13.** Temporal evolution of the basal melting rates in ice shelves around Antarctica for the period from 1850 to 2100. All simulations start under historical conditions and continue after 2005 under the RCP8.5 (solid lines), RCP4.5 (dashed lines) or RCP2.6 (dotted lines) scenario. After the year 2100, the forcing of the last thirty years until 2100 drives the model recurrently. The thin blue lines are all ensemble members starting from the initial state PISM1Eq, where the Eigen-calving parameter amounts $10^{18}$, while the green lines are the corresponding simulations starting from PISM2Eq (Eigen-calving parameter $10^{17}$). A running mean with a window of 5 years has been applied for the thin lines. The thick black lines represent the ensemble mean of the three future scenarios with a moving window length of 25 years. The applied running means shift the apparent maximum backward in time so that it occurs visually before the year 2100. Recent estimates of the basal melting (blue diamonds) and the calving (red squares) ice mass loss are given for three studies (legend at the bottom, Liu et al. (2015); Depoorter et al. (2013); Rignot et al. (2013)). Related uncertainties are given as vertical lines if the uncertainties are larger than the symbol size.

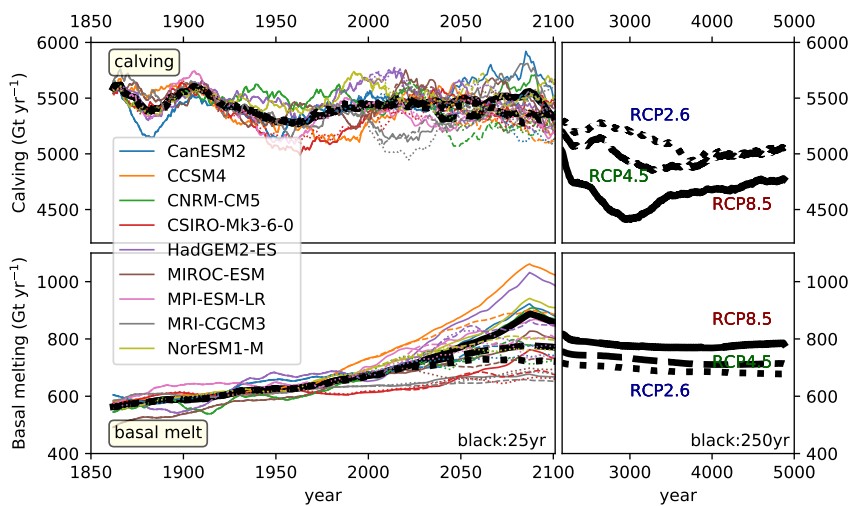

**Figure A14.** Long-term temporal evolution of the ensemble mean basal melting and calving ice loss rates from 1850 to 2100 and beyond until the year 5000. The upper panels show the calving rates, while the lower panels depict the basal melting rates. In the left columns, individual model simulations (colored lines according to the legend) are grouped together, while the thick black lines are the overall means as shown in the corresponding figures Figure A12 and A13. For the ensemble means of the period 1850–2100, a smoothing with a moving window of 25 years is applied (left column), while the smoothing window length is 250 years for the right column covering the period from 2100 until 5000. The applied running means shift the apparent maximum backward in time so that it occurs visually before the year 2100. Solid (dashed, dotted) lines represent the RCP8.5 (RCP4.5, RCP2.6) scenario.

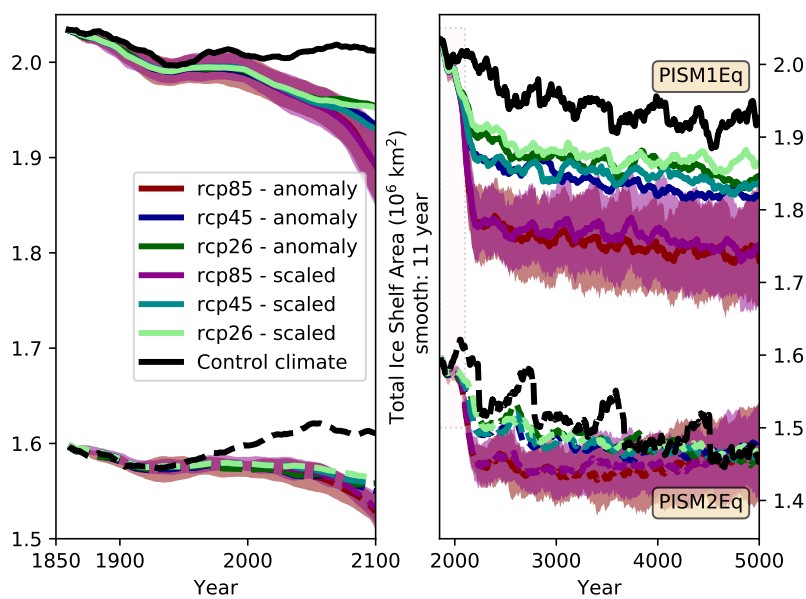

**Figure A15.** Area of floating ice shelves. A running mean window of 11 years is used for the entire ensemble of ice-sheet simulations. The solid lines represent the ensemble averages for the PISM1Eq starting conditions, while the dashed lines are the corresponding PISM2Eq condition. For the RCP8.5 scenario, the shading highlights the standard deviation (1-sigma) as a measure of the variability among the ice-sheet ensemble members driven by various climate models (Table 1). Please note the different axes for both subfigures. The dashed frame in the right subfigure depicts the value range of the left subfigure.

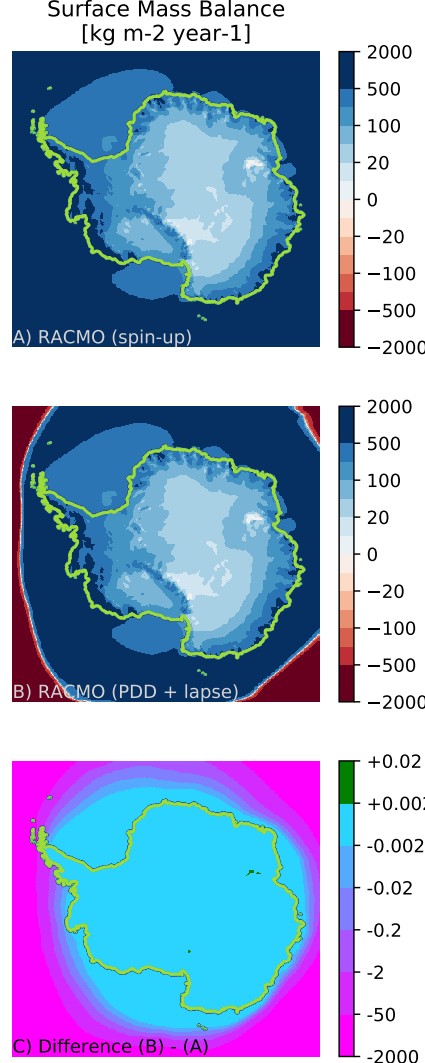

**Figure A16.** Surface mass balance (SMB) used during the spin-up (upper subplot, a), the surface mass balance computed via the here applied positive degree day (PDD) approach (middle subplot, b) under pre-industrial conditions and the difference between these (lower subplot, c). For the subplot under pre-industrial conditions (middle subplot, b), also the lapse correction is active, which does not impact the results because the height difference is neglectable initially. The unit of the surface mass balance and its difference are $\mathrm{kg\,m^{-2}\,year^{-1}}$; note $1\,\mathrm{kg\,m^{-2}\,year^{-1}}$ equals $1\,\mathrm{mm(WE)\,year^{-1}}$. In each subplot, the light-green contour lines represent the outer edge of the ice sheet or ice shelves, respectively. Approximately south of the annual sea ice edge, the difference between both SMB fields is essentially zero except for two restricted areas. One near the Amery Ice Shelf and the other at the Transantarctic Mountain Range east of the Ross Ice Shelf. These two regions are characterized by a negative SMB in the spin-up data distribution (upper subplot, a), which is absent in the PDD-deduced SMB (middle subplot, b). Besides, only the northern tip of the Antarctic Peninsula experiences a different forcing. Further north, the difference is significant; however, this difference does not impact the Antarctic ice sheet.

## Appendix B: Appendix Table

**Table A1.** Characteristics of the both initial states PISM1Eq (Figure A10) and PISM2Eq (Figure A11). These are the total areas covered of grounded ($A_g$) and floating ice ($A_f$). It shows the volumes of all grounded ice ($V_g$), grounded ice above the sea level of $z = 0$ ($V_{g0}$), and all floating ice ($V_f$). The last row presents the ratio, expressed as a percentage value, between grounded ice above the sea level and all grounded ice. The right column represents the ratio of the quantities between both initial states.

| Quantity | PISM1Eq | PISM2Eq | Ratio (PISM1Eq/PISM2Eq) |
|---|---|---|---|
| Area ($A_f$): grounded ice ($km^2$) | $1.255 \cdot 10^7$ | $1.257 \cdot 10^7$ | 0.9985 |
| Area ($A_g$): floating ice ($km^2$) | $2.005 \cdot 10^6$ | $1.569 \cdot 10^6$ | 1.278 |
| Volume ($V_g$): grounded ice ($km^3$) | $2.588 \cdot 10^7$ | $2.605 \cdot 10^7$ | 0.9936 |
| Volume ($V_{g0}$): grounded ice above $z = 0$ ($km^3$) | $2.313 \cdot 10^7$ | $2.325 \cdot 10^7$ | 0.9947 |
| Volume ($V_f$): floating ice ($km^3$) | $6.681 \cdot 10^5$ | $5.421 \cdot 10^5$ | 1.232 |
| Ratio of grounded ice: $V_{g0}/V_g$ (%) | 89.35 | 89.25 | 1.001 |