# Peer review of "Future Sea Level Contribution from Antarctica inferred from CMIP5 Model Forcing and its Dependence on Precipitation Ansatz"

_Earth System Dynamics, 2019_

## Referee Comment (RC1) · Torsten Albrecht (Referee) · 8 Feb 2020

**General comments:**

Rodehacke and colleagues investigate the effects of multiple climate model forcings (from CMIP5) in Antarctica. They assess the spatial heterogeneity in temperature and precipitation estimates over the period 1850-2100 for different emission scenarios and the spread among the 9 selected climate models. The ratio of precipitation anomalies and temperature anomalies is compared to paleo estimates at 6 ice core locations and regional variations from the spatial mean are discussed. Applied to the ice sheet model

PISM they run an ensemble of simulations up to the year 5000 with both the anomaly forcing fields from the climate models and the simplified (spatial mean) parameterization (as often used in previous studies) and find quite some differences in the projected ice mass changes (converted in units of sea-level equivalents).

While in this study the temperature-scaled precipitation results in long-term ice losses, the directly applied precipitation anomalies generate net mass gains. Given the numerous previous projection studies, this is a surprising result. However, for given model settings this discrepancy can to some extent be explained in the manuscript.

Overall, the study is well structured and the manuscript clearly-arranged. The main manuscript is separated into introduction, material and methods, results and discussions, conclusions and an appendix part. Due to its length of 42 pages including figures and references and 20 pages in the Appendix, it is sometimes difficult for the reader to follow the line of thought. The conclusions with almost 3 pages should be condensed, many discussed aspects could be merged into the introduction and discussion part. In general, the manuscript needs some additional work to improve the readability and to clarify the main key messages for the reader and avoid redundant informations. Also typos and the german-style syntax sometimes hampers the reader to fully grasp the content of the manuscript.

Figures have good quality and are informative, some are overloaded with up to 27 curves. Literature is sufficiently covered with 97 references. The investigation of the impact of climate boundary conditions on the future evolution of the Antarctic ice sheet supports the publication in ESD. However, as the main focus seems to be on the evaluation of climate model result and the systematic and comprehensive sensitivity analysis of the ice sheet model to the two different types of precipitation forcing, this study would also very well fit into a model-specific journal like GMD.

This study by Rodehacke et al. has the potential to be a valuable contribution to the scientific community of ice sheet modelers, as it considers relevant aspects of com-

monly used boundary conditions with potentially serious consequences for estimates of future sea-level change.

I encourage the authors to consider the following detailed suggestions and to improve the manuscript accordingly.

**Specific comments:**

1. The title should be refomulated. It is not easy to understand the content before having read the abstract. Also I wonder, if the more word "Ansatz" is commonly known in the wider scientific community apart from mathematicians. I would suggest: "Future sea-level contribution from the Antarctic Ice Sheet for different precipitation forcings based on CMIP5 models"

2. As a surprising result the simulated Antarctic Ice Sheet gains mass under future global warming for directly applied climate model anomalies (temperature and precipitation). As this part has some delicate political implications it needs a clear discussion of the responsible model settings.

2a. Although the equilibrium state fits well observations of ice thickness and grounding line, the involved mass fluxes may not. Total ice loss rates by melting and in particular by calving are overestimated by a factor of 2-3 depending on the used eigencalving rate constant. Hence also the surface mass balance seems to be overestimated accordingly. The authors imply that the uncertainties in the regional climate model results (RACMO), which are used as a present day reference field, are large enough to overestimate in particular the large slow-flowing and very dry inner-continental regions of the EAIS, where small absolute changes in precipitation can have large consequences for the total mass balance of the equilibrium state. Also, it is not clear from the description in the manuscript how the yearly cycle in the PDD scheme is estimated from the climate models (annual mean and summer temperatures) in order to obtain estimates of the

surface mass balance components for given air temperature and precipitation forcing. A potential misfit in boundary conditions may be compensated for by a well-chosen set of model parameters, such that the equilibrium state bounds observational constraints. However, this potential overfitting of the initial equilibrium state may then have consequences for the projected ice mass changes, as the authors already speculate. In general, the equilibrium state method favors rather stable ice sheet configurations, which may not be realistic.

2b. The authors state one main difference to previous studies related to the forcing after the year 2100, which is commonly extrapolated into the future, while in this study it remains quasi constant (within 30 years variability). There seems to be another important difference in the methodology of this study in comparison to other studies. All climate model anomalies are inferred with respect to a preindustrial control simulation. However, the reference climate of the 19th century does not really match the modern climate, which the used mean background fields (from RACMO as mean over period 1979–2011 and from World Ocean Atlas as climatological mean) are related to. This precedure minimized the shock at the beginning of each simulation at 1850, but it adds some anomaly to the present-day background field when arriving at present-day in the simulations and it overestimates the future temperatures and precipitation rates applied to the Antarctic Ice Sheet. I encourage the authors to run some test simulations with shifted anomaly (negative anomaly in the preindustrial era and vanishing anomaly in present-day period).

2c. Regarding the basal melt parameterization, no details on sensitivity can be found in this study nor in the cited study by Sutter et al., 2019. What technique is used to extrapolate ocean temperatures into the ice shelf cavities? Are basin-wise overflow depths considered? Are extrapolated ocean temperatures vertically interpolated at the ice shelf base? Can refreezing occur? A simimar melt parameterization with quadratic dependency on thermal forcing has been calibrated in Jourdain et al., 2019 (https://doi.org/10.5194/tc-2019-277). They show that the choice of the particular parameterization and the associate parameters can have a huge impact on the ice sheet response. The authors discuss that the used melt parameterization may underestimate the melting and therefore apply bias-corrected melt rates as a sensitivity check, which does not cause considerable changes in the ice sheet response. The effect of basal melt could be also strongly intensified by using the melt interpolation across the grounding line (optional in PISM).

3. The authors define different ways of expressing (integrated) mass changes in terms of sea-level equivalent changes and I would wish that it should be made clear when just a theoretical unit conversion is applied (potential sea level change) or when it is an actual sea-level contribution in terms of projected ice mass change. And if the latter it should be clearly defined whether only ice masses above flotation are considered. What diagnostic has been in fact used here?

4. The manuscript states that the used coarse resolution of 16km may have consequences for the adequate representation of ice stream dynamics. I assume that this model choice is a consequence of the initial equilibriumn state, which requires hundred thousands model years to evolve. The authors state that basal resistance is described by a Mohr-Coulomb law with plastic till, but they do not discuss relevant parameters involved, such as the till friction angle or the till water decay rate. What is the vertical resolution of the enthalpy module? These parameters can strongly affect the ice stream dynamics also for coarse resolutions.

5. The authors use a rather old PISM version (v0.7), most likely for consistency reasons (initMIP and other model intercomparisons). However, PISM has evolved over the last years and some relevant aspects have been improved, which may affect the results of this study. For instance, the authors mention a bug in the elastic part of the LC solid Earth model, but also the viscous part was flawed and considered changes in ice shelf thickness as loads. Accordingly strong melt would cause uplift of the cavity bed and hence result in a stabilized grounding line. Also the till water distribution along the grounding line has been fixed meanwhile causing a much higher grounding line

sensitivity. I guess also the sea-level potential diagnostic has been fixed meanwhile (now substracting the part below flotation). These are many good arguments in favor of a more recent PISM version and they suggest that Antarctic Ice Sheet simulations could respond with much higher sensitivity to the same forcing applied.

**Technical corrections:**

l.2: "heavier precipitation fallen on Antarctica will counteract any stronger iceberg discharge..."→ "precipitation will likely increase even more and may counteract stronger iceberg discharge..."

l.3: "from nine CMIP5 models future projections" → "future projections from nine CMIP5 models"

l.5 : "The spatial and temporal varying climate forcings drive ice-sheet simulations. Hence, our ensemble inherits all spatial and temporal climate patterns, which is in contrast to a spatial mean forcing.:" → "The spatially and temporally varying climatic forcing drive the ice-sheet simulations, such that all climate patterns are represented in our ensemble, which is fundamentally different from using spatial means as forcing."

l.7: Regardless of the applied boundary condition and forcing, some areas will lose ice in the future, such as the glaciers from the West Antarctic Ice Sheet draining into the Amundsen Sea." → ..., our ensemble study suggests, that some areas will lose ice in the future, ...

l.10: "This strip also shows..." instead of using "... too."

l.25: "How strong the precipitation grows in a warming atmosphere, may be explained by the dissimilarity between the applied methods to describe the precipitation." → The discrepancy of the simulation results between the applied methods to describe the precipitation illustrates the uncertainty of the possible range of future precipitation

growth in a warming atmosphere.

l.30: "...impacts globally numerous economic activities..." → "...impacts numerous economic activities globally..."

l.31: "... or dedicated model simulations of, for instance, ice-sheet models." → "... or process-based model simulation, e.g. ice-sheet models."

l.34: "... are simplified descriptions by analytical equations" → "... are either simplified descriptions based on linear multiple-regression analysis ... or"

l.36: "The simplified forcing, which usually does not show a dedicated spatial structure" → As surface elevation is a key variable in those parameterizations, the geometry of the ice sheet in fact leave some characteristic spatial structure.

l.58: "temperature scaling" → "temperature scaling factor for precipitation" or "precipitation-temperature scaling"

l.71: Add comma before "probably"

l.124: Maybe omit "full" here.

l.126: As in the title, I would recommend to use: "The type of precipitation forcing" or "the used method for applying precipitation forcing" instead of "the Ansatz of the precipitation".

l.130: "The latter is common, while some keep the surface mass balance constant." → "The latter approach is commonly used, in particular in paleo applications, while some sensitivity studies keep the surface mass balance constant."

l.138: It could help the reader to have some definition of the piControl simulation here, e.g. "pre-industrial coupled atmosphere/ocean are performed at constant pre-industrial $CO_2$ levels for x model years".

l.140: "... differ commonly marginally." → "...show in general marginal differences."

l.144: How does this extrapolation works? Is there a diffusion scheme applied for each vertical ocean temperature level? What are the source regions, the continental shelf or also the deeper ocean regions (this is not so clear from Fig. 3e), which are separated from the deeper cavity regions by the continental shelf? There is also no detailed description in Sutter et al., 2019, even though sub-shelf melting is a key process here.

l.146: "... following the positive degree day (PDD) approach, where the annual 2m-air temperature standard deviation comes from daily CMIP5 model values." Does this mean that every year one different PDD standard deviation is applied to the whole computational setup or is it grid-cell wise? In l.143 it is mentioned that "annual mean forcing" is ued, but what about the summer temperature anomaly to estimate the yearly cycle?

l.148: "16 km" → What is the reason for this relatively coarse resolution, the availability of an equilibrium state?

l.149: "utilizes" → "applies"

l.151: Also the viscous part in v0.7 was somewhat unrealistic, as also ice shelf thickness change has been considered as loads in the LC bed deformation model, which has strong effects on grounding line sensitivity.

l.154: "... pressure-dependent melting temperature" → Add "...of the ice"

l.157: "...while the grounding line position is determined on a sub-grid space (Feldmann et al., 2014)." → Add "... to interpolate basal friction."

l.161: "...stress field divergence... " → "... divergence of the strain/velocity field" or "trace of the strain-rate field"

l.161: You should add units "m s" here as the Levermann et al. 2012 paper uses "m a".

l.162: "(PISM1Eq and PISM2Eq)" → This can be confusing, either you switch the order here or the order of the eigencalving constants in the sentence before.

[Figure]

l.163: "Ocean temperatures from the World Ocean Atlas 2009 (Locarnini et al., 2010) and the multi-year mean surface mass balance (SMB) from the RACMO 2.3/ANT model (Van Wessem et al., 2014) drive PISM during spin-up (Table 2)." → Hence, this is a present-day forcing equilibrium.

l.169: "... releasing less carbon dioxide." → Maybe add "(e.g. RCP2.6)."

l.171: "the RCP8.5 scenario path" → "the high emission RCP8.5 scenario path"

l.175: maybe add a "\," in the unit "\unit{cmyear−1}"

l.176: "...warms by nearly $1 \pm 0.18\,°C$ (Figure 3c)." → Add "in the same period"

l.177: "... these increases become stronger." → "this warming trend/rate becomes stronger."

l.179: "current trends" → "currently observed trends"

l.189: "Areas of heavy precipitation under the reference climate (Figure 2b) also receive the highest increments."

l.195: "Also, the Amundsen Sea in front of Pine Island Glacier and Thwaites Glacier is cold. Here, the temperature might be too cold, which justifies the applied melting correction." → Which melt correction did you use? And "too cold" with respect to World Ocean Atlas?

l.205: "...do not necessarily grow in parallel." → Do you mean they are "not necessarily correlated"?

l.209: "ice-sheet" → "Ice-sheet"

l.214: The unit of Eq. 1 should read $\%\,K^{-1}$, hence $\Delta T$ should be in the denominator.

l.215: So P0 equals $P_{t=0}$ in Eq. 1?

l.217: Eq. 2 should have a number. And should $\Delta P$ be replaced by P0?

l.221: "...these locations. The difference is distinct for Vostok..." → "...ice core locations. The difference is most prominent for Vostok ice core..."

l.226: "Thus, we can safely restrict the analysis on the first 50 years." → of which simulation?

l.230: "Map 1" → "Map in Figure 1"

l.233: "... c-like area" → "...half-moon-shaped area" or just "... c-shaped area"

l.242:"We detect a slight trend to higher values if we restrict the analysis to ground ice." Maybe mention at this point that the difference results from excluded ice shelf regions, which are associated with x% of the total glacierized area and which are characterized by relatively shallow surface elevation along the ocean margin

l.243: "the difference between scenarios is more decisive" → "the impact of the choice of the scenarios is larger..."

l.245: "... Within their variability, many ensemble members are invariant against the applied scenario..." → "The sensitivity of many ensemble members to the range of applied scenario is within their variability..."

l.250: "... Antarctica's large-scale drainage basins." → Please provide a reference here, e.g. Zwally et al., 2015

l.251: "This division..." → "This chosen division..."

l.253: "...with a tendency of higher values..." → "...with a tendency towards higher values..."

l.261: "The region "Siple Coast" as a part of the "WAIS" region is different in many aspects. It has the smallest area.." → so it has a low weight in the spatial mean?! l.263: "...while the spread of trends among individual ensemble members is substantial." → Why not provide a number range at some points in the text?

l.267: "... trend in snow accumulation ..." → "... trend in observed snow accumulation..." to make sure that you switched from model results to observations in this paragraph

l.273: "... a unrealistic declining February sea ice trend" → "... an unrealistically declining February sea ice trend"

l.280: "which is also reflected by the maxima in these regions." → maxima in scaling factors?

l.281: "Also, the Ross Ice Shelf and the adjacent Siple Coast feature on average the lowest scaling factors across the entire ice sheet. Some individual ensemble members project even negative scaling: precipitation deficit for rising temperatures." Is this related to the Frieler et al., 2015 study or does this repeat the previous paragraph?

l.296: "The integrated precipitation shows a more pronounced temporal change, because the integral and not the mean precipitation is calculated, where the vast light precipitation regions lessen the average precipitation signal." Isn't the difference just a scaling factor, i.e. the considered area? I guess you are talking about a power-law distribution with a large weight of the continental areas with very low precipitation?

l.301: "... under the precipitation anomalies," → "... for applied precipitation anomalies,"

l.306: "if we would apply this low scaling of 2 % $K^{-1}$." Isn't this mentioned in the beginning of the sentence?

l.316: "Over the entire Antarctic continent, precipitation and temperature grow simultaneously in climate model simulations of the future." → To summarize, precipitation and temperature, as average over the entire Antarctic continent, grow simultaneously in climate model simulations of the future."

l.320: "the on Antarctica accumulated snowfall" → "the snowfall accumulated on Antarctica"

l.325: "... the implemented precipitation boundary condition..." → "... the applied

precipitation boundary condition..." or "... the choice of the precipitation boundary condition..."

l.327: "These together constitute the ensemble of ice-sheet simulations." → It would be nice to provide the size of the ensemble (3 sceanarios x 9 climate models x 2 reference periods x 2 precipitation forcing = 108 simulations?)

l.332: "...detected trend of about 2 mm decade−1 (sea-level equivalent) fades within the first 400 years..." → How can this trend be justified? Is the present-day reference forcing different from the one used in the spin-up? Or is this due to bed deformation? What figure shows this trend? It should be shown somewhere (Fig. 6?) as it amount to about 2cm after 100 model year and is substracted from the prejection results, right?

l.337: " than the simulations" → than in the simulations

l.338: Insert comma

l.340: "A ring of a pronounced negative thickness difference follows the coast, where the precipitation anomaly (Figure 2e, h, k) is enhanced." → "However, we find a negative thickness difference within a narrow band along the coast, where the precipitation anomalies (Figure 2e, h, k) suggest less accumulation that the scaling."

l.344 "... are negative" Please be more precise in this paragraph, what quantity is negative.

l.347: K-1 superscript

l.349: "... the ice thicknesses of the ensemble means.." → "... the mean ice thickness of each of the respective sub ensembles..."

l.354: "This reduction marks those outlet glaciers and ice shelves that are extremely vulnerable." Doesn't it say that ice losses under global warming are larger than gains?

l.359: "... ice-shelf weakening, ice thinning ..." → "... ice-shelf weakening, as well as ice thinning ..."

l.365: "...and restrict ourselves first to the model year 2100, where the precipitation anomalies of the period 1850-2100 shape the ice-sheet thickness distribution of the year 2100." → "the history of precipitation anomalies"

l.367: "Directly at margins apart from the vast ice shelves, the attributed model that drives either the maximum or minimum ice thickness shows a noisy small scale pattern, which is driven by the variety of the involved models (Figure 8d, e)." → I guess you want to say, that the maximum or minimum ice thickness in marginal regions cannot be associated with a particular climate model, while in contrast, for ice shelf regions...

l.372: "... while it also drives its thinning of the Ross Ice Shelf (Figure 8e) predominantly." → "... while it causes predominantly thinning within the main Ross Ice Shelf (Figure 8e)."

l.373: "Since the spatial pattern of the atmospheric and ocean forcing that promotes or undermines the ice thickness is not necessarily aligned, this may explain the small scale noisy pattern along the coast." → Maybe this explanation is not sufficient. The coastal regions is where most of the (nonlinear) dynamical changes on the considered time scales occur in response to both ocean and atmospheric forcing.

l.388: "NorESM1-M influences the WAIS, which is in accordance with the detected lowest scaling in the Siple Coast (Figure 5), CSIRO-Mk3-6-0 has an impact around the South Pole, MRI-CGCM3 has coastal zone in the EAIS, while the control of MPI-ESM-LR and, to a lesser extent, HadGEM2-ES spreads across the entire continent." → The reader may get lost here by the wording. Make sure that you are talking about the attribution of the minimum ice thickness to different climate models. You could also add percentages of the Antarctic area in the text to quantify the dominance. Similar issue for the maximum in l.398 ff.

l.393: "If we now turn towards the temperature scaled model simulations, the mean, maximum, and minimum ice thickness distribution..." → "If we now turn towards those model simulations, in which the temperature-scaled precipitation forcing has been ap-
plied, both the mean, maximum, and minimum ice thickness distribution..."

l.396: "The latter shows that the ocean controls ice-shelf thickness changes in our simulations primarily." → "The latter shows that primarily the ocean controls ice-shelf thickness changes in our simulations." or "The latter shows that the ocean primarily controls ice-shelf thickness changes in our simulations."

l.402: "precipitation driven" → "precipitation-driven"

l.409: Are you referring to all three scenarios here or just RCP8.5?

l.413: "...is quasi-constant until 2000 and declines afterward (Figure A15). For RCP8.5, the basal melting increases at the end of the 21st century quadratic." → "...remains quasi-constant until 2000 and declines afterwards (Figure A15). For RCP8.5, the basal melting increases at the end of the 21st century quadratically."

l.415: "... while the basal melting increases by approximately 33 % since the year 2000." → until 2100?

l.417: "The basal melting rates for PISM1Eq and PISM2Eq are similar, however, the loss rates for PISM1Eq are slightly larger than PISM2Eq (Figure A13)." → This means more basal melting for smaller ice shelf area? What is the portion of refreezing?

l.420: "Since floating ice shelves nourish both ice losses, these ice losses do not impact the sea-level directly." → "Although floating ice shelves are subject to both types of ice loss, these ice losses do not directly impact the sea-level."

l.423: " generates " → " would consequently generate "

l.425: "is not a 1:1 relation." → "is obviously not a 1:1 relation."

l.426: Shouldn't there be a time period involved, e.g. by 2100?

l.427: "It is less than integrated precipitation anomalies..." → "This is less than the integrated precipitation anomalies..., which explains the total mass gains."

l.429: "Anyhow, the integrated basal melting rates are too low and the calving rates are too high compared to observational estimates in our ensemble of ice-sheet model simulations." → What does too low and too high mean here, beyond observational uncertainty? Maybe quantify in terms of percent?

l.436: " loses mass " → " lost mass "

l.449: "the basal melting rated of grounded ice" → "the basal melt rate at he base of the grounded ice"

l.449: "Please note that this is not driven by any trend in the continued ice-sheet simulations under the reference climate (Table 2) since we have substracted this trend." → "Please note that there is no drift involved, as we substracted the trend from the continued ice-sheet simulations under the reference climate (Table 2)."

l.451: "We also detect an amplified signal for the simulations driven by the precipitation anomalies than scaled precipitation, which corresponds to the above diagnosed sea-level impact of the precipitation (Figure 6)." → Please reformulate!

l.453: Maybe add "net mass gain", which is associated with a negative sea-level contribution, but whether the global sea level falls is not only determined by Antarctica.

l.455: Please reformulate, such that the reader understands that you talk about a constant rate on the one hand and a linearly increasing integrated melt rate on the other hand.

l.456: "Ultimately, the more vibrant growth of the accumulation in comparison to the negligible increasing combined loss of iceberg calving and basal melting of ice shelves drive the falling sea level in our simulations after the year 2000 (Figure 12)." → "Also the combined loss of iceberg calving and basal melting of ice shelves does not vary much over the considered period. Consequently, the growth of the accumulation in our simulations explains the net mass gains and hence the negative sea-level contributions from Antarctica after the year 2000 (Figure 12)."

l.461: " temperature scaled precipitation " Add hyphen!

l.462: "As a consequence, these will contribute after the year 3200 (RCP8.5) and 3900 (RCP2.6) to a globally rising sea level on average in our simulations, which outruns the formerly fallen sea level since 1850." → "As a consequence, these simulations produce on average a positive contribution to the global sea level after the year 3200 (RCP8.5) and 3900 (RCP2.6), which compensates for the negative contributions since 1850."

l.470: "the deduced Antarctica's sea level contribution" → Please reformulate

l.471: "representing the observational-based ocean-driven basal melting." So you directly apply basal melt fluxes and no ocean-temperature based melt parameterization any more?

l.475: "Under the assumption that only a fraction of the adjusted basal mass contributes to the global sea level, we apply the simulated ratio of the sea level change to the total ice mass change." → The authors should better motivate that this conversion serves to express mass changes in terms of sea-level equivalents.

l.478: " sea level correction" Or do you mean "adjusted basal melt flux"?

l.480: Maybe omit "as its evolution, which considers the correction, highlights"

l.481: "..., we obtain too extensive corrections..." → "... we would obtained large corrections..." l.482: "This sea-level rise is larger" → "This corresponding sea-level rise would be larger"

l.485: "raises " → "could raise"

l.486: " do not impact the sea level." → " do not impact the sea level directly."

l.487: "ration" → "ratio"

l.490: " how the precipitation is implemented in ice-sheet simulations" → Better say: " how precipitation forcing is applied/estimated in ice-sheet simulations"

l.493: " In this case, numerical projections" → " In this case, our numerical projections"

l.498: "such as the ocean-ice-shelf-ice-sheet interactions." → "such as the interaction between ocean, ice shelves and ice sheet."

l.497: " thence"

l.506: " overwhelm " or better overcompensate

l.508: ", the total amount would be identical," → ", the average amount of precipitation change would be identical to the average precipitation anomaly,"

l.509: "proper" → "adequate" or "realistic"

l.510: "shall" → "should"

l.514: "... which have been identified across sixteen models" → You should add "within a recent model intercomparison exercise"

l.523: "This observed retreat and the related ice loss will continue in our simulations under RCP8.5." → "This observed retreat and the related ice loss will continue, most likely represented in our simulations by the scenario RCP8.5."

l.527: " further to the west" → relative to where?

l.531: Maybe put references after "lose ice", if they say so.

l.532: "according to our simulations." → "which is consistent in our simulations."

l.532: " will thin in the future." → reference or does the ensemble suggest so?

l.537: " reproduces appropriate " → " adequately reproduces "

l.548: "Even if we apply anomalies on top of the reference background fields, we can not exclude a shock-like behavior of the simulations entirely directly following the decades after the year 1850." → This is strange, could you quantify the variability around the 50-years mean?

l.853: "because the water masses of this range flow into the ice-sheet cavities and are in contact with large parts of ice shelve bases. " → "because the water masses at this depth potentially can flow into the ice-sheet cavities and reach large parts of ice shelves' bases. "

l.854: "Highest temperature increases occur in the Bellingshausen and Amundsen Seas..." → Is this an observation or does the climate models suggest so?

l.856: " flow already " → "already flow" as observations suggest?

l.857: "massive" → "largest"

l.865: " Temperature Scaling" → " Estimate of temperature scaling of precipitation from climate models"

l.868: " depend on if we determine " → " depend on the time period we chose as a reference "

l.871: "However, all these differences do not changes the spatial structure significantly, and they have a neglectable impact compared to the choice of the driving model." → "However, these differences do not significantly change the spatial structure. Their impact is negligible compared to the choice of the driving model."

l.877: "The detected precipitation deficit..." → Could you provide a definition here, is this negative scaling or just scaling below average?

l.880: "is small" → you could mention the relative size of the ice shelves, or you could account for ice shelves separately?

l.907: "while in both cases the thickness calving is active" → It would be very interesting if PISM could differentiate between the three calving styles in the reporting.

l.909: Make sure you the reader notices that you switched to observations.

l.919: "just termed basal melting rates" → why not "basal melt rates"

l.919: "the second ice mass loss process" → second largest process or does this just relate to the previous paragraph?

l.920: "The basal melting rate anomaly is computed relative to the 50 years between 1951 and 2000." Please indicate how this period compares to the observations of the World ocean atlas used as reference field?

l.921: " We could identify immediately that the basal melting rates have risen between 10 % and 100 % since the 1850s (Figure A13)" → "The inferred an increase in basal melt rates by 10-100% over the period 1850-x?"

l.922: "independent of the initial state selection" → "independent of the selection of the initial state" or simply "independent of the initial state"

l.922: " and reference to compute the" → " as well as to the reference period selected for the computation of the"

l.925: " subject to not negligible trend " → Please be more precises!

l.926: "In the future, the basal melting rate will further increase between 10 % and more than 100 %." → In future projections, the modeled basal melt rate further increases ... until the year x"

l.927: " specialized ocean simulations" → "high-resolution ocean simulations"

l.931: " is apparent." → " is clear/distinct."

l.937: " or reach a maximum of around 2100 and scenarios" → "and reach a maximum around the year 2100. Scenarios..." The maximum in basal melting in Fig. A13 and A14 seems to occur for all climate forcings a few decades before 2100, is there an explanation for this phenomenon?

l.939: "our approach works where the last 30 years of the forcing until 2100 is recurrently applied afterward." Please reformulate

l.942: " show a minimum of around 3500 " → " show a minimum around the year 3500 "

l.945: " ocean temperature anomalies are warmer" → " ocean temperature anomalies are larger" or "more pronounced"

l.948: " and an average decrement for RCP4.5 " → What does this mean?

l.954: "while the highest calving occurs under scenarios with a lower forcing." → This is surprising, do you have ideas for an explanation? Might this be related to the much smaller ice shelf area and hence shorter ice shelf front? The sentence in l.964 is not so clear on this assessment.

l.967: "Starting from original simulated ablation flux..." → Please start even earlier and explain briefly what the intention of this correction is. You take the modeled fluxes, modify them and apply them in additional sensitivity simulations? Is the reference flux usually obtained from observations? Maybe provide a figure to visualize the magnitudes.

l.980: Please provide some motivation here: "In order to provide an estimate of how ice shelf mass changes result in equivalent sea-level changes..."?

l.998: "the sea level rise of 30 cm is larger than the actual sea level rise " → "the corresponding sea level rise of 30 cm would be larger than the observed sea level rise" Please make sure in the wording that this is just a unit conversion and no dynamical estimate.

l.1000: "rise the" → "contributes to the"

l.1001: omit "(Equation A5)"

l.1002: "If" → "Whether"

l.1006: " losses" → "lose"

[Figure]

l.1009 and l.1010 and l.1012: "temperature scaled" → "temperature-scaled"

**Figures:**

Fig. 1: As this is the overview figure, the reader may expect the sector definitions of Table 4 visualized here, as done in Fig. 4.

Fig. 2: The color scale in panel a is somewhat counterintuitive with the coldest areas in red. Why not using a temperature colorscheme similar to panel c)?

Fig. 3: "is an extension into the sea" → maybe provide some estimate of the width. Also, the anomaly seems to be relative to the start period (at 1850), while for the ocean forcing, in l.920 in the Appendix a reference period 1950-2000 is indicated?

Fig. 4: Which are the dotted regions here? Sector outline seem to overlay each other.

Fig. 6: This figure is simply overloaded, I recommend to split somehow.

Fig. 11: You should mention that ice loss is the combination of calving and melt.

Fig. A1: You should mention in the caption that the 50 cm year$-1$ contour is larger than in previous figures.

Fig. A2: Where are the "white-grey lines" mentioned in the caption?

Fig. A3: Please increase the size of the climate model labels.

Fig. A4: It would help if the individual panels would use the same y-axis. Why is it important to distinguish between grounded ands glacierized here? Why not between grounded and floating?

Fig. A5: "where all additional mass loss rises immediately the sea level" → "assuming that all additional mass loss is converted into a sea-level equivalent"
Fig. A6: unit for y-axis is also m?

Fig. A7: Omit the in "of the each" in the caption.

Figs. A10+11: It would be nice to indicate that the difference is simply the eigencalving parameter and describe whether and where differences (in calving front location) occur.

Fig. A15: Could you state to what extent the trends can be attribute to grounding line retreat vs. calving front retreat?

**References:**

Frieler, Katja, Peter U. Clark, Feng He, Christo Buizert, Ronja Reese, Stefan RM Ligtenberg, Michiel R. Van Den Broeke, Ricarda Winkelmann, and Anders Levermann. "Consistent evidence of increasing Antarctic accumulation with warming." Nature Climate Change 5, no. 4 (2015): 348-352.

Jourdain, N. C., Asay-Davis, X., Hattermann, T., Straneo, F., Seroussi, H., Little, C. M., and Nowicki, S.: A protocol for calculating basal melt rates in the ISMIP6 Antarctic ice sheet projections, The Cryosphere Discuss., https://doi.org/10.5194/tc-2019-277, in review, 2019.

Levermann, A., Albrecht, T., Winkelmann, R., Martin, M. A., Haseloff, M., & Joughin, I. (2012). Kinematic first-order calving law implies potential for abrupt ice-shelf retreat. The Cryosphere, 6, 273-286.

Sutter, J., Fischer, H., Grosfeld, K., Karlsson, N. B., Kleiner, T., Van Liefferinge, B., & Eisen, O. (2019). Modelling the Antarctic Ice Sheet across the mid-Pleistocene transition–implications for Oldest Ice. The Cryosphere, 13(7), 2023-2041.

Zwally, H. Jay, Jun Li, John W. Robbins, Jack L. Saba, Donghui Yi, and Anita C. Brenner. "Mass gains of the Antarctic ice sheet exceed losses." Journal of Glaciology 61,

no. 230 (2015): 1019-1036.

---

## Referee Comment (RC2) · Anonymous Referee #2 · 23 Feb 2020

The authors used CMIP5 RCPs outputs for driving icesheet simulations to test how the precipitation boundary condition determines Antarctica's sea-level contribution. They found that the simulated ice-sheet thickness generally grows in a broad marginal strip where incoming storms deliver topographically governed precipitation. They further conducted scaling analysis showing that the scaling is higher across the East Antarctic Ice Sheet but lower across the West Antarctic Ice Sheet and lowest around the Siple Coast.

This study focuses on an interesting topic and potentially contributes to our understanding of further Antarctic icesheet change and sea level rise. Thereby, I would like

to support this manuscript be published in Earth System Dynamics after minor revisions.

First, the authors may want to notice the effect of evaporation and atmospheric moisture budget on Antarctic icesheet. Evaporation (E) is large and comparable with precipitation (P) over most of Antarctic during SON and DJF. In the atmospheric moisture budget over Antarctic, P-E is in generally balanced by horizontal convergence of vertically integrated moisture transport. Given the projected different responses of atmosphere circulation in various RCPs, it is would be nice to discuss the potential roles of atmospheric winds, moisture transports and in turn, P-E in Antarctic icesheet change.

Also, I am wondering how the results of authors' icesheeting simulations will affect Antarctic sea ice and deepwater formation. How will they modulate the Antarctic sea ice projection in various RCPs? How will they modulate deep convection in the marginal seas of the Antarctica, the formation of Antarctic Bottom Water and the strength of abyssal circulation?

---

## Author Response (AR1)

Torsten Albrecht (Referee)

**General comments:**

Rodehacke and colleagues investigate the effects of multiple climate model forcings (from CMIP5) in Antarctica. They assess the spatial heterogeneity in temperature and precipitation estimates over the period 1850-2100 for different emission scenarios and the spread among the 9 selected climate models. The ratio of precipitation anomalies and temperature anomalies is compared to paleo estimates at 6 ice core locations and regional variations from the spatial mean are discussed. Applied to the ice sheet model PISM they run an ensemble of simulations up to the year 5000 with both the anomaly forcing fields from the climate models and the simplified (spatial mean) parameterization (as often used in previous studies) and find quite some differences in the projected ice mass changes (converted in units of sea-level equivalents).

While in this study the temperature-scaled precipitation results in long-term ice losses, the directly applied precipitation anomalies generate net mass gains. Given the numerous previous projection studies, this is a surprising result. However, for given model settings this discrepancy can to some extent be explained in the manuscript. Overall, the study is well structured and the manuscript clearly-arranged. The main manuscript is separated into introduction, material and methods, results and discussions, conclusions and an appendix part. Due to its length of 42 pages including figures and references and 20 pages in the Appendix, it is sometimes difficult for the reader to follow the line of thought. The conclusions with almost 3 pages should be condensed, many discussed aspects could be merged into the introduction and discussion part. In general, the manuscript needs some additional work to improve the readability and to clarify the main key messages for the reader and avoid redundant informations. Also typos and the german-style syntax sometimes hampers the reader to fully grasp the content of the manuscript.

Figures have good quality and are informative, some are overloaded with up to 27 curves. Literature is sufficiently covered with 97 references. The investigation of the impact of climate boundary conditions on the future evolution of the Antarctic ice sheet supports the publication in ESD. However, as the main focus seems to be on the evaluation of climate model result and the systematic and comprehensive sensitivity analysis of the ice sheet model to the two different types of precipitation forcing, this study would also very well fit into a model-specific journal like GMD.

This study by Rodehacke et al. has the potential to be a valuable contribution to the scientific community of ice sheet modelers, as it considers relevant aspects of commonly used boundary conditions with potentially serious consequences for estimates of future sea-level change.

I encourage the authors to consider the following detailed suggestions and to improve the manuscript accordingly.

Thank you very much for reviewing our manuscript, your engagement, and your encouraging comments.

**Specific comments:**

1. The title should be refomulated. It is not easy to understand the content before having read the abstract. Also I wonder, if the more word "Ansatz" is commonly known in the wider scientific community apart from mathematicians. I would suggest: "Future sea-level contribution from the Antarctic Ice Sheet for different precipitation forcings based on CMIP5 models"

We are aware that the word Ansatz of German-origin is not widely used. But we prefer concise wording. Ansatz describes precisely the problem. The way how the precipitation is described/implemented in a mathematically and physical sense. In contrast, the suggested title is misleading because we do not use a different set of forcings, actually. However, we implement the available climate states differently to drive our ensemble of ice-sheet simulations.

2. As a surprising result the simulated Antarctic Ice Sheet gains mass under future global warming for directly applied climate model anomalies (temperature and precipitation). As this part has some delicate political implications it needs a clear discussion of the responsible model settings.

We are aware of the political implications of our results. Therefore, we have already discussed various influencing factors in the conclusion section, which you suggest to shorten. Hence, we do not see the need to expand this conclusion and the extensive discussion further because those parts would not be read by those who would like to misuse our results for their agenda. However, we extend both the "abstract" and "Plain Language Summary" sections to indicate the limitations of our study. In addition, we introduce the limitation of our forcing choices and its physical foundation.

In the abstract, we add "In contrast to various former studies, only the historical (1850--2005) and scenario (2006--2100) forcing drive our ensemble of simulations, which neglects unavoidable continuous warming consistent with the higher climate scenarios beyond the year 2100."

We append to the "Plain Language Summary" section: "Since we use only the available climate scenarios until the year 2100, any additional warming after 2100 may turn the ice gain into an ice loss under a strongly changing climate."

In addition, we add to the "Material and Methods" section, a paragraph discussing the above-indicated limitation: "The repetition of the last 30 years of climate forcing beyond the year 2100 is a simplification, which is not entirely consistent with the applied climate scenarios. An ongoing growing atmospheric greenhouse concentration triggers changes in the climate system. While the atmospheric radiation reacts immediately, the redistribution of the accompanied heating within the global ocean is much slower (Hansen et al., 2011). This delay is critical because the majority of the additional heat ends in the worldwide ocean (Church et al., 2011, 2013b). Consequently, further warming is inevitable after the cessation of greenhouse emissions (Hansen et al., 2005). Our simulations do not reflect this ongoing warming. Also, a disintegrating Greenland ice sheet will raise the global sea level, and, as a consequence of Greenland's reduced gravitational pull (Whitehouse, 2018), the sea level rise is in particular pronounced around Antarctica (Mitrovica et al., 2001). A rising sea level potentially migrates the grounding lines inshore, which ultimately destabilizes ice shelves and causes a more vulnerable Antarctic ice sheet. Despite, the same gravitational effect may buttress Antarctica, whether Antarctica's ice loss is slow enough (Gomez et al., 2010) and Greenland stabilizes. However, the ongoing thermal expansion of the ocean, which is currently the driver of the rising sea level (Rietbroek et al., 2016), will probably destabilize Antarctica. Therefore, our ensemble of ice sheet simulations is not a projection."

2a. Although the equilibrium state fits well observations of ice thickness and grounding line, the involved mass fluxes may not. Total ice loss rates by melting and in particular by calving are overestimated by a factor of 2-3 depending on the used eigencalving rate constant. Hence also the

surface mass balance seems to be overestimated accordingly. The authors imply that the uncertainties in the regional climate model results (RACMO), which are used as a present day reference field, are large enough to overestimate in particular the large slow-flowing and very dry inner-continental regions of the EAIS, where small absolute changes in precipitation can have large consequences for the total mass balance of the equilibrium state. Also, it is not clear from the description in the manuscript how the yearly cycle in the PDD scheme is estimated from the climate models (annual mean and summer temperatures) in order to obtain estimates of the surface mass balance components for given air temperature and precipitation forcing. A potential misfit in boundary conditions may be compensated for by a well-chosen set of model parameters, such that the equilibrium state bounds observational constraints. However, this potential overfitting of the initial equilibrium state may then have consequences for the projected ice mass changes, as the authors already speculate. In general, the equilibrium state method favors rather stable ice sheet configurations, which may not be realistic.

PDD: We have utilized the in PISM implemented PDD. During our analysis, we've compared the original surface mass balance (SMB) coming from the named RACMO data set with the SMB calculated by the PDD approach with and without considering a lapse rate of -7K km$^{-1}$. All these SMB are identical. Only if the surface elevation between the RACMO data set and the simulated elevation differ, the lapse rate shifts the air temperature, which introduces a difference in the SMB. The contemporary near-surface air temperature (we use the 2m-air temperature), except for the Antarctic Peninsula, is well below the freezing point temperature of meteoric freshwater (Figure A1a). Even the maximal warming of 8 Kelvin (Figure A1d), which occurs offshore, does not drive widespread surface melting generating ice mass loss. Therefore, the here used parametrized SMB calculated by the PDD approach is not critical. Please see, also, specific technical comment l.143 below, where we address the issue with the used PDD approach in more detail. In addition, we added a new appendix figure highlighting the surface mass balance; please see figure A16.

2b. The authors state one main difference to previous studies related to the forcing after the year 2100, which is commonly extrapolated into the future, while in this study it remains quasi constant (within 30 years variability). There seems to be another important difference in the methodology of this study in comparison to other studies. All climate model anomalies are inferred with respect to a preindustrial control simulation. However, the reference climate of the 19th century does not really match the modern climate, which the used mean background fields (from RACMO as mean over period 1979–2011 and from World Ocean Atlas as climatological mean) are related to. This precedure minimized the shock at the beginning of each simulation at 1850, but it adds some anomaly to the present-day background field when arriving at present-day in the simulations and it overestimates the future temperatures and precipitation rates applied to the Antarctic Ice Sheet. I encourage the authors to run some test simulations with shifted anomaly (negative anomaly in the preindustrial era and vanishing anomaly in present-day period).

It is right that we have not corrected the difference between the pre-industrial and the recent historical period (RACMO: 1979—2011, WOA: 1955—2006), since this difference is neglectable, e.g., 2m-air temperature (Figure 3b).

We would have liked to use instead of the RACMO atmospheric fields verified and validated distributions representing the pre-industrial state. However, these are not available (to our knowledge), except you consider spatially regridded shallow ice cores, which are subject to a distinct uncertainty, or fields coming from global climate models or could be deduced from these climate models. Since global climate models are not free of biases, we would have introduced an error, which is probably much larger than the mentioned offset. We have done experiments where CMIP5 model output has driven our ice-sheet model directly. We see that the simulations, which are driven by anomalies, are much less impacted by climate model biases.

Nevertheless, let us do a gedankenexperiment (Popper, 1935), where we would have considered this effect. A most probably slightly reduced surface mass balance would have been balanced by a slightly lower lateral ice loss via iceberg calving and basal melting to obtain a quasi-equilibrium state. This state would have been our initial state. Now we would run our ensemble from 1850 to the year 2100 and continue as we have done. The precipitation increase would be slightly less, the basal melting loss would also be slightly less, and the balance between these would be similar. We are confident about this conclusion because of our already existing ensemble with members showing a small increase. These members support this conclusion. With the current setup, which considers a slightly too strong forcing, we may promote enhanced basal melting (quadratic equation). However, the diagnostically deduced sea-level impact of the two ways to describe the sea-level dependence of the precipitation boundary condition (Figure 6b and 6d) is independent of the used reference state.

To conclude, the here used setup could be improved, but the conclusion is robust about the mentioned reference period of the background data set.

Popper, Karl. 1935. Logik Der Forschung. Edited by Philipp Frank and Moritz Schlick. Schriften zur Wissenschaftlichen Weltauffassung. Vienna: Springer Vienna. ISBN: 978-3-7091-2021-7.

2c. Regarding the basal melt parameterization, no details on sensitivity can be found in this study nor in the cited study by Sutter et al., 2019. What technique is used to extrapolate ocean temperatures into the ice shelf cavities? Are basin-wise overflow depths considered? Are extrapolated ocean temperatures vertically interpolated at the ice shelf base? Can refreezing occur? A simimar melt parameterization with quadratic dependency on thermal forcing has been calibrated in Jourdain et al., 2019 (https://doi.org/10.5194/tc-2019-277). They show that the choice of the particular parameterization and the associate parameters can have a huge impact on the ice sheet response. The authors discuss that the used melt parameterization may underestimate the melting and therefore apply bias-corrected melt rates as a sensitivity check, which does not cause considerable changes in the ice sheet response. The effect of basal melt could be also strongly intensified by using the melt interpolation across the grounding line (optional in PISM).

We have extended the description regarding the extrapolation (please see specific issue l.144), which also answers the question regarding the overflow depths. As part of the ocean forcing, we use the full 3D ocean temperature distribution so that the actual temperature at the ice shelf base drives the ocean melting, whereas our approach does not consider refreezing. In our understanding, ocean-driven basal melting does not occur beyond (inshore) the grounding line, because grounded ice stifles the flow of ocean water.

Since we use a sub-grid scale grounding line parameterization, we could have allowed partial basal melting proportional to the floating fraction. In pilot studies, the impact of the fractional basal melting has shown a minor effect on our simulations. Studies suggest that tides amplify frontal melting and locally also basal melting (Padman et al., 2018). In contrast, a tidal-driven front, which develops near the groundling line, hinders the flow of water which ultimately lowers the basal melting near the grounding line (Holland, 2008). First ocean observations in the groundline zone of the Ross Ice Shelf show low basal melting rates (Begemann et al., 2018). We decided to compute melting only for fully floating grid cells.

Begeman, Carolyn Branecky, Slawek M. Tulaczyk, Oliver J. Marsh, Jill A. Mikucki, Timothy P. Stanton, Timothy O. Hodson, Matthew R. Siegfried, Ross D. Powell, Knut Christianson, and Matt A. King. 2018. "Ocean Stratification and Low Melt Rates at the Ross Ice Shelf Grounding Zone." Journal of Geophysical Research: Oceans 123 (10): 7438–52. https://doi.org/10.1029/2018JC013987.

Holland, Paul R. 2008. "A Model of Tidally Dominated Ocean Processes near Ice Shelf Grounding Lines." Journal of Geophysical Research 113 (C11): C11002.

https://doi.org/10.1029/2007JC004576.

Padman, Laurie, Matthew R. Siegfried, and Helen A. Fricker. 2018. "Ocean Tide Influences on the Antarctic and Greenland Ice Sheets." Reviews of Geophysics 56 (1): 142–84. https://doi.org/10.1002/2016RG000546.

If we drive a newer PISM version (V1.1.4), where the ocean-driven basal melting is parameterized with the PICO submodel as part of PISM (Reese et al., 2018), with climate forcing from our climate model (AWI-CM, which is not part of the here use CMIP5 models), we also detect a growing Antarctic ice sheet. Therefore, we are confident that our parameter choices are not decisive and that our results are robust.

Reese, Ronja, Torsten Albrecht, Matthias Mengel, Xylar Asay-Davis, and Ricarda Winkelmann. 2018. "Antarctic Sub-Shelf Melt Rates via PICO." The Cryosphere 12 (6): 1969–85. https://doi.org/10.5194/tc-12-1969-2018.

3. The authors define different ways of expressing (integrated) mass changes in terms of sea-level equivalent changes and I would wish that it should be made clear when just a theoretical unit conversion is applied (potential sea level change) or when it is an actual sea-level contribution in terms of projected ice mass change. And if the latter it should be clearly defined whether only ice masses above flotation are considered. What diagnostic has been in fact used here?

Clarified by adding the following paragraph and applying related changes through the manuscript: "In this manuscript, we distinguish between potential sea level and simulated sea level. The potential sea level is the transformation of an ice mass or freshwater volume into a global sea level by applying a global ocean area of $3.61 \cdot 10^{14}\,m^2$ (Gill, 1982). In contrast, the simulated sea level is a diagnostic of the ice sheet model, which takes into account the total mass above flotation and the global ocean area. "

4. The manuscript states that the used coarse resolution of 16km may have consequences for the adequate representation of ice stream dynamics. I assume that this model choice is a consequence of the initial equilibriumn state, which requires hundred thousands model years to evolve. The authors state that basal resistance is described by a Mohr-Coulomb law with plastic till, but they do not discuss relevant parameters involved, such as the till friction angle or the till water decay rate. What is the vertical resolution of the enthalpy module? These parameters can strongly affect the ice stream dynamics also for coarse resolutions.

The till water decay rate amounts to 3.1687646154128e-11 meter seconds[-1]. We have not activated the hydrological model. The till fraction angle ranges from 10° for our bedrock at and below 700 m below the contemporary sea level at the start in the year 1950. With rising bedrock altitude, the friction angle increases linearly until 200 m above sea level and stabilizes at 30° (Figure I). Our mean vertical resolution is 67.9 m. Albrecht et al. (2020) show that a lower vertical resolution leads to a bigger ice sheet under transient forcing and, also, it promotes a more stable ice sheet.

Albrecht, Torsten, Ricarda Winkelmann, and Anders Levermann. 2020. "Glacial-Cycle Simulations of the Antarctic Ice Sheet with the Parallel Ice Sheet Model (PISM) -- Part 1: Boundary Conditions and Climatic Forcing." The Cryosphere 14 (2): 599–632. https://doi.org/10.5194/tc-14-599-2020.

[Figure]

*Figure I: Till fraction angle for till under grounded ice. The left (right) side shows the fraction angle for the initial states PISM1Eq (PISM2Eq). The red contour outlines the ice sheet or ice shelf outer edge. Under ice, the dash, and white contour line follow the -700m and the 200m, respectively, bedrock altitude in the year 1850.*

5. The authors use a rather old PISM version (v0.7), most likely for consistency reasons (initMIP and other model intercomparisons). However, PISM has evolved over the last years and some relevant aspects have been improved, which may affect the results of this study. For instance, the authors mention a bug in the elastic part of the LC solid Earth model, but also the viscous part was flawed and considered changes in ice shelf thickness as loads. Accordingly strong melt would cause uplift of the cavity bed and hence result in a stabilized grounding line. Also the till water distribution along the grounding line has been fixed meanwhile causing a much higher grounding line sensitivity. I guess also the sea-level potential diagnostic has been fixed meanwhile (now substracting the part below flotation). These are many good arguments in favor of a more recent PISM version and they suggest that Antarctic Ice Sheet simulations could respond with much higher sensitivity to the same forcing applied.

In our code basis, we fixed some code issues, such as the reproducibility and restart issues, which are also related to the bedrock code. In our simulations, we consider the viscous-part of the glacial isostatic adjustment (GIA) and do not use the elastic part. As explained below, an improved GIA model would maintain a more stable configuration of the grounding line, so that the discussed sea-level decline under a warming climate would be even more pronounced. Please see the special technical comment l.151. Please see also our reply to the technical issue of the appendix figure A15, where we state that we have a relatively stable grounding line in our simulations while the calving front retreat is more pronounced.

**Technical corrections:**

l.2: "heavier precipitation fallen on Antarctica will counteract any stronger iceberg discharge..."=>"precipitation will likely increase even more and may counteract stronger iceberg discharge..."

A warming climate accompanies increased precipitation due to the Clausius-Clapeyron relation. Hence we prefer: "Simulated future projections reveal that heavier precipitation, fallen on Antarctica, may counteract amplified iceberg discharge and increased basal melting of floating ice shelves driven by a warming ocean."

l.3: "from nine CMIP5 models future projections"=>"future projections from nine CMIP5 models"

We follow your request.

l.5: "The spatial and temporal varying climate forcings drive ice-sheet simulations. Hence, our ensemble inherits all spatial and temporal climate patterns, which is in contrast to a spatial mean forcing.:" => "The spatially and temporally varying climatic forcing drive the ice-sheet simulations, such that all climate patterns are represented in our ensemble, which is fundamentally different from using spatial means as forcing."

We follow your suggestion and use: "The spatially and temporally varying climatic forcing drives ice-sheet simulations, such that our ensemble represents all climate patterns, which is fundamentally different from using spatial means as forcing."

l.7: Regardless of the applied boundary condition and forcing, some areas will lose ice in the future, such as the glaciers from the West Antarctic Ice Sheet draining into the Amundsen Sea." => ..., our ensemble study suggests that some areas will lose ice in the future, …

Done.

l.10: "This strip also shows..." instead of using "... too."

Rewritten as requested.

l.25: "How strong the precipitation grows in a warming atmosphere, may be explained by the dissimilarity between the applied methods to describe the precipitation." => The discrepancy of the simulation results between the applied methods to describe the precipitation illustrates the uncertainty of the possible range of future precipitation growth in a warming atmosphere.

We follow your suggestion and use: "The discrepancy of the simulation results between both methods describing the precipitation illustrates the uncertainty of the possible range of future precipitation growth in a warming atmosphere."

l.30: "...impacts globally numerous economic activities..." => "...impacts numerous economic activities globally..."

Done.

l.31: "... or dedicated model simulations of, for instance, ice-sheet models." => "... or process-based model simulation, e.g. ice-sheet models."

Changed.

l.34: "... are simplified descriptions by analytical equations" => "... are either simplified descriptions based on linear multiple-regression analysis ... or"

Rephrased as requested.

l.36: "The simplified forcing, which usually does not show a dedicated spatial structure" => As surface elevation is a key variable in those parameterizations, the geometry of the ice sheet in fact leave some characteristic spatial structure.

We might have been misunderstood, hence we clarify: "The simplified temporal forcing, which usually does not show a dedicated spatial structure ..."

l.58: "temperature scaling" => "temperature scaling factor for precipitation" or "precipitation-temperature scaling "

Thanks, we use: "temperature scaling factor for precipitation"

l.71: Add comma before "probably"

Done.

l.124: Maybe omit "full" here.

It's like the term "Full Stokes", which is not entirely correct but everybody uses this term. We would like to keep it for clarity.

l.126: As in the title, I would recommend to use: "The type of precipitation forcing" or "the used method for applying precipitation forcing" instead of "the Ansatz of the precipitation".

Since a noun has to start with a lower case at the beginning, we correct it. As stated above, the word ansatz describes what we mean precisely.

l.130: "The latter is common, while some keep the surface mass balance constant." => "The latter approach is commonly used, in particular in paleo applications, while some sensitivity studies keep the surface mass balance constant."

We follow your request.

l.138: It could help the reader to have some definition of the piControl simulation here, e.g. "pre-industrial coupled atmosphere/ocean are performed at constant pre-industrial CO2 levels for x model years".

We cite literature and clarify the sentence introducing this paragraph: "Nine CMIP5 models deliver the following climate scenarios (see Table 1, Taylor et al. (2012)): control run under pre-industrial conditions (piControl), the historical period (1850-2005), as well as RCP2.6, RCP4.5, and RCP8.5 (2006-2100, Vuuren et al. (2011))."

l.140: "... differ commonly marginally." => "...show in general marginal differences."

Rephrased as suggested.

l.144: How does this extrapolation works? Is there a diffusion scheme applied for each vertical ocean temperature level? What are the source regions, the continental shelf or also the deeper ocean regions (this is not so clear from Fig. 3e), which are separated from the deeper cavity regions by the continental shelf? There is also no detailed description in Sutter et al., 2019, even though sub-shelf melting is a key process here.

We clarify: "... extrapolated horizontally into the ice shelves to mimic isopycnical flow: The operator 'fillmiss2' of the Climate Data Operators' (\href{https://code.mpimet.mpg.de/projects/cdo}{cdo}) tool kit acts on the original CMIP5 ocean grid."

l.146: "... following the positive degree day (PDD) approach, where the annual 2mair temperature standard deviation comes from daily CMIP5 model values." Does this mean that every year one different PDD standard deviation is applied to the whole computational setup or is it grid-cell wise?

In l.143 it is mentioned that "annual mean forcing" is ued, but what about the summer temperature anomaly to estimate the yearly cycle?

We use the PDD implementation of the PISM and perform the computation for each grid-cell; we clarify: "To allow for surface melting under a warming climate, the surface mass balance (SMB) is calculated following the positive degree day (PDD) approach (Braithwaite, 1995; Hock, 2005; Ohmura, 2001) as implemented in the PISM model (ThePISM Authors, 2015a, b). The turn of the hydrological year occurs on day 91 and the PDD factor for snow and ice are 0.3296 cm(IE) Kelvin$^{-1}$ day$^{-1}$ and 0.8792 cm(IE) Kelvin$^{-1}$ day$^{-1}$, respectively. Here, the evolving annual 2m-air temperature standard deviation is derived from daily CMIP5 model values for each ice sheet model grid-cell." Besides, at the end-of-the-21-century, summer temperatures are in general still too cold to drive widespread surface ablation.

l.148: "16 km" => What is the reason for this relatively coarse resolution, the availability of an equilibrium state?

We have run a much larger ensemble of several thousand members. Here we analyze only a subset of this full ensemble.

l.149: "utilizes" => "applies"

Changed to "employs"

l.151: Also the viscous part in v0.7 was somewhat unrealistic, as also ice shelf thickness change has been considered as loads in the LC bed deformation model, which has strong effects on grounding line sensitivity.

We are aware that the viscous part of the GIA model could better represent the impact of a changing ice shelf thickness. In our understanding, the updated implementation would maintain a more stable configuration of the grounding line, which would support firmer ice shelves and a more durable ice sheet. Ultimately, the here discussed sea-level decline under a warming climate would be even more pronounced.

l.154: "... pressure-dependent melting temperature" => Add "...of the ice"

Done.

l.157: "...while the grounding line position is determined on a sub-grid space (Feldmann et al., 2014)." => Add "... to interpolate basal friction."

Rephrased.

l.161: "...stress field divergence... " => "... divergence of the strain/velocity field" or "trace of the strain-rate field"

We accept the second suggestion, joyfully.

l.161: You should add units "m s" here as the Levermann et al. 2012 paper uses "m a".

Good point, we add as requested the units.

l.162: "(PISM1Eq and PISM2Eq)"=>This can be confusing, either you switch the order here or the order of the eigencalving constants in the sentence before.

We changed as suggested.

l.163: "Ocean temperatures from the World Ocean Atlas 2009 (Locarnini et al., 2010) and the multi-year mean surface mass balance (SMB) from the RACMO 2.3/ANT model (Van Wessem et al., 2014) drive PISM during spin-up (Table 2)." => Hence, this is a present-day forcing equilibrium.

This is correct. Please, also see our related reply to your specific comments 2b.

l.169: "... releasing less carbon dioxide." => Maybe add "(e.g. RCP2.6)."

We followed your suggestion.

l.171: "the RCP8.5 scenario path" => "the high emission RCP8.5 scenario path"

Done.

l.175: maybe add a "\," in the unit "\unit{cmyear−1}"

Good catch; done as suggested.

l.176: "...warms by nearly $1 \pm 0.18$ #C (Figure 3c)." => Add "in the same period"

Done.

l.177: "... these increases become stronger." => "this warming trend/rate becomes stronger."

We use: "this warming trend becomes stronger."

l.179: "current trends" => "currently observed trends"

Done.

l.189: "Areas of heavy precipitation under the reference climate (Figure 2b) also receive the highest increments."

Sorry, but it is unclear.

l.195: "Also, the Amundsen Sea in front of Pine Island Glacier and Thwaites Glacier is cold. Here, the temperature might be too cold, which justifies the applied melting correction." => Which melt correction did you use? And "too cold" with respect to World Ocean Atlas?

"Here, the climatological temperature distribution might be too cold because it does not replicate the confined flow of warm water masses through glacier-scoured troughs towards ice shelves. To overcome this limitation, we apply a spatially restricted melting correction. It increases the melting by 50% for the Ronne Ice Shelf region, and it quadruples melting for coastal parts of the West Antarctic Ice Sheet between the Antarctic Peninsula and the Getz Ice Shelf (east of the Ross Ice Shelf)."

l.205: "...do not necessarily grow in parallel." => Do you mean they are "not necessarily correlated"?

We use: "not necessarily correlate."

l.209: "ice-sheet" => "Ice-sheet"

Changed.

l.214: The unit of Eq. 1 should read % K−1, hence #T should be in the denominator.

Indeed! We corrected.

l.215: So P0 equals Pt=0 in Eq. 1?

We correct: "$P_{t=0}=0=P(t_{ref})$"

l.217: Eq. 2 should have a number. And should #P be replaced by P0?

We follow your advice and label this equation. In addition, we expanded the equation, so that it

becomes clear that it should be ΔP.

l.221: "...these locations. The difference is distinct for Vostok..." =>"...ice core locations. The difference is most prominent for Vostok ice core..."

We revise as requested.

l.226: "Thus, we can safely restrict the analysis on the first 50 years." => of which simulation?

To avoid ambiguity, we expand the sentence: "Thus, we can safely restrict the analysis and use as a reference for the computation of the anomalies the first 50 years of the piControl climate."

l.230: "Map 1" => "Map in Figure 1"

We rephrase as suggested.

l.233: "... c-like area" => "...half-moon-shaped area" or just "... c-shaped area"

We prefer to use the second suggestion: "c-shaped area"

l.242: "We detect a slight trend to higher values if we restrict the analysis to ground ice." Maybe mention at this point that the difference results from excluded ice shelf regions, which are associated with x% of the total glacierized area and which are characterized by relatively shallow surface elevation along the ocean margin

We extend to: "We detect a slight trend to higher values if we restrict the analysis to ground ice (87.5 % of the glaciated area, see Table 4); it excludes floating ice shelves with low elevation along the coasts."

l.243: "the difference between scenarios is more decisive" => "the impact of the choice of the scenarios is larger..."

Rephrased with "However, the scenario selection is decisive, while the choice between … "

l.245: "... Within their variability, many ensemble members are invariant against the applied scenario..." => "The sensitivity of many ensemble members to the range of applied scenario is within their variability..."

Taken.

l.250: "... Antarctica's large-scale drainage basins." => Please provide a reference here, e.g. Zwally et al., 2015

As suggested we added citations for different oceanographic zones and drainage basins.

l.251: "This division..." => "This chosen division..."

Done.

l.253: "...with a tendency of higher values..." => "...with a tendency towards higher values..."

Done.

l.261: "The region "Siple Coast" as a part of the "WAIS" region is different in many aspects. It has the smallest area.." =>so it has a low weight in the spatial mean?!

Each area has its own spatial mean regardless of its actual area size. We clarify: "The region 'Siple Coast' (area 0.69·10$^6$km$^2$, see Table 4) as a part of the 'WAIS' region (area 4.26·10$^6$km$^2$) is different in many aspects. It has the smallest area compared to the other regions (Table 4)"

l.263: "...while the spread of trends among individual ensemble members is substantial." => Why

not provide a number range at some points in the text?

We link to the requested information provided in one of our figures: " the spread of trends among individual ensemble members is substantial (Figure 5)."

l.267: "... trend in snow accumulation ..." =>"... trend in observed snow accumulation..." to make sure that you switched from model results to observations in this paragraph

Yes, we adjust as requested.

l.273: "... a unrealistic declining February sea ice trend" => "... an unrealistically declining February sea ice trend"

We follow your suggestion.

l.280: "which is also reflected by the maxima in these regions." => maxima in scaling factors?

Clarified by "the scaling factor maxima in these regions"

l.281: "Also, the Ross Ice Shelf and the adjacent Siple Coast feature on average the lowest scaling factors across the entire ice sheet. Some individual ensemble members project even negative scaling: precipitation deficit for rising temperatures." Is this related to the Frieler et al., 2015 study or does this repeat the previous paragraph?

Clarified: "As before, the Ross Ice Shelf and the adjacent Siple Coast feature, on average, the lowest scaling factors across the entire ice sheet (Figures 4 and 5). Some individual ensemble members project even negative scaling: precipitation deficit for rising temperatures (Figures 4 and 5)."

l.296: "The integrated precipitation shows a more pronounced temporal change, because the integral and not the mean precipitation is calculated, where the vast light precipitation regions lessen the average precipitation signal." Isn't the difference just a scaling factor, i.e. the considered area? I guess you are talking about a power-law distribution with a large weight of the continental areas with very low precipitation?

Clarified: "The integrated precipitation shows a more pronounced temporal change because the vast interior, characterized by light precipitation, governs the integral."

l.301: "... under the precipitation anomalies," =>"... for applied precipitation anomalies,"

Rephrased as requested.

l.306: "if we would apply this low scaling of 2 % K−1." Isn't this mentioned in the beginning of the sentence?

Indeed, hence it is discarded.

l.316: "Over the entire Antarctic continent, precipitation and temperature grow simultaneously in climate model simulations of the future." => To summarize, precipitation and temperature, as average over the entire Antarctic continent, grow simultaneously in climate model simulations of the future."

Done.

l.320: "the on Antarctica accumulated snowfall" => "the snowfall accumulated on Antarctica"

Done.

l.325: "... the implemented precipitation boundary condition..." => "... the applied precipitation boundary condition..." or "... the choice of the precipitation boundary condition..."

We follow your second suggestion.

l.327: "These together constitute the ensemble of ice-sheet simulations." =>It would be nice to provide the size of the ensemble (3 sceanarios x 9 climate models x 2 reference periods x 2 precipitation forcing = 108 simulations?)

We write " … ensemble of 208 ice-sheet simulations (Table 1)" and add additional information to the corresponding table caption "Since we do not use the RCP2.6 scenario of the CCSM4 model, the ensemble comprises 26 anomaly forcing scenarios. The climate anomalies are computed relative to the first or last 50 years of the corresponding piCtrl. Each scenario starts from the initial condition PISM1Eq (Figure A10) or PISM2Eq (Figure A11) and is driven by two precipitation conditions (see main text for details, e.g. section 3.2). Hence, the ensemble of anomaly ice sheet simulations has 208 members."

l.332: "...detected trend of about 2 mm decade−1 (sea-level equivalent) fades within the first 400 years..." => How can this trend be justified? Is the present-day reference forcing different from the one used in the spin-up? Or is this due to bed deformation? What figure shows this trend? It should be shown somewhere (Fig. 6?) as it amount to about 2cm after 100 model year and is substracted from the prejection results, right?

We take the explanation from the discussion to here and address in addition the remark l548.

l.337: " than the simulations" => than in the simulations

Done.

l.338: Insert comma

Done.

l.340: "A ring of a pronounced negative thickness difference follows the coast, where the precipitation anomaly (Figure 2e, h, k) is enhanced." => "However, we find a negative thickness difference within a narrow band along the coast, where the precipitation anomalies (Figure 2e, h, k) suggest less accumulation that the scaling."

We do not come to this conclusion about the scaling because the ocean also influences the ice thickness along the coast. So we keep the descriptive character.

l.344 "... are negative" Please be more precise in this paragraph, what quantity is negative.

Rephrased: "Furthermore, as part of the WAIS these values are present in the coastal strip from the Antarctic Peninsula to the Ross Ice Shelf and along the eastern flank of the Transantarctic Mountain Range (Figure 4)."

l.347: K-1 superscript

Good catch; done.

l.349: "... the ice thicknesses of the ensemble means.." => "... the mean ice thickness of each of the respective sub ensembles..."

Rephrased as requested.

l.354: "This reduction marks those outlet glaciers and ice shelves that are extremely vulnerable." Doesn't it say that ice losses under global warming are larger than gains?

Regardless of the applied RCP8.5 climate forcing coming from our pool of climate models (Table 1), these outlet glaciers and ice shelves lose mass, where the corresponding ice thickness is negative. Consequently, the loss outweighs the gain.

l.359: "... ice-shelf weakening, ice thinning ..." => "... ice-shelf weakening, as well as ice thinning ..."

Rephrased as suggest and reordered: "ice-shelf thinning, as well as ice-shelf weakening, "

l.365: "...and restrict ourselves first to the model year 2100, where the precipitation anomalies of the period 1850-2100 shape the ice-sheet thickness distribution of the year 2100." => "the history of precipitation anomalies"

Rephrased: "and restrict ourselves first to the model year 2100, when the transient forcing of period 1850--2100 excites changing ice thicknesses."

l.367: "Directly at margins apart from the vast ice shelves, the attributed model that drives either the maximum or minimum ice thickness shows a noisy small scale pattern, which is driven by the variety of the involved models (Figure 8d, e)." => I guess you want to say, that the maximum or minimum ice thickness in marginal regions cannot be associated with a particular climate model, while in contrast, for ice shelf regions...

Thanks, we modified as requested.

l.372: "... while it also drives its thinning of the Ross Ice Shelf (Figure 8e) predominantly." => "... while it causes predominantly thinning within the main Ross Ice Shelf (Figure 8e)."

Done.

l.373: "Since the spatial pattern of the atmospheric and ocean forcing that promotes or undermines the ice thickness is not necessarily aligned, this may explain the small scale noisy pattern along the coast." => Maybe this explanation is not sufficient. The coastal regions is where most of the (nonlinear) dynamical changes on the considered time scales occur in response to both ocean and atmospheric forcing.

Various studies show commonly a linear behavior, but we add this though, joyfully.

l.388: "NorESM1-M influences the WAIS, which is in accordance with the detected lowest scaling in the Siple Coast (Figure 5), CSIRO-Mk3-6-0 has an impact around the South Pole, MRI-CGCM3 has coastal zone in the EAIS, while the control of MPI-ESM-LR and, to a lesser extent, HadGEM2-ES spreads across the entire continent." => The reader may get lost here by the wording. Make sure that you are talking about the attribution of the minimum ice thickness to different climate models. You could also add percentages of the Antarctic area in the text to quantify the dominance. Similar issue for the maximum in l.398 ff.

Adding more information, such as the suggested percentages, would inflate the text and its complexity. Also, these values are most probably specific to the here used selection of climate models and do not represent a particular physical process. Therefore, we would like to drop this idea. Nevertheless, we simplify the sentence structure and break the sentences into pieces to help the reader: "NorESM1-M influences the WAIS, which is supported by its lowest scaling in the Siple Coast region (Figure 5). CSIRO-Mk3-6-0 has an impact around the South Pole, MRI-CGCM3 affects the coastal zone in the EAIS. The control of MPI-ESM-LR and, to a lesser extent, HadGEM2-ES spreads across the entire continent."

l.393: "If we now turn towards the temperature scaled model simulations, the mean, maximum, and minimum ice thickness distribution..." => "If we now turn towards those model simulations, in which the temperature-scaled precipitation forcing has been applied, both the mean, maximum, and minimum ice thickness distribution..."

Changed as requested.

l.396: "The latter shows that the ocean controls ice-shelf thickness changes in our simulations

primarily." => "The latter shows that primarily the ocean controls ice-shelf thickness changes in our simulations." or "The latter shows that the ocean primarily controls ice-shelf thickness changes in our simulations."

Changed as suggested.

l.402: "precipitation driven" => "precipitation-driven"

Done.

l.409: Are you referring to all three scenarios here or just RCP8.5?

Clarified: "For all climate scenarios, ..."

l.413: "...is quasi-constant until 2000 and declines afterward (Figure A15). For RCP8.5, the basal melting increases at the end of the 21st century quadratic." => "...remains quasi-constant until 2000 and declines afterwards (Figure A15). For RCP8.5, the basal melting increases at the end of the 21st century quadratically."

Done.

l.415: "... while the basal melting increases by approximately 33 % since the year 2000." => until 2100?

Rephrased: "between the years 2000 and 2100."

l.417: "The basal melting rates for PISM1Eq and PISM2Eq are similar, however, the loss rates for PISM1Eq are slightly larger than PISM2Eq (Figure A13)." => This means more basal melting for smaller ice shelf area? What is the portion of refreezing?

It is correct that smaller ice shelves are subject to more basal melting, while refreezing does not occur.

l.420: "Since floating ice shelves nourish both ice losses, these ice losses do not impact the sea-level directly." => "Although floating ice shelves are subject to both types of ice loss, these ice losses do not directly impact the sea-level.

We follow your suggestion partly and write: "Since floating ice shelves nourish both ice losses, these ice losses do not directly impact the sea-level."

l.423: " generates " => " would consequently generate "

Done.

l.425: "is not a 1:1 relation." => "is obviously not a 1:1 relation."

Done.

l.426: Shouldn't there be a time period involved, e.g. by 2100?

To avoid any ambiguity, we added as requested, even if we haven't changed the discussed period (1850-2100).

l.427: "It is less than integrated precipitation anomalies..." => "This is less than the integrated precipitation anomalies..., which explains the total mass gains."

Added.

l.429: "Anyhow, the integrated basal melting rates are too low and the calving rates are too high compared to observational estimates in our ensemble of ice-sheet model simulations." => What does too low and too high mean here, beyond observational uncertainty? Maybe quantify in terms

of percent?

We link to the corresponding figures, because in all figures depicting the total mass loss (Figure 11), calving rates (Figure A12), and basal melting rates (Figure A13), independent estimates are provided by symbols. In addition, the reported uncertainties of these estimates are provided, if they are larger than the corresponding symbol sizes, as stated in each corresponding figure caption.

l.436: " loses mass " => " lost mass "

Rephrased to "have lost mass"

l.449: "the basal melting rated of grounded ice" => "the basal melt rate at he base of the grounded ice"

I'm sorry, but to my knowledge, basal melting occurs at the base. Since we would like to avoid this pleonasm, we keep the original sentence.

l.449: "Please note that this is not driven by any trend in the continued ice-sheet simulations under the reference climate (Table 2) since we have substracted this trend." => "Please note that there is no drift involved, as we subtracted the trend from the continued ice-sheet simulations under the reference climate (Table 2)."

Thanks, changed.

l.451: "We also detect an amplified signal for the simulations driven by the precipitation anomalies than scaled precipitation, which corresponds to the above diagnosed sea level impact of the precipitation (Figure 6)." => Please reformulate!

Reformulated: "We also detect an amplified signal for the simulations driven by the precipitation anomalies compared to those forced by temperature-scaled precipitation anomalies, which corresponds to the above diagnosed sea-level impact of the precipitation (Figure 6)"

l.453: Maybe add "net mass gain", which is associated with a negative sea-level contribution, but whether the global sea level falls is not only determined by Antarctica.

Rephrased: "... , gain mass causing a falling sea level"

l.455: Please reformulate, such that the reader understands that you talk about a constant rate on the one hand and a linearly increasing integrated melt rate on the other hand.

Rewritten: "The basal melting of grounded ice does not impact the sea-level evolution, because this basal melting rate is nearly constant so that the corresponding integrated sea-level equivalent grows linearly for all scenarios from 1850 until 2100, and only after the year 2500 these curves diverge."

l.456: "Ultimately, the more vibrant growth of the accumulation in comparison to the negligible increasing combined loss of iceberg calving and basal melting of ice shelves drive the falling sea level in our simulations after the year 2000 (Figure 12)." => "Also the combined loss of iceberg calving and basal melting of ice shelves does not vary much over the considered period. Consequently, the growth of the accumulation in our simulations explains the net mass gains and hence the negative sea-level contributions from Antarctica after the year 2000 (Figure 12)."

We follow your advice and use "Also, the combined loss of iceberg calving and basal melting of floating ice shelves does not vary considerably over the considered period. Consequently, the growth of simulated accumulation explains the net mass gains and, hence, the negative sea-level contributions from Antarctica after the year 2000 (Figure 12)."

l.461: " temperature scaled precipitation " Add hyphen!

Thanks for indicating. We followed your suggestion and adjusted the manuscript accordingly.

l.462: "As a consequence, these will contribute after the year 3200 (RCP8.5) and 3900 (RCP2.6) to a globally rising sea level on average in our simulations, which outruns the formerly fallen sea level since 1850." =>"As a consequence, these simulations produce on average a positive contribution to the global sea level after the year 3200 (RCP8.5) and 3900 (RCP2.6), which compensates for the negative contributions since 1850."

Done.

l.470: "the deduced Antarctica's sea level contribution" => Please reformulate

Reformulated: "simulated sea-level contribution of Antarctica"

l.471: "representing the observational-based ocean-driven basal melting." So you directly apply basal melt fluxes and no ocean-temperature based melt parameterization any more?

We still use the growth of the simulated basal melting rates covering the period from 1850 to 5000, but the time series are adjusted to reproduce current estimates of basal melting rates. Unfortunately, these observational estimates present only the contemporary period. Please see the appendix for further information. We replace "representing" by "emulating" so that we obtain: "a corrected time series emulating the observational-based ocean-driven basal melting."

l.475: "Under the assumption that only a fraction of the adjusted basal mass contributes to the global sea level, we apply the simulated ratio of the sea level change to the total ice mass change." => The authors should better motivate that this conversion serves to express mass changes in terms of sea-level equivalents.

The introduction of this section has been modified and offers a clear motivation: "This analysis shall reveal if a more vibrant basal melting rate in concert with the simulated ice sheet mass evolution leads to a less pronounced ice sheet growth or drives even ice loss. Ultimately, does a more vigorous melting of ice shelves raise the simulated sea level of all ensemble members?"

l.478: " sea level correction" Or do you mean "adjusted basal melt flux"?

We clarify: "By adjusting the basal melting flux, the determined temporal evolution of the sea level correction"

l.480: Maybe omit "as its evolution, which considers the correction, highlights"

Good point, we follow your suggestion.

l.481: "..., we obtain too extensive corrections..." => "... we would obtain large corrections..."

Done.

l.482: "This sea-level rise is larger" => "This corresponding sea-level rise would be larger"

Changed.

l.485: "raises " => "could raise"

We rephrase: "would raise"

l.486: " do not impact the sea level." => " do not impact the sea level directly."

Done.

l.487: "ration" => "ratio"

Fixed.

l.490: " how the precipitation is implemented in ice-sheet simulations" => Better say: "how

precipitation forcing is applied/estimated in ice-sheet simulations"

Rephrased: "specified"

l.493: " In this case, numerical projections" => " In this case, our numerical projections"

Extended as suggested.

l.498: "such as the ocean-ice-shelf-ice-sheet interactions." => "such as the interaction between ocean, ice shelves, and ice sheet."

Done.

l.497: " thence"

Rarely use adverb with the meaning of 'therefrom.'

l.506: " overwhelm " or better overcompensate

Exchanged "overwhelm" by "overcome."

l.508: ", the total amount would be identical," => ", the average amount of precipitation change would be identical to the average precipitation anomaly,"

We prefer: "the integrated precipitation would be identical"

l.509: "proper" => "adequate" or "realistic"

Rephrased as suggested: "realistic"

l.510: "shall" => "should"

We use US English, where, to our knowledge, the auxiliary (modal) verb 'shall' is used in formal writing and expresses determination. It's different in British English. Hence, we would like to keep it.

l.514: "... which have been identified across sixteen models" => You should add "within a recent model intercomparison exercise"

Done.

l.523: "This observed retreat and the related ice loss will continue in our simulations under RCP8.5." => "This observed retreat and the related ice loss will continue, most likely represented in our simulations by the scenario RCP8.5."

Reordered: "In our simulations under the RCP8.5 scenario, this observed retreat and the related ice loss will continue."

l.527: " further to the west" => relative to where?

Further to the west of the discussed area (Wilkens Basin in the hinterland of George V Land) as part of the EAIS.

l.531: Maybe put references after "lose ice", if they say so.

Indeed, references should be at the end.

l.532: "according to our simulations." => "which is consistent in our simulations."
l.532: " will thin in the future." => reference or does the ensemble suggest so?

Rephrased and clarified: "According to the ensemble projecting the future, for them, continuous ice loss is inevitable. It also shows that the Ferringo Ice Stream flowing into the Bellingshausen Sea

will thin in the future.”

l.537: “ reproduces appropriate ” => “ adequately reproduces ”

Changed by “reasonably reproduces”

l.548: “Even if we apply anomalies on top of the reference background fields, we can not exclude a shock-like behavior of the simulations entirely directly following the decades after the year 1850.“ => This is strange, could you quantify the variability around the 50-years mean?

It is not strange because, at the start of the simulations, the climatic anomaly fields of the first years are not necessarily identical to the 50-years averages. The first few years may be warmer or colder than the mean. Please inspect the early decades of the ensemble mean's forcing (Figure 3) to obtain an impression of the climate variability.
We combine this point with the issue l332 above, where most of the paragraph is located now.

l.853: “because the water masses of this range flow into the ice-sheet cavities and are in contact with large parts of ice shelve bases. “ => “because the water masses at this depth potentially can flow into the ice-sheet cavities and reach large parts of ice shelves’ bases. “

I disagree with “potentially” because it is actually in contact with the ice shelf base either after direct inflow (for example in the Amundsen Sea) or after modification (for example in the Filchner-Ronne Ice Shelf); see for example Thompson et al. (2018). To avoid any ambiguity, we write: “because these water masses flow into the ice-sheet cavities and are in contact with the ice shelve bases.”

l.854: “Highest temperature increases occur in the Bellingshausen and Amundsen Seas...” => Is this an observation or does the climate models suggest so?

Clarified: “Highest temperature increases occur in the Bellingshausen and Amundsen Seas as part of the West Antarctic Ice Sheet (WAIS) and some spots along the East Antarctic Ice Sheet (EAIS) according to observations (Schmidtko et al., 2014; Jacobs, 2006).”

l.856: “ flow already ” => “already flow” as observations suggest?

We write: “In the Bellingshausen and the Amundsen Sea, warm water masses flow into ice-shelf cavities as indicated by observations (Arneborg et al., 2012; Thompson et al., 2018) and model simulations (Nakayama et al., 2018).”

l.857: “massive” => “largest”

Exchanged “tremendous” for “massive.”

l.865: “ Temperature Scaling” => “ Estimate of temperature scaling of precipitation from climate models”

We use instead: “Temperature Scaling of Precipitation derived from Climate Models”

l.868: “ depend on if we determine ” => “ depend on the time period we chose as a reference ”

Replace: “depend on if we determine the anomalies” by “depend on the time period we chose as a reference”.

l.871: “However, all these differences do not changes the spatial structure significantly, and they have a neglectable impact compared to the choice of the driving model.” => “However, these differences do not significantly change the spatial structure. Their impact is negligible compared to the choice of the driving model.”

Done as suggested.

l.877: "The detected precipitation deficit..." => Could you provide a definition here, is this negative scaling or just scaling below average?

Clarified: "the detected precipitation deficit (shrinking precipitation rates), captured by reanalysis data … ."

l.880: "is small" => you could mention the relative size of the ice shelves, or you could account for ice shelves separately?

Yes, we could, but the manuscript is already pretty long. Therefore we will not further investigate this point.

l.907: "while in both cases the thickness calving is active"=>It would be very interesting if PISM could differentiate between the three calving styles in the reporting.

I agree it would be nice, but I have not kept the data that allows differentiating the individual contributions due to storage space limitations.

l.909: Make sure you the reader notices that you switched to observations.

Clarified: "According to observational estimates control iceberg calving and basal ice-shelf melting the overall mass loss of Antarctica, while the relative contribution is the subject of current research."

l.919: "just termed basal melting rates" => why not "basal melt rates"
l.919: "the second ice mass loss process" => second largest process or does this just relate to the previous paragraph?

We rephrase: "The basal melting rate of floating ice shelves (hereinafter basal melting rates) is the second ocean-driven ice mass loss process beside iceberg calving."

l.920: "The basal melting rate anomaly is computed relative to the 50 years between 1951 and 2000." Please indicate how this period compares to the observations of the World ocean atlas used as reference field?

The WOA2009 says: "For the present atlas we attempted to reduce the effects of irregular space-time sampling by the averaging of five 'climatologies' computed for the following time periods: 1955-1964, 1965-1974, 1975-1984, 1985- 1994, and 1995-2006. The first-guess field for each of these climatologies is the 'all- data' monthly mean objectively analyzed temperature data." (Locarnini, et al., 2010; page 6).

Locarnini, R. A., A. V. Mishonov, T. P. Antonov, T.P. Boyer, and H.E. Garcia. 2010. "World Ocean Atlas 2009, Volume 1: Temperature." Edited by S Levitus. Vol. 1. U.S. Government Printing Office, Washington, D.C. https://www.nodc.noaa.gov/OC5/WOA09/pr_woa09.html.

l.921: " We could identify immediately that the basal melting rates have risen between 10 % and 100 % since the 1850s (Figure A13)" => "The inferred an increase in basal melt rates by 10-100% over the period 1850-x?"

Rephrased to: "In general, the basal melt rate increases by 10 %--100 % over the period 1850-2100 (Figure A13)."

l.922: "independent of the initial state selection" => "independent of the selection of the initial state" or simply "independent of the initial state"

Done.

l.922: " and reference to compute the" => " as well as to the reference period selected for the computation of the"

Done.

l.925: " subject to not negligible trend " => Please be more precises!

We quantified the trend: "For instance, the average of the global absolute 2m-air temperature difference between the first and last 50 years of piControl amounts 0.17 K (median 0.12 K) for all CMIP5 models considered in our study. In contrast, MIROC-ESM's value is 0.67 K."

l.926: "In the future, the basal melting rate will further increase between 10 % and more than 100 %." => In future projections, the modeled basal melt rate further increases … until the year x"

To avoid a pleonasm, we write: "In future projections, the basal melting rate increases between 10% and more than 100% until the year 2100"

l.927: " specialized ocean simulations" => "high-resolution ocean simulations"

Rephrased to "dedicated ocean simulations."

l.931: " is apparent." => " is clear/distinct."

Changed to "is self-evident."

l.937: " or reach a maximum of around 2100 and scenarios" => "and reach a maximum around the year 2100. Scenarios..." The maximum in basal melting in Fig. A13 and A14 seems to occur for all climate forcings a few decades before 2100, is there an explanation for this phenomenon?

The applied running mean of 5 years and the constructed forcing after the year 2100 causes the visual shift of the maximum. We use the forcing until the year 2100 and repeat afterward recurrently the last 30 years of the forcing (2071-2100). For instance, the depicted forcing in the year 2100 corresponds to the weighted sum of the forcing of the years 2098, 2099, 2100, 2101(=2071), and 2102(=2072). We have clarified it by reordering the sentences in the figure captions of Figure A12/A13 and adding, in addition, these two sentences: "After the year 2100, the forcing of the last thirty years until 2100 drives the model recurrently." and "The applied running means shift the apparent maximum backward in time so that it occurs visually before the year 2100.". The last sentence is added to figure caption A14 too.

l.939: "our approach works where the last 30 years of the forcing until 2100 is recurrently applied afterward." Please reformulate

We reformulate: "Since the temporal variability remains high also after 2100, our approach works to construct the forcing beyond the year 2100 (see section 2: "Material and Methods")."

l.942: " show a minimum of around 3500 " => " show a minimum around the year 3500"

We rephrase as suggested.

l.945: " ocean temperature anomalies are warmer" => " ocean temperature anomalies are larger" or "more pronounced"

We write now: "This result reflects the dependence of the basal melting on the ocean temperature because a warmer climate scenario induces higher ocean temperature anomalies."

l.948: " and an average decrement for RCP4.5 " => What does this mean?

Rephrased: "and an intermediate decrement of RCP4.5"

l.954: "while the highest calving occurs under scenarios with a lower forcing." => This is surprising, do you have ideas for an explanation? Might this be related to the much smaller ice shelf area and hence shorter ice shelf front? The sentence in l.964 is not so clear on this assessment.

Why is this surprising? However, we sharpen the last sentence of this paragraph: "The total area of ice shelves is, in general, smaller when a warmer climate scenario impacts these ice shelves (Figure A15) and the degraded total ice shelf area downgrades the calving probability. Ultimately, the integrated calving rate is lower under a warmer climate."

l.967: "Starting from original simulated ablation flux..." => Please start even earlier and explain briefly what the intention of this correction is. You take the modeled fluxes, modify them and apply them in additional sensitivity simulations? Is the reference flux usually obtained from observations? Maybe provide a figure to visualize the magnitudes.

We write "Since the simulated ocean-driven basal melting rates are lower than observational-based estimates (Figure A13), the impact of flux corrected basal melting rates on the model results are discussed in the main text (Section 3.7.1 "Sea level contribution of corrected basal melting" on page 15). This section describes the method."

l.980: Please provide some motivation here: "In order to provide an estimate of how ice shelf mass changes result in equivalent sea-level changes..."?

We write now: "To relate the sea-level change to the ice mass evolution, we define the ratio $p(t)$ of the sea level temporal deviation to the ice mass temporal deviation as $p(t) = \dots$ ."

l.998: "the sea level rise of 30 cm is larger than the actual sea level rise " => "the corresponding sea level rise of 30 cm would be larger than the observed sea level rise" Please make sure in the wording that this is just a unit conversion and no dynamical estimate.

Changed as requested.

l.1000: "rise the" => "contributes to the"

Done.

l.1001: omit "(Equation A5)"

Done.

l.1002: "If" => "Whether"

Done.

l.1006: " losses" => "lose"

Corrected.

l.1009 and l.1010 and l.1012: "temperature scaled" => "temperature-scaled"

Thanks for indicating; we've adjusted the entire text accordingly.

**Figures:**

Fig. 1: As this is the overview figure, the reader may expect the sector definitions of Table 4 visualized here, as done in Fig. 4.

As requested, we added the boundaries of the defined regions and adjusted the figure caption accordingly.

Fig. 2: The color scale in panel a is somewhat counterintuitive with the coldest areas in red. Why not using a temperature colorscheme similar to panel c)?

We prefer different properties of different color schemes to avoid confusion. In addition, we use color schemes that are aware of different types of color blindnesses.

Fig. 3: "is an extension into the sea" => maybe provide some estimate of the width. Also, the anomaly seems to be relative to the start period (at 1850), while for the ocean forcing, in l.920 in the Appendix a reference period 1950-2000 is indicated?

We extend the caption:"… is an extension into the sea (typical width of about 500 km)."
It is indeed correct that we talk about a different reference period (1951-2000) in line 920 as part of the appendix section "A2 Marginal ice loss by ocean-driven basal melting and iceberg calving." There, we compare our growth rates of the basal melting rates with independent studies cited a few lines below (line 926-929). These studies determine the growth rates until the end of the century relative to 1951-2000.

Fig. 4: Which are the dotted regions here? Sector outline seem to overlay each other.

Solved. Indeed, in the submitted version, the dotted regions are absent while it is available in our version. Apparently, the size reduction of the submitted file has striped the dotted pattern. We apologize for being sloppy.

Fig. 6: This figure is simply overloaded, I recommend to split somehow.

We have worked with different versions of this plot because we initially shared your impression. However, a printed article allows taking the time needed to see the differences in this information-rich plot. In addition, splitting the figure may hinder the direct comparison of different model forcing scenarios. Therefore, we would like to keep the current figure.

Fig. 11: You should mention that ice loss is the combination of calving and melt.

The extended figure caption begins with: "Temporal evolution of the ocean-driven ice loss rates of the fringing ice shelves around Antarctica for the period from 1850 to 2100. The ice loss comprises iceberg discharge and basal melting of ice shelves. … "

Fig. A1: You should mention in the caption that the 50 cm year−1 contour is larger than in previous figures.

I assume you refer to Figure 2 when you talk about the previous figure. The extended figure caption ends with "Figure 2 shows the corresponding ensemble mean fields, where the white contour line in the precipitation field corresponds to 30 cm year-1."

Fig. A2: Where are the "white-grey lines" mentioned in the caption?

We replaced "white-grey" with "light-grey"

Fig. A3: Please increase the size of the climate model labels.

The entire figure has a new label layout.

Fig. A4: It would help if the individual panels would use the same y-axis. Why is it important to distinguish between grounded ands glacierized here? Why not between grounded and floating?

This appendix figure has deliberately individual y-axis for each panel. Otherwise, the bunch of lines could not be separated in each panel. We are aware that this is a busy plot, but we are confident that the interested reader welcomes the provided rich set of information.

In simplified terms, the melting of grounded ice contributes to the global sea level, while the mere disintegration of floating ice does not. Hence the diagnosed sea-level contribution of grounded ice is vital.

To get an idea of how much snow accumulates on all glacierized ice, regardless if it is floating or grounded, we present these. Since we analyze precipitation anomalies, the change of precipitation rates may be of interest to the following communities. Those who have an interest in the evolution of the surface mass balance, use remote satellite products to analyze precipitation or water vapor transport changes in the Southern Ocean, and those running automated weather stations in coastal areas or floating ice shelves, for instance.

Fig. A5: "where all additional mass loss rises immediately the sea level" => "assuming that all additional mass loss is converted into a sea-level equivalent"

We change this part to "assuming that all additional mass loss rises the global sea level"

Fig. A6: unit for y-axis is also m?

Yes, we have indicated that this Figure is like Figure A5 and the y-axis says "Sea level contribution (m)." However, since your question expresses a misunderstanding, we adjust the figure caption and add explicitly the unit "(in meter)": "... The sea level (in meter) is computed relative to the year 2000. ..."

Fig. A7: Omit the in "of the each" in the caption.

We assume you suggest to erase "of the each ensemble member." We follow your suggestion.

Figs. A10+11: It would be nice to indicate that the difference is simply the eigencalving parameter and describe whether and where differences (in calving front location) occur.

We have added for each state the total ice volume and ice area (grounded ice and floating ice) in an additional table as part of the appendix. We link from the figure captions of both Figures (A10 and A12) to this table. This table should allow the reader to distinguish both states. A copy of this table is listed below:

| Field | PISM1Eq | PISM2Eq | Ratio: PISM1Eq/PIMS2Eq |
|---|---|---|---|
| Area: grounded ice (km2) | 1.255e7 | 1.257e7 | 0.9985 |
| Area: floating ice (km2) | 2.005e6 | 1.569e6 | 1.278 |
| Volume: grounded ice (km3) | 2.588e7 | 2.605e7 | 0.9936 |
| Volume: grounded ice above z=0 (km3) | 2.313e7 | 2.325e7 | 0.9947 |
| Ratio: all grounded ice / grounded ice above z=0 (%) | 89.35 | 89.25 | 1.001 |
| Volume: floating ice (km3) | 6.681e5 | 5.421e5 | 1.232 |

Fig. A15: Could you state to what extent the trends can be attribute to grounding line retreat vs. calving front retreat?

If the grounding line retreated while the calving front stayed fixed, the ice shelf area would increase. Since the grounding does not advance on a large scale, a calving front that retreats faster

than the grounding line is required to obtain a shrinking ice shelf area. Therefore the shrinking ice shelf area is dominated by a retreating calving front. We hope this answers your question.

(This page is left empty intentionally.)

Anonymous Referee #2

The authors used CMIP5 RCPs outputs for driving icesheet simulations to test how the precipitation boundary condition determines Antarctica's sea-level contribution. They found that the simulated ice-sheet thickness generally grows in a broad marginal strip where incoming storms deliver topographically governed precipitation. They further conducted scaling analysis showing that the scaling is higher across the East Antarctic Ice Sheet but lower across the West Antarctic Ice Sheet and lowest around the Siple Coast.

This study focuses on an interesting topic and potentially contributes to our understanding of further Antarctic icesheet change and sea level rise. Thereby, I would like to support this manuscript be published in Earth System Dynamics after minor revisions.

Thank you very much for reviewing our manuscript and your encouraging comments.

First, the authors may want to notice the effect of evaporation and atmospheric moisture budget on Antarctic icesheet. Evaporation (E) is large and comparable with precipitation (P) over most of Antarctic during SON and DJF. In the atmospheric moisture budget over Antarctic, P-E is generally balanced by horizontal convergence of vertically integrated moisture transport. Given the projected different responses of atmosphere circulation in various RCPs, it is would be nice to discuss the potential roles of atmospheric winds, moisture transports and in turn, P-E in Antarctic icesheet change.

Thanks for indicating these very intriguing points.

We would have liked to compute the surface mass balance with a more physical based surface mass balance scheme that takes into account the balance between radiative and turbulent fluxes as well as the conductivity of heat within the snowpack besides phase changes between liquid water and solid ice. Our model at hand would have been able to determine also the impact of sublimation, which balances, for example in the Dry Valley accumulation (Bliss et al., 2011). However, the data required are not available for all here used CMIP5 models. Hence, we have used a parameterization to compute the surface mass balance. Here, we decided to utilize the widely accepted and used positive degree day (PDD) approach (Hock, 2003), which is justified by the high correlation between the main drivers of ablation (radiation) and the near-surface air temperature (Ohmura, 2001).

Bliss, Andrew K., Kurt M. Cuffey, and Jeffrey L. Kavanaugh. 2011. "Sublimation and Surface Energy Budget of Taylor Glacier, Antarctica." Journal of Glaciology 57 (204): 684–96. https://doi.org/10.3189/002214311797409767.

Hock, Regine. 2003. "Temperature Index Melt Modelling in Mountain Areas." Journal of Hydrology 282 (1–4): 104–15. https://doi.org/10.1016/S0022-1694(03)00257-9.

Ohmura, Atsumu. 2001. "Physical Basis for the Temperature-Based Melt-Index Method." Journal of Applied Meteorology 40 (4): 753–61. https://doi.org/10.1175/1520-0450(2001)040<0753:PBFTTB>2.0.CO;2.

Regionally, the surface mass balance is influenced by sublimation/evaporation in Antarctica. The strength of this process differs by a factor of two between model studies (Agosta et al., 2019; Wessem et al., 2018). The sublimation is strongly correlated with the surface temperature and only significant during summer (Lenaerts et al., 2012). This effect is already included in our background fields to which we add the anomalies of the 2m-air temperature and precipitation. We add the figure A16 to highlight the quality of the here used approach computing the surface mass balance.

In the RACMO model, the snow sublimation includes a wind-driven process, which dominates the sublimation (Wessem et al., 2018). Over the Antarctic continent, surface sublimation and blowing

snow sublimation lose mass on the order of 29 mm yr$^{-1}$ and dispose 17–20% of the total annual precipitation over this region (Déry and Yau, 2002). However, the large-scale effect of surface blowing snow redistribution is negligible (Déry and Yau, 2002). We are confident that the differences between the used CMIP5 models are larger than the described effects and dominate the results. Further analysis of the changes in moisture transport is beyond the scope of this study and would extend this already lengthy manuscript.

Agosta, Cécile, Charles Amory, Christoph Kittel, Anais Orsi, Vincent Favier, Hubert Gallée, Michiel R. van den Broeke, et al. 2019. "Estimation of the Antarctic Surface Mass Balance Using the Regional Climate Model MAR (1979–2015) and Identification of Dominant Processes." The Cryosphere 13 (1): 281–96. https://doi.org/10.5194/tc-13-281-2019.

Déry, Stephen J., and M.K. Yau. 2002. "Large-Scale Mass Balance Effects of Blowing Snow and Surface Sublimation." Journal of Geophysical Research 107 (D23, 4679): 17pp. https://doi.org/10.1029/2001JD001251.

Lenaerts, J.T.M., M.R. van den Broeke, W.J. van de Berg, E. van Meijgaard, and P. Kuipers Munneke. 2012. "A New, High-Resolution Surface Mass Balance Map of Antarctica (1979–2010) Based on Regional Atmospheric Climate Modeling." Geophysical Research Letters 39 (L04501): 5pp. https://doi.org/10.1029/2011GL050713.

Wessem, Jan Melchior van, Willem Jan Van De Berg, Brice P.Y. Noël, Erik Van Meijgaard, Charles Amory, Gerit Birnbaum, Constantijn L. Jakobs, et al. 2018. "Modelling the Climate and Surface Mass Balance of Polar Ice Sheets Using RACMO2 - Part 2: Antarctica (1979-2016)." The Cryosphere 12 (4): 1479–98. https://doi.org/10.5194/tc-12-1479-2018.

Also, I am wondering how the results of authors' ice sheeting simulations will affect Antarctic sea ice and deepwater formation. How will they modulate the Antarctic sea ice projection in various RCPs? How will they modulate deep convection in the marginal seas of the Antarctica, the formation of Antarctic Bottom Water and the strength of abyssal circulation?

Here we could only speculate since we do not simulate the actual processes in the ocean. Since our manuscript is already long, we prefer to keep this short and do not discuss these important points. In particular, the other referee suggested shortening instead of expanding our manuscript.

**Precipitation Ansatz dependent Future Sea Level Contribution by Antarctica based on CMIP5 Model Forcing.**

Christian B. Rodehacke[1,2], Madlene Pfeiffer[1], Tido Semmler[1], Özgür Gurses[1], and Thomas Kleiner[1]

[1]Alfred Wegener Institute Helmholtz Centre for Polar and Marine Research, D-27570 Bremerhaven, Germany
[2]Danish Meteorological Institute, DK-2100 Copenhagen Ø, Denmark

**Correspondence:** Christian Rodehacke (christian.rodehacke@awi.de)

**Abstract.** Various observational estimates indicate growing mass loss at Antarctica's margins but also heavier precipitation across the continent. Simulated future projections reveal that heavier precipitation, fallen on Antarctica, may counteract amplified iceberg discharge and increased basal melting of floating ice shelves driven by a warming ocean. Here, we use models future projections from nine CMIP5, ranging from strong mitigation efforts to business-as-usual, to run an ensemble of ice-sheet simulations. In contrast to various former studies, only the historical (1850–2005) and scenario (2006–2100) forcing drive our ensemble of simulations, which neglects unavoidable continuous warming consistent with the higher climate scenarios beyond the year 2100. We test how the precipitation boundary condition determines Antarctica's sea-level contribution. The spatially and temporally varying climatic forcing drives ice-sheet simulations, such that our ensemble represents all climate patterns, which is fundamentally different from using spatial means as forcing. Regardless of the applied boundary and forcing conditions, our ensemble study suggests that some areas will lose ice in the future, such as the glaciers from the West Antarctic Ice Sheet draining into the Amundsen Sea. In general, the simulated ice-sheet thickness grows in a broad marginal strip, where incoming storms deliver topographically controlled precipitation. This strip also shows the largest ice thickness differences between the applied precipitation boundary conditions. On average Antarctica's ice mass shrinks for all future scenarios if the precipitation is scaled by the spatial temperature anomalies coming from the CMIP5 models. In this approach, we use the relative precipitation increment per degree warming as invariant scaling constant. In contrast, Antarctica gains mass in our simulations if we apply the simulated precipitation anomalies of the CMIP5 models directly. Here, the scaling factors show a distinct spatial pattern across Antarctica. Furthermore, the diagnosed mean scaling across all considered climate forcings is larger than the values deduced from ice cores. In general, the scaling is higher across the East Antarctic Ice Sheet, lower across the West Antarctic Ice Sheet, and lowest around the Siple Coast. The latter is located on the east side of the Ross Ice Shelf.

**Plain Language Summary** In the warmer future, the ice sheet of Antarctica will lose more ice at the margin, because more icebergs may calve and the warming ocean melts more floating ice shelves from below. However, the hydrological cycle is also stronger in a warmer world. As a consequence, more snowfall precipitates on Antarctica, which may balance the amplified marginal ice loss. In this study, we have used future climate scenarios from various global climate models to perform numerous ice-sheet simulations. These simulations represent the Antarctic Ice Sheet. We analyze whether Antarctica will grow or shrink. In all our simulations, we find that certain areas will lose ice under all circumstances. However, depending on the method used to describe the precipitation reaching Antarctica in our simulations, parts of the Antarctic Ice Sheet may either grow or

shrink in the future. The discrepancy of the simulation results between both methods describing the precipitation illustrates the uncertainty of the possible range of future precipitation growth in a warming atmosphere. Furthermore, the dissimilarity is pronounced differently between the West Antarctic and East Antarctic Ice Sheet. Since we use only the available climate scenarios until the year 2100, any additional warming after 2100 may turn the ice gain into an ice loss under a strongly changing climate.

**1 Introduction**

Sea level rise as a symptom of the progressing climate warming is of foremost importance for coastal societies because it impacts numerous economic activities globally and threatens the population along coasts. Antarctica's contribution to the future sea level is projected by either statistical approaches that take advantage of the deduced past behavior or process-based model simulations, e.g. ice-sheet models (Church et al., 2013a). To run ice-sheet models, commonly simplified temporal forcing anomalies for the entire continent are applied on top of spatial background fields (e.g. Golledge et al., 2015; Winkelmann et al., 2012; Pollard and DeConto, 2009). Those background or reference fields are either descriptions based on linear multiple-regression analysis (e.g., surface elevation and latitude dependence (Fortuin and Oerlemans, 1990)) or come from regional climate models or climatological data sets. The simplified temporal forcing, which usually does not show a dedicated spatial structure, follows some ad hoc assumed temporal evolution or is constructed from a set of CMIP5 model simulations, for instance. Here, we use various climate scenarios of an ensemble of CMIP5 models to drive the Parallel Ice Sheet Model (PISM, e.g. Bueler and Brown, 2009; Winkelmann et al., 2011), where we exploit the full temporal and spatial pattern in the atmospheric and oceanographic forcing. This approach is an enhancement to previous studies utilizing CMIP5 ensembles to infer only the temporal evolution of the future forcing (Golledge et al., 2015; Winkelmann et al., 2012).

The coupling between atmospheric warming and the enhanced hydrological cycle is often described as a Clausius-Clapeyron process, where the saturation pressure of water vapor scales exponentially by about $7\%$ per Kelvin warming (Held and Soden, 2006) — it is implicitly assumed that the relative humidity does not change. Climate modeling studies representing the Last Glacial Maximum (LGM), the pre-industrial (piControl) and historical period as well as climate warming scenarios (1pctCO2, abrupt4xCO2) show that the global precipitation increases in warmer climates and decreases in colder climates with a rate of $1\%\mathrm{K}^{-1}$ to $4\%\mathrm{K}^{-1}$ (Held and Soden, 2006; Li et al., 2013). Globally, this rate, which is the mean precipitation scaling, is less than the suggested thermodynamically justified Clausius-Clapeyron process for various reasons. Decreasing precipitation rates 
[revised manuscript text omitted]
models, which cause extreme changes. Before we conclude, we estimate differences in Antarctica's sea-level contribution for
the variety of applied forcing and precipitation boundary conditions.

**2 Material and Methods**

The full temporally and spatially varying forcings are obtained from an ensemble of CMIP5 models representing a suite of climate scenarios. These climate forcings drive the Parallel Ice Sheet Model (PISM) in order to estimate Antarctica's future sea-level contribution. In particular, the ansatz of the precipitation determines whether the global sea level rises or falls. We consider two precipitation boundary conditions. (1) On top of the reference background distributions (see Table 2), which drive the ice-sheet model during the spin-up, we utilize both the temperature and the precipitation anomalies from CMIP5 models. (2) We take only the temperature anomalies from CMIP5 models and compute the precipitation anomalies scaled by the temperature anomalies. The latter approach is commonly used, in particular in paleo applications (Applegate et al., 2012; Bakker et al., 2016; de Boer et al., 2013, e.g), while some sensitivity studies keep the surface mass balance constant (Feldmann and Levermann, 2015; Hughes et al., 2017). In some cases, negative temperature scaling is considered unrealistic (Frieler et al., 2012).

Nine CMIP5 models deliver the following climate scenarios (see Table 1, Taylor et al. (2012)): control run under pre-industrial conditions (piControl), the historical period (1850-2005), as well as RCP2.6, RCP4.5, and RCP8.5 (2006-2100, Vuuren et al. (2011)). These models stem from different model families (Knutti et al., 2013) and cover the range of current atmospheric (Agosta et al., 2015) and oceanographic (Sallée et al., 2013a) model uncertainties although model deficiencies such as insufficient resolution can exist across all models. The transient forcing from 1850 until 2100 comprises the historical and scenario periods. Afterward, the last 30 years (2071-2100) are repeated until the year 5000 to keep the natural variability. From the control run of the climate model (piControl), we use either the first or the last 50 years. By this procedure, we could quickly identify a drift in CMIP5 models and assess its impact. Additionally, the number of scenarios is twice as large, since the mean states of the first and last 50 years show in general marginal differences. Anomaly forcing is computed relative to either the first or last 50 years of the control run. In the following, the first 50 years act generally as reference.

The repetition of the last 30 years of climate forcing beyond the year 2100 is a simplification, which is not entirely consistent with the applied climate scenarios. An ongoing growing atmospheric greenhouse concentration triggers changes in the climate system. While the atmospheric radiation reacts immediately, the redistribution of the accompanied heating within the global ocean is much slower (Hansen et al., 2011). This delay is critical because most of the additional heat ends in the worldwide ocean (Church et al., 2011, 2013b). Consequently, further warming is inevitable after the cessation of greenhouse emissions (Hansen et al., 2005). Our simulations do not reflect this ongoing warming. Also, a disintegrating Greenland ice sheet will raise the global sea level, and, as a consequence of Greenland's reduced gravitational pull (Whitehouse, 2018), the sea level rise is in particular pronounced around Antarctica (Mitrovica et al., 2001). A rising sea level potentially migrates the grounding lines inshore, which ultimately destabilizes ice shelves and causes a more vulnerable Antarctic ice sheet. Despite, the same gravitational effect may buttress Antarctica, whether Antarctica's ice loss is slow enough (Gomez et al., 2010) and Greenland stabilizes. However, the ongoing thermal expansion of the ocean, which is currently the driver of the rising sea level (Rietbroek et al., 2016), will probably destabilize Antarctica. Therefore, our ensemble of ice sheet simulations is not a projection.

Atmospheric and oceanic forcing is applied as annual mean forcing on top of the forcing used to spin-up the ice-sheet model (Table 2). Since CMIP5 models do not resolve ice shelves, ocean temperatures are extrapolated horizontally into the ice shelves to mimic isopycnical flow: The operator "fillmiss2" of the Climate Data Operators' (cdo) tool kit acts on the original CMIP5 ocean grid. To allow for surface melting under a warming climate, the surface mass balance (SMB) is calculated following the positive degree day (PDD) approach (Braithwaite, 1995; Hock, 2005; Ohmura, 2001) as implemented in the PISM model (The PISM Authors, 2015a, b). The turn of the hydrological year occurs on day 91 and the PDD factor for snow and ice are $0.3296\,\mathrm{cm(IE)}\,\mathrm{Kelvin}^{-1}\,\mathrm{day}^{-1}$ and $0.8792\,\mathrm{cm(IE)}\,\mathrm{Kelvin}^{-1}\,\mathrm{day}^{-1}$, respectively. Across Antarctica, the surface mass balance computed via the PDD approach (Figure A16b) is identical to the one used during the spin-up (Figure A16a). Here, the evolving annual 2m-air temperature standard deviation is derived from daily CMIP5 model values for each ice sheet model grid-cell.

The ice-sheet model PISM — based on version 0.7 — runs on a 16 km equidistant polar stereographic grid and it utilizes a hybrid system combining the Shallow Ice Approximation (SIA) and Shallow Shelf Approximation (SSA). The model employs a generalized version of the viscoelastic Lingle-Clark bedrock deformation model (Bueler et al., 2007; Lingle and Clark, 1985). In our simulations, only the viscous part has been used because of known implementation flaws in the elastic part in our and later PISM versions. The basal resistance is described as plastic till by a Mohr-Coulomb formula to perform the yield stress computation (Bueler and Brown, 2009; Schoof, 2006). The basal melting of ice shelves is proportional to the squared thermal temperature forcing ($\Delta T_{\mathrm{force}}^2$), which is the difference between the pressure-dependent melting temperature of the ice and the actual ocean temperature above melting. Here, the parameterization considers the full depth-dependence of the ocean temperature field, as described in Sutter et al. (2019). Basal ice-shelf melting occurs only in fully floating grid points, while the grounding line position is determined on a sub-grid space (Feldmann et al., 2014) to interpolate basal friction.

The calving occurs at the ice-shelf margin, and three sub-schemes determine it. (1) At the ocean-ice-shelf margin, ice-shelf grid points with a thickness of less than $150\,\mathrm{m}$ calve. (2) Ice shelves calve that extend across the continental shelf edge and progress into the depth ocean (defined by the $1500\,\mathrm{m}$ depth contour). (3) The Eigen-calving parameterization exploits the divergence of the strain/velocity field (Levermann et al., 2012), with the proportionality constant of either $1 \cdot 10^{18}\mathrm{m\,s}$ or $1 \cdot 10^{17}\mathrm{m\,s}$. Two independent spin-up runs delivering our initial conditions (PISM1Eq and PISM2Eq) utilize these constants. Ocean temperatures from the World Ocean Atlas 2009 (Locarnini et al., 2010) and the multi-year mean surface mass balance (SMB) from the RACMO 2.3/ANT model (Van Wessem et al., 2014) drive PISM during spin-up (Table 2). A similar model setup has taken part in the initMIP-Antarctica exercise under the name AWI_PISM1Eq with an adjusted Eigen-calving proportionality constant of $2 \cdot 10^{18}$ and no bed deformation (Seroussi et al., 2019a).

**3   Results and Discussions**

Depending on the applied CMIP5 forcing scenario, the ensemble mean climate signal is weaker for those scenarios following an aggressive mitigation path and, hence, releasing less carbon dioxide (e.g. RCP2.6). Around Antarctica, the here analyzed

ensemble follows the same pattern (Figure 2 and Figure 3). Since in the past decade greenhouse gas concentrations have followed most closely the high-emission RCP8.5 scenario path, we will focus on RCP8.5 if not otherwise stated.

**3.1 Ensemble Forcing**

From 1850 until the end of the 21st century, the ensemble mean 2m-air temperature in Antarctica (see the map of Figure 3d) rises steadily by $6\,K$ with a spread of $1\,K$ (one standard deviation) (Figure 3a) while the mean precipitation accumulates $9 \pm 3\,\mathrm{cm\,year}^{-1}$ (water equivalent) in addition (Figure 3b). The average potential ocean temperature in the depth range of $150\mathrm{m}$ to $500\mathrm{m}$ depth along Antarctica's coast (see the map of Figure 3e) warms by nearly $1 \pm 0.18°\mathrm{C}$ in the same period (Figure 3c). In particular, since the beginning of 21st century, these warming trend becomes stronger.

[revised manuscript text omitted]

$$P(t,\boldsymbol{x}) = \Delta P(t,\boldsymbol{x}) + P_{t=0}(\boldsymbol{x}) = \Delta P(t,\boldsymbol{x}) \left[1 + \Delta T(t,\boldsymbol{x}) \cdot S(t,\boldsymbol{x})\right]. \tag{2}$$

The scaling deduced from ice cores varies in Antarctica between $5\,\%\mathrm{K}^{-1}$ and $7\,\%\mathrm{K}^{-1}$, with a 2-sigma uncertainty of about $1\,\%\mathrm{K}^{-1} - 3\,\%\mathrm{K}^{-1}$ (Figure 4, Table 3).

The corresponding scaling of the ensemble mean is generally larger at these ice core locations. The difference most prominent for the Vostok ice core and, to a less degree, also for EDML and EDC, while, within the uncertainties, the scaling of the Law Dome, Talos Dome, and WAIS ice cores are indistinguishable from the corresponding ensemble means. Here, we have computed the scaling by averaging the precipitation of the piControl run (first 50 years) to obtain the reference data and the last 50 years of the RCP8.5 scenario from 2051 until 2100 to get the anomalies. If we replace the reference period by the first 50 years of the historical period (1850-1899), the results are similar, and the values change only slightly. Thus, we can safely restrict the analysis and use as a reference for the computation of the anomalies the first 50 years of the piControl climate.

The spatial distribution of the scaling derived from our ensemble data is spatially heterogeneous and varies stronger than the ice core data suggest. Values in the range between $4\,\%\mathrm{K}^{-1}$ and $6\,\%\mathrm{K}^{-1}$ occur at the Filchner-Ronne Ice Shelf and in the coastal Terre Adélie region (see Map in Figure 1 for place names). Furthermore, as part of the WAIS these values are present in the coastal strip from the Antarctic Peninsula to the Ross Ice Shelf and along the eastern flank of the Transantarctic Mountain Range (Figure 4).

The highest scaling factor emerges on the EAIS, where a c-shaped area as part of the high plateau has factors exceeding $12\,\%\mathrm{K}^{-1}$. This area reaches out to the Dronning Maud Land with very high scaling factors too. The West Antarctic Ice Sheet has scaling factors generally lower than $8\,\%\mathrm{K}^{-1}$ and only on the elevated interior values up to $10\,\%\mathrm{K}^{-1}$ are detected. Over the Ross Ice Shelf and the eastward adjacent Siple Coast, scaling factors are the lowest (Figure 4). Since we detect raised scaling factors at a higher elevation, we aimed at determining whether we could find a relationship between elevation and scaling. However, neither for the entire Antarctic continent nor for defined subregions (see below), we could identify any robust relationship (not shown).

Our analysis focuses now on the scaling factors of all grounded ice, which, if lost, contributes to a rising potential sea level by Antarctica. Additionally, we analyze the scaling factors for the entire continent (label "glaciered"), and four glaciated regions labeled "EAIS Atl", "EAIS Ind", "WAIS", and "Siple Coast" (Figure 5 and Table 4). We detect a slight trend to higher values if we restrict the analysis to ground ice ($87.5\,\%$ of the glaciated area, see Table 4); it excludes floating ice shelves with low elevation along the coasts. However, the scenario selection is decisive, while the choice between "glaciered" and "grounded" is unessential for the ensemble mean as well as for numerous individual ensemble members (e.g., CCSM4, CanESM2, HadGEM2-ES, NorESM1-M). The sensitivity of many ensemble members to the range of applied scenario is within their variability (e.g., CSIRO-Mk3-6-0, CNRM-CM5, MIROC-ESM, MRI-CGCM3) or may hint at an enlarged scaling for weaker scenarios (e.g., MPI-ESM-LR). Frieler et al. (2015) found a low dependence of the scaling factors to four RCP scenarios for the whole Antarctic continent. Anomalies are not as distinctly pronounced in RCP2.6 as in the other scenarios due to the weaker forcing scenario. Please note, that CCSM4 is missing in the RCP2.6 (hence we have hatched the corresponding bar).

The boundaries of the three regions "EAIS Atl", "EAIS Ind", and "WAIS" resemble different oceanographic zones (Whitworth III et al., 2013; Orsi et al., 1999; Foldvik and Gammelsrød, 1988) under the consideration of Antarctica's large-scale drainage basins (Zwally et al., 2015). This chosen division of Antarctica does not produce surface area of equal size. As already indicated by the spatial distribution (Figure 4), the ordering from high to low scaling factors would be "EAIS Atl", "EAIS Ind", and "WAIS". The difference between both "EAIS" regions is minor, with a tendency towards higher values in "EAIS Atl" in the ensemble mean and some individual ensemble members. Some ensemble members do not show a clear trend between the scenario strength and scaling factor. For example, for MRI-CGCM3 the scaling decreases in "EAIS Atl" from RCP4.5 over RCP8.5 to RCP2.6, while in "EAIS Ind" the order is different from RCP8.5, RCP2.6, to RCP4.5 (Figure 5). It indicates again, that regional differences matter.

The region "WAIS" has significantly lower scaling factors than both "EAIS" regions. This difference exists for all ensemble means regardless of the applied scenarios and for almost all individual ensemble members (Figure A3), except some individual ensemble members under the RCP2.6 scenario (MIROC-ESM, MPI-ESM-LR) and the HadGEM2-ES.

The region "Siple Coast" (area $0.69 \cdot 10^6\,\mathrm{km}^2$, see Table 4) as a part of the "WAIS" region (area $4.26 \cdot 10^6\,\mathrm{km}^2$) is different in many aspects. It has the smallest area compared to the other regions (Table 4), and it shows the lowest ensemble mean scaling factors for all scenarios. Also, as before, no clear trend exists between different scenarios across the entire ensemble, while the spread of trends among individual ensemble members is substantial (Figure 5). Furthermore, some members exhibit a negative

scaling, where precipitation decreases for rising temperatures: MPI-ESM-LR under the RCP8.5 scenario and NorESM1-M under all scenarios (RCP8.5, RCP4.5, and RCP2.6). The inverted sign of the scaling is in stark contrast to the ensemble mean.

295     In the last decades, the detected downward trend in snow accumulation in this area (Wang et al., 2017) occurs while the wider West Antarctic Ice Sheet region belongs to the most rapidly warming regions globally (Bromwich et al., 2012). It underpins that less accumulation can befall under a warming climate. Furthermore, sea ice has expanded in the Ross Sea (Haumann et al., 2016; Liu, 2004). Hence, some ensemble members seem to imitate that expanding sea ice modifies the evaporation from the ocean and impacts the atmospheric circulation, which controls the flow of humid air masses, delivering precipitation to the

300     Siple Coast. Even if NorESM1-M reproduces the overall seasonal sea ice extent cycle better than most CMIP5 models (Turner et al., 2013), it shows an unrealistically declining February sea ice trend in the Ross Sea over 1979-2005 (Turner et al., 2013). MPI-ESM-LR has large negative errors in sea ice extent over the year (Turner et al., 2013). Hence the mimicry of observed features in models may occur for the wrong reason.

     In all four large regions ("glaciered", "grounded", "EAIS Atl", "EAIS Ind", and "WAIS"), we see a trend towards lower

305     scalings for weaker forcing scenarios in the ensemble mean, with the exception of "EAIS Ind", where the factors for RCP8.5 and RPC4.5 are indistinguishable. Also, Frieler et al. (2015) found a low dependence of the scaling factors to the RCP scenario in comparison with the dependence on the specific climate model. Here, the region "WAIS" has on average a smaller precipitation scaling than both regions of the East Antarctic Ice Sheet ("EAIS Atl" and "EAIS Ind"), which is also reflected by the scaling factor maxima in these regions (Figure 4). As before, the Ross Ice Shelf and the adjacent Siple Coast feature,

310     on average, the lowest scaling factors across the entire ice sheet (Figures 4 and 5). Some individual ensemble members project even negative scaling: precipitation deficit for rising temperatures (Figures 4 and 5).

[revised manuscript text omitted]

**3.4 Relation between Precipitation Boundary Condition and Ice Thickness**

Starting in the year 1850, we performed numerous ice-sheet simulations to analyze how the applied precipitation boundary condition impacts the ice-sheet thickness distribution. Each climate scenario from an individual climate model (as part of the

ensemble) drives an independent ice-sheet simulation. These together constitute the ensemble of 208 ice-sheet simulations (Table 1). Hence, the average across ice-sheet simulations forms the ensemble mean. For the diagnostic, we also inspect the maximum and minimum thickness at each grid point across all ensemble members. Therefore, the field of joined extreme values could come from a diverse set of ice-sheet ensemble members and, hence, does not necessarily lead to dynamically consistent distribution.

Complementary ice-sheet (control) simulations are performed under the sole utilization of the reference forcing fields (Figure 2a-c). In these simulations, the detected trend of about $2\,\mathrm{mm}\,\mathrm{decade}^{-1}$ (sea-level equivalent) fades within the first 400 years and differs slightly between the two initial states (PISM1Eq and PISM2Eq). Even if we apply anomalies on top of the reference background fields, we can not exclude a shock-like behavior of the simulations entirely directly following the decades after the year 1850. Since we compute the anomalies relative to the average over the first or the last 50 years, respectively, of the control run for each climate model, these anomalies are not necessarily zero at the beginning of the year 1850. Hence, the ice-sheet model may experience a small jump, which cause an artificial trend initially. Hence, in the following, the subtracted trend for each single ensemble member depends on its initial state.

In the year 2100, the ice thickness for both precipitation boundary conditions (precipitation anomaly deduced from the applied climate models versus scaled precipitation) increase over large parts of the Antarctic continent (Figure 7b-e). The thickness for the simulations driven by scaled precipitation grows less over substantial parts of the interior than in the simulations forced by the precipitation anomalies (Figure 7a), as the difference between scaled precipitation and applied precipitation anomaly is mostly negative. Thus, simulations driven by the precipitation anomalies accumulate more snow and grow thicker ice, which leads to a stronger sea-level drop. This result supports the analysis above (Figure 6). A ring of a pronounced negative thickness difference follows the coast, where the precipitation anomaly (Figure 2e, h, k) is enhanced. This ring emerges for a significant part of the coastal East Antarctic Ice Sheet (EAIS) and West Antarctic Ice Sheet (WAIS). For the latter ice sheet, the negative area is shifted away from the coast towards the interior (Figure 7a). Also, a negative strip of the thickness difference appears at the south side of the Transantarctic Mountain Range, and some grounded ice streams flowing into the Filchner-Ronne Ice Shelf.

Regions of positive differences coincide with thicker ice for simulations driven by scaled precipitation. These are located south of the Transantarctic Mountain Range at the northern edge of the Ross Ice Shelf, along the coastline of the WAIS, and in the coastal Terre Adélie region. There, the scaling is generally lower or falls behind the constant scaling of $5\,\%\mathrm{K}^{-1}$. However, this does not explain exclusively positive areas.

For both precipitation boundary conditions, the mean ice thickness of each of the respective sub ensembles reveals a widespread weakening of the floating ice shelves, such as Filchner-Ronne, Ross, and Amery Ice Shelves (Figure 7b, d). In the WAIS, both Pine Island Glacier and Ferrigno Ice Stream (an ice stream that flows into the Filchner Ice Shelf) thin drastically. Along the Antarctic Peninsula, general shrinking occurs along the coasts. Also along the marginal EAIS, ice thins.

For some places, the ice thickness thins for both precipitation boundary conditions across all ensemble members as the reduction of the maximal ice thickness highlights (Figure 8c, e). This reduction marks those outlet glaciers and ice shelves that are extremely vulnerable. These are around the Rutford Ice Stream, Foundation Ice Stream, Ronne Ice Shelf, Amery Ice Shelf,

395 three outlet glaciers (in "EAIS Ind" as part of Wilkens Land, Terre Adélie, and George V Land), northwestern Ross Ice Shelf (Ross Island), and Pine Island together with Thwaites Glacier in the Amundsen Sea (Figure 8c, e).

To conclude, the ice thickness is indeed thicker for simulations driven by the precipitation anomalies (Figure 8). Regardless of the applied precipitation boundary condition, there is widespread ice-shelf thinning, as well as ice-shelf weakening, at the margins in the ensemble mean. Ice thinning for the ensemble member of the maximal thickness highlights the most vulnerable

400 regions, such as Pine Island and Thwaites Glaciers, Amery Ice Shelf and some outlet glaciers of the EAIS.

**3.5 Attribution of the driving model**

All ensemble members contribute to the ensemble mean, while at a given grid location the maximum and minimum are determined by climate forcing from one particular climate model. We inspect which climate model may lead to ice thickness growth or shrinking and restrict ourselves first to the model year 2100, when the transient forcing of period 1850–2100 excites

405 changing ice thicknesses.

Directly at margins apart from the vast ice shelves, the attributed model that drives either the maximum or minimum ice thickness shows a noisy small-scale pattern (Figure 8d, e). Hence, the marginal regions cannot be associated with a particular climate model. In contrast, the mean and minimum thicknesses of the Filchner-Ronne and Ross Ice Shelves, and also to some extent the Amery Ice Shelf, are highlighted by a nearly unique color patch indicating a reduced thickness. These patches are

410 separated from the surroundings showing either a reduced thinning or even thickening. Intriguingly, the MIROC-ESM model forcing, for instance, thickens grounded ice east and west of the Ross Ice Shelf (Figure 8d), while it also predominantly thins the Ross Ice Shelf (Figure 8e). Hence, the ocean forcing drives the ice-shelf thinning here. Since the spatial pattern of the atmospheric and ocean forcing that promotes or undermines the ice thickness is not necessarily aligned, this may explain the small scale noisy pattern along the coast. Also (nonlinear) dynamical changes on the considered time scales may occur in

415 response to both ocean and atmospheric forcing.

Beyond the direct coast strip, larger areas appear where the forcing from one climate model determines the maximum or minimum thickness, respectively. However, these extended continuous regions are often interrupted by spots controlled by the climate from other models. Also, the pattern is changing during the transient simulation starting in 1850 because the temporal evolution of the 2m-air temperature and precipitation anomalies are different for each climate model as the integrated

420 precipitation highlights (Figure 6a, b). Furthermore, after the year 2100, where the same 30 years forcing period (2071–2100) drives the ice-sheet model recurrently, the pattern evolves further (Figure 9). Because the ice sheet has not reached the quasi-equilibrium to the last 30 years forcing, the pattern alteration is ongoing.

For grounded ice, three models (CCSM4, CNRM-CM5, MIROC-ESM) determine predominantly the growing ice until the year 2100 (Figure 8d), which is in-line with the diagnosed sea-level contribution (solid line, Figure 6a, b). CCSM4 dominates

425 the "EAIS Atl" sector, while CNRM-CM5 dominates a band from the "EAIS Ind" sector clockwise to the Antarctica Peninsula, which is interrupted by regional-scale patches of the MIROC-ESM. A spatial dominance is not apparent for the minimum ice thickness, because the patchwork of five models (CSIRO-Mk3-6-0, HadGEM2-ES, MPI-ESM-LR, MRI-CGCM3, NorESM1-M) dominates the year 2100. NorESM1-M influences the WAIS, which is supported by its lowest scaling in the Siple Coast

region (Figure 5). CSIRO-Mk3-6-0 has an impact around the South Pole, MRI-CGCM3 affects the coastal zone in the EAIS. The control of MPI-ESM-LR and, to a lesser extent, HadGEM2-ES spreads across the entire continent. If we progress into the year 2200, where we have applied the 30 years forcing more than three times, the emerging picture shows a consolidation of the influential spheres of the different models for both the maximum and minimum thicknesses (Figure 9).

If we now turn towards those model simulations, in which the temperature-scaled precipitation forcing has been applied, both the mean, maximum, and minimum ice thickness distribution (Figure 10) are similar to the ones driven by the precipitation anomalies as discussed above (Figure 7). Also, the same models determine the ice-shelf thickness of the Filchner-Ronne and Ross Ice Shelves. The latter shows that primarily the ocean controls ice-shelf thickness changes in our simulations. However, we detect a stark contrast of the model determining the maximum and minimum ice thickness. For the maximum, we still have the same three models (CCSM4, CNRM-CM5, MIROC-ESM). However, the pattern has changed. CCSM4 controls a smaller area in the interior around the South Pole, and MIROC-ESM some coastal regions of the East Antarctic Continent. The remaining majority of the grounded ice is under the control of CNRM-CM5. The most striking changes occur for the minimum. Now, NorESM1-M determines the entire WAIS and also some parts of "EAIS Ind". MRI-CGCM3 dominates the remaining East Antarctic Ice Sheet.

In the latter case, temperature variations force the precipitation-driven ice thickness evolution exclusively (see Equation 1). These temperature changes do not necessarily reflect dynamical changes in the atmosphere that are accompanied by modified circulation patterns that ultimately transport and deliver the precipitation for Antarctica. Hence, the applied scaling or precipitation boundary condition impacts the temporal evolution of the Antarctic Ice Sheet geometry, which ultimately shapes Antarctica's contribution to the global sea level.

**3.6 Ice losses**

After the spin-up, the simulations have reached a quasi-equilibrium. For the discussion of the ice losses, we concentrate on the transient period 1850-2100. For all climate scenarios, the calving rate hardly changes (Figure A12), whereas the total ice-shelf area is nearly constant until 2000 and declines afterward (Figure A15). The ocean-driven basal melting is proportional to the squared temperature difference between the pressure-dependent melting temperature and the actual ocean temperature. Since the ocean temperature increases in general (Figure 2f, i, l and Figure 3c), also the mass loss by basal melting increases, while the total shelf ice area remains quasi-constant until 2000 and declines afterwards (Figure A15). For RCP8.5, the basal melting increases at the end of the 21st century quadratically. To conclude, the calving rate is nearly constant, while the basal melting increases by approximately $33\%$ between the years 2000 and 2100.

The mean calving rate is about $8000\,\mathrm{Gt\,year^{-1}}$ and $5000\,\mathrm{Gt\,year^{-1}}$ for the ensemble member utilizing the parameters and the initial state of PISM1Eq and PISM2Eq, respectively (Figure A12). The basal melting rates for PISM1Eq and PISM2Eq are similar, however, the loss rates for PISM1Eq are slightly larger than PISM2Eq (Figure A13). The ensemble mean starts at about $550\,\mathrm{Gt\,year^{-1}}$ in 1850 and reaches $900\,\mathrm{Gt\,year^{-1}}$ in 2100.

Since floating ice shelves nourish both ice losses, these ice losses do not directly impact the sea-level. Under the assumption that the inflow of former grounded ice compensates any shelf mass loss, the reported ice losses of $8500\,\mathrm{Gt\,year^{-1}}$–

9000 Gt year$^{-1}$ (5500–6000 Gt year$^{-1}$) would correspond to a sea-level rise of 2.58 cm year$^1$–2.74 cm year$^1$ (1.67 cm year$^1$–1.83 cm year$^1$). The Integration over 250 years to match the period from 1850 to 2100 would generate a potential sea-level

465   equivalent of 6.47 m − 6.85 m (4.19 m − 4.57 m). However, the actual ratio between total ice mass change and the corresponding potential sea level response is obviously not a 1:1 relation. Instead, on average less than 5 % of the total mass lost by both iceberg calving and floating ice-shelf melting is compensated by grounded ice that raises the sea level (Figure A8). Considering this ratio of 5 %, the sea level impact reduces to 0.32 m − 0.34 m (0, 21 m − 0.23 m) by 2100. It is less than integrated precipitation anomalies across the Antarctic continent (Figure 6a), which explains the total mass gains.

470     Anyhow, the integrated basal melting rates are too low (Figure A13 and the calving rates are too high (Figure A12) compared to observational estimates in our ensemble of ice-sheet model simulations. Besides the fact the total loss exceeds recent observational estimates, our ice sheet is in a quasi-equilibrium after the spin-up. All this may indicate that the integrated precipitation driven accumulation resulting from the RACMO precipitation reference field might be too large. However, the surface mass balance of RACMO agrees well with observational estimates (Wang et al., 2016), while the uncertainty of the

475   surface mass balance (sea-level equivalent of $\sim 0.25$ mm year$^{-1}$ (Van Wessem et al., 2014)) is of almost the same size as Antarctica's observational-based sea-level contribution ($\sim 0.2$ mm year$^{-1}$ between 1992 and 2011 (Shepherd et al., 2012; Wang et al., 2016)). Additionally, recent satellite-based estimates indicate clearly that the Antarctica Ice Sheet have lost mass (sea-level equivalent: 0.4 mm year$^{-1}$) in the period 2011–2017 (Sasgen et al., 2019).

[revised manuscript text omitted]

By construction, the correct time series preserve the fluxes' amplification over time, which is essentially the ratio of the higher end value to the lower start value. Hence, the corrected basal melt flux replicates the original simulated amplification while the flux is identical to the observed reference value ($F_{ref}(t_{ref})$) at the reference time ($t_{ref}$). Under the assumption that only a fraction of the adjusted basal mass contributes to the global sea level, we apply the simulated ratio of the sea level change to the total ice mass change. For each ensemble member, this ratio is the median ratio over its entire time series (for details see Section A3 "Bias-corrected fluxes" on page 49 in the appendix). Since we examine enhanced mass loss, we do not adjust the iceberg calving rates that are already higher than observed.

By adjusting the basal melting flux, the determined temporal evolution of the sea level correction (Figure A5, Equation A8) does impact the global simulated sea level. Still, it does not change the sign of the contemporary sea-level evolution. Consequently, the impact on the simulated sea level is very small (Figure A6). If we assume instead that all of the additional mass loss of floating ice shelves rises the simulated ea level immediately, we would obtain too extensive corrections of 30 cm between 1850 and 2000. This corresponding sea-level rise would be larger than the observed integrated sea level rise of about 20 cm since 1850 (Church and White, 2011), which has been driven by world-wide land-water storage changes, shrinking glaciers

around the globe, enhanced melting from Greenland, and thermal expansion of the ocean (Cazenave and Remy, 2011; Leclercq et al., 2011; Church and White, 2011).

To conclude: The correction exceeds observational estimates significantly under the unrealistic assumption that all additional basal melting of ice shelves would raise the simulated sea level. It is unrealistic because disintegrating floating ice shelves do not impact the sea level directly. The correction hardly corrects the discrepancy if we apply the inferred ratio of about $5\%$ between the simulated total ice mass loss and the simulated sea level contribution.

**4   Conclusions**

It is crucial for numerical simulations of Antarctica's sea-level contribution, how the precipitation is specified in ice-sheet simulations. The commonly used method of scaling the precipitation changes with the simulated temperature changes from ice cores or global climate models leads to a positive Antarctic simulated sea-level contribution, i.e., a simulated sea-level rise. However, when considering the simulated precipitation changes from the global climate models, the situation changes. In this case, our numerical projections simulate a negative sea-level contribution. Major uncertainties affect these simulations, such as the partitioning of ice losses into calving and basal melt — which is quite different from observational estimates due to very crude representations in the ice-sheet model — or the omission of important processes, such as the interaction between ocean, ice shelves, and ice sheet. While we could improve some aspects of the involved process descriptions, our simulations are state-of-the-art and suffer, thence, the same limitations as others.

In all CMIP5 models, the 2m-air temperature warms across the entire Antarctic continent without any exception (Figure 2d, g, j, and 3a), because even the minimum 2m-air temperature anomaly is positive everywhere (Appendix Figure A2d, g, j). The warming enhances the hydrological cycle, which causes generally heavier precipitation (Figure 3b) in particular along the coast of Antarctica (Figure 2e, h, k). However, the changing precipitation does not increase at the same rate with increasing temperature because it is not only thermodynamically influenced but also dynamically controlled. Given that the ensemble mean temperature scaling is different for the West and East Antarctic Ice Sheet (Figure 5) and has a considerable spatial dependence, the dynamical component is not negligible. Instead, the region of reduced precipitation under rising air temperatures, which we have identified along the Siple Coast, highlights that the dynamics could compensate or even overcome the thermodynamics. The continent-wide scaling is per se problematic, even if we would adjust the scaling factor to reproduce the continental-wide average scaling. In this case, the integrated precipitation would be identical, but the spatial structure is still entirely different (Figure 4). Hence for a realistic projection of Antarctica's sea-level contribution, the spatial pattern of the future accumulation of precipitation shall also consider the dynamical effect.

Independent of the applied precipitation boundary condition, we detect regions where the ice thickness thins for all ensemble members. These regions are the Amundsen Sea Embayment with both Pine Island and Thwaites Glaciers, some outlet glaciers of the East Antarctic Ice Sheet (EAIS) between George V and Wilkens Land, Amery Ice Sheet, and the Northern Antarctica Peninsula. These regions correspond to those, which have been identified across sixteen models within a recent model intercomparison exercise, where ocean warming wanes marginal ice (Seroussi et al., 2019a).

The ocean (Etourneau et al., 2019) and atmosphere (Mulvaney et al., 2012; Thomas et al., 2009; Morris and Vaughan, 2003) is already warming along the Antarctic Peninsula. This results in a southward progressing of the annual 2m-air temperatures of -9°C or -5°C isotherm, which presents the range of thresholds for the stability of ice shelves suggested by Morris and Vaughan (2003, -9°C) and Doake (2001, -5°C), respectively. It may also enable the formation of meltwater ponds on ice shelves (Kingslake et al., 2017) that precedes (van den Broeke, 2005) or even triggers ice-shelf disintegration (Banwell et al., 2013, 2019). After an ice shelf has decayed, the feeding ice streams are losing more ice, as seen for Larsen-B (Rott et al., 2011), which lowers the thickness of grounded ice. Anyhow, ice shelves along the Antarctic Peninsula have collapsed or are retreating (Cook and Vaughan, 2010; Rott et al., 1996). In our simulations under the RCP8.5 scenario, this observed retreat and the related ice loss will continue.

For part of the EAIS, simulations show that grounded ice of the Wilkens Basin in the hinterland of George V Land may be prone to a massive ice loss if the ice front loses its buttressing effect (Mengel and Levermann, 2014). Our ensemble shows, on average, a stable situation here. Ice in deep troughs that are in contact with the warming ocean thins at some spots further to the west. It happens in front of the Astrolabe Trench (in Terre Adélie) and on the coast of the Wilkens Land, for example near the Totten Glacier. Ice also thins in the deep trench leading to the Amery Ice Shelf.

Both the Pine Island and Thwaites Glaciers in the Amundsen Sea as part of the marginal West Antarctic Ice Sheet lose ice (Jeong et al., 2016; Milillo et al., 2019; Rignot et al., 2014; Scambos et al., 2017). According to the ensemble projecting the future, for them, continuous ice loss is inevitable. It also shows that the Ferringo Ice Stream flowing into the Bellingshausen Sea will thin in the future.

Since our simulations presented here are in contrast to others that project a sea-level contribution from a shrinking Antarctic Ice Sheet, we highlight the differences before we discuss the limitations of our simulations. Some 
[revised manuscript text omitted]
, only for MIROC-ESM the reference state (first vs. last 50 years piControl) matters, because this model is subject to not negligible trend (0.08 m) during the piControl phase. For instance, the average of the global absolute 2m-air temperature difference between the first and last 50 years of piControl amounts 0.17 K (median 0.12 K) for all CMIP5 models considered in our study. In contrast, MIROC-ESM's value is 0.67 K.

In future projections, the basal melting rate increases between 10 % and more than 100 % until the year 2100 relative to the 50 years reference period 1951–2000. The latter increase is consistent with results from dedicated ocean simulations. These simulations resolve ice shelves, include the ocean-ice-sheet interaction explicity, are driven by future projection from various climate models (Naughten et al., 2018; Hellmer et al., 2012).

The temporal evolution of the actual basal melting rate (Figure A13) increases until 2100 and falls back afterward onto the value of the year 2071 because we apply the last 30-years-forcing recurrently after 2100. Also, for the basal melting the separation of ensemble members starting from PISM1Eq and PISM2Eq is self-evident. However, both groups are close to the ensemble mean, which is in contrast to the calving rate. The basal melting rates of all ensemble members underestimate the observational basal melting rates.

[revised manuscript text omitted]

$$q = q(t_{\text{ref}}) = \frac{F_{\text{ref}}(t_{\text{ref}})}{F_{\text{org}}(t_{\text{ref}})}. \tag{A2}$$

The corrected flux $F_{\text{cor}}$ using Equation A1 is defined as

$$F_{\text{cor}}(t) = r(t) \cdot F_{\text{ref}}(t_{\text{ref}}), \tag{A3}$$

so that the flux difference $\Delta F(t)$ is

$$\begin{aligned}
\Delta F(t) &= F_{\text{cor}}(t) - F_{\text{org}}(t) \\
&= F_{\text{org}}(t) \left[ \frac{F_{\text{ref}}(t_{\text{ref}})}{F_{\text{org}}(t_{\text{ref}})} - 1 \right].
\end{aligned}$$

With Equation A2 we obtain

$$\Delta F(t) = F_{\text{org}}(t) \left[ q - 1 \right]. \tag{A4}$$

To relate the sea-level change to the ice mass evolution, we define the ratio $p(t)$ of the sea level temporal deviation to the ice mass temporal deviation as

$$p(t) = \frac{\dfrac{dz_l(t)}{dt}}{\dfrac{dm_{\text{ice}}(t)}{dt}}, \tag{A5}$$

where $z_l$ is the sea level and $m_{\text{ice}}$ the total ice mass, which includes grounded and floating ice. We use here $p = \text{median}(p(t))$ so that each ensemble member is characterized by one value for its entire time series. If $p = \frac{1}{\rho A_{\text{oce}}}$, $100\,\%$ of flux difference (Equation A4) contributes immediately to the sea level of the global ocean with an area of $A_{\text{oce}}$.

The total ice mass ($m_{\text{ice}}$) changes are driven by four terms

$$\frac{m_{\text{ice}}}{dt} = \underbrace{\left[ F^{SMB}(t) + F^G(t) \right]}_{\text{unchanged under correction}} + F^B(t) + F^D(t),$$

where $F^{SMB}(t)$ is the surface mass balance flux and $F^G(t)$ the basal mass flux of grounded ice (Figure A9). We assume that these two terms in the brackets do not change regardless of the applied corrections to the last two terms $F^B$ and $F^D$. Hence the difference in the ice mass change is

$$\Delta \frac{dm_{\text{ice}}}{dt} = \Delta F^B(t) + \Delta F^D(t) \tag{A6}$$

Now we relate the temporal evolution of the sea level to the total ice mass changes by utilizing the Equation A5

$$\frac{z_{l\,\text{cor}}}{dt} = \frac{z_{l\,\text{org}}}{dt} + p(t) \cdot \left[ \Delta F^B_{\text{cor}}(t) + \Delta F^D_{\text{cor}}(t) \right],$$

so that we obtain:

$$z_{l_{\text{cor}}} = z_{l_{\text{org}}} + \Delta z_l(t), \tag{A7}$$

1110     where the sea-level difference $\Delta z_l(t)$ is

$$\Delta z_l(t) = \int_{t_0}^{t} p(\hat{t}) \left[ \Delta F_{\text{cor}}^{B}(\hat{t}) + \Delta F_{\text{cor}}^{D}(\hat{t}) \right] d\hat{t}. \tag{A8}$$

The Figures A5 and A7 depict the sea-level difference for two cases. If the additional mass loss contributes immediately to a rising sea level (Figure A7), the corresponding sea level rise of $30\,\text{cm}$ would be larger than the actual sea level rise since 1850 of about $20\,\text{cm}$ (Church and White, 2011).This case is not realistic because a melting floating ice shelf does not impact

1115     the sea level. Only the flow of grounded ice across the grounding line, to feed an ice shelf, or the direct loss of grounded ice contributes to the sea level.

In contrast, the sea level changes hardly (Figure A5) if the deduced ratio $\overline{p(t)}$, which corresponds to the ratio defined in Equation A5. It is computed for each ensemble mean as median of its time series. Whether the ratio between ice loss and sea-level rise is constant under amplified basal melting of ice shelves is an open question. Strongly intensified ocean-driven ice

1120     loss will probably cause a retreating grounding line on a longer time scale, which ultimately releases grounded ice into the sea and increases the sea level.

Figure A8 shows the proportion of the deduced ratio to the $100\,\%$ ratio. Only very few ensemble member lose about 15% of the maximum value of $p = 1/(\rho \cdot A_{\text{oce}})$. In contrast, the mean and median value of this proportion is generally less then $5\,\%$. For all ensemble members driven by the precipitation anomaly, this proportion is on average $4.7\,\%$ with a median of $3.9\,\%$.

1125     It is even lower for ensemble members driven by the temperature-scaled precipitation. The median amounts $0.7\,\%$ and the corresponding mean is $0.9\,\%$. Please note that some ensemble members under the temperature-scaled precipitation are subject to a negative scaling. This result confirms the above presented low positive and negative scaling seen for restricted regions (Figure 5). It highlights also that simulations driven by temperature-scaled precipitation could show unexpected results.

[Figure]

**Figure A1.** Atmospheric (a, b) and oceanographic (c) reference forcing; ensemble maximum anomalies (d–l). The top row represents the reference fields to spin-up the ice-sheet model (Table 2). The 2m-air temperature (a) and the total precipitation (b) are mean fields from the regional RACMO model, while the ocean temperatures come from the World Ocean Atlas 2009 (c). Below each reference field, the related maximum anomalies are compiled for the period 2071–2100. Here, the second (third and fourth) row shows the anomalies for RCP8.5 (RCP4.5, RCP2.6). The dark-gray line follows the current coastline. All potential ocean temperatures (c, f, i, l) are a vertical mean of the depth interval from $150\,\text{m}$ to $500\,\text{m}$. The white contour lines in the anomaly plots highlight the following thresholds. 2m-air temperature (d, g, j): $8°\text{C}$; total precipitation (e, h, k) $50\,\text{cm\,year}^{-1}$. All these anomalies are the ensemble maximum of the models listed in Table 1; CCSM4 is not part of RCP2.6. Figure 2 shows the corresponding ensemble mean fields, where the white contour line in the precipitation field corresponds to $30\,\text{cm\,year}^{-1}$.

[revised manuscript text omitted]

---

## Referee Report (RR2)

Review for the "Future Sea Level Contribution from Antarctica inferred from CMIP5 Model Forcing and its Dependence on Precipitation Ansatz" by Rodehacke et al.

In this manuscript, the authors describe an ice-sheet modeling study of the evolution of the Antarctic Ice Sheet under a number of different forcing scenarios from various climate models. In particular, the authors are interested in determining how ice sheet mass balance changes in response to the treatment of accumulation within the simulation. The results of the ensemble illustrate that simulations forced with anomalies respond differently than simulations forced with precipitation that is scaled according to changes in temperature (a method that is often utilized for long ice sheet model simulations). The authors conclude that the latter method is unsuitable for capturing the dynamic response of accumulation rates due to atmosphere and ocean to warming. Therefore, a precipitation scaling scheme is not as realistic as forcing an ice-sheet model with the spatial and temporal anomalies derived from climate model projections. While specific areas of the ice sheet contribute to sea level with the use of either precipitation-forcing scheme, other areas have responses that disagree on the sign of their sea-level contribution. The authors caution that the choice of how future precipitation is simulated may strongly influence future projections of the Antarctica Ice Sheet and contribute to significant model uncertainty.

This study uses PISM, a state-of-the-art ice-sheet model to run a number of simulations to investigate the sensitivity of future projections of the Antarctic Ice Sheet to the model treatment of accumulation. The authors conduct a variety of simulations and thoroughly present results from experiment variations in order to illustrate that their results, most importantly, are robust with respect to the comparison between forcing schemes. The text and figures are comprehensive and the conclusions will be of interest to the ice-sheet modeling community. In addition, the authors have made significant improvements on the manuscript language and structure, as well as the title and abstract. The figures are very readable and support the main discussion points. As a result, I support the publishing of this manuscript in ESD.

Note, however, that I still find that in a few locations in the manuscript, the language could be improved, for readability. The meaning of some statements tends to get lost from time-to-time. For these such statements, and some other minor points, I make suggestions and comments, listed below.

Comments/suggestions:

Line 37: "An" ice sheet's contribution

Line 45: "the temporal evolution is spatially homogeneous" => Please be clearer here. Maybe "often, in these experiments, the anomalies forced through time are spatially homogeneous" ? Or something similar.

Line 53: "globally" => "when considered globally, this rate"

Line 53: "termed" => "hereafter, referred to as mean precipitation scaling ... "

Line 59: "limits the scaling" => "exposes a limitation of this scaling"?

Line 77: "this approach" is unprecedented

Line 83: "Following, is the description" is what I think you mean here

Lines 85-94: I suggest that this section go after or is appended to the end of section 1.2 (or it could even be moved to the methods before the scaling description). It seems out of place here, and may be more impactful if it is present after the reader is introduced to the idea of precipitation scaling.

Line 94: "further on" => "from this point on"

Line 136: "they differ in" => "they differ on"

Line 137: "Past studies suggest that the overall mass loss…" or something similar

Line 162: However, "in reality", atmospheric dynamics … (or an equivalent statement for clarity)

Line 173: I think you mean here that piControl is of different lengths for the different models. Could the length of these piControl runs be added to a table somewhere for reference?

Line 183: "ends in" => "ends up in"

Line 185: "For instance", a disintegrating Greenland Ice Sheet (or a similar phrase)

Line 189: "On the other hand, locally …" (?)

Line 191: "probably" -> "likely"

Line 251: utilize these constants "respectively" (?)

Line 235: The last part of this sentence is awkward. Maybe something like: "any signal, though the SMB computed via PDD does allow for melting …"

Line 253: I think here you refer to the mean ensemble, spatial mean over Antarctica? Please make this clear if so.

Line 280: "Least" -> "The least"

Line 307: It is unclear what "(probably)" means in this context. Please remove it, or clarify what it means for the reader.

Line 319: "risen" -> "elevated" or similar

Line 321: "we could" => "could we"

Line 327: I am not sure what "it excludes floating ice shelves with low elevation along the coasts" means. Please rephrase.

Line 391: Can you add a statement here (or when you discuss this later in the text) about what year in the simulation the response to this shock probably becomes negligible?

Line 423: "rated" => "rates"

Line 484: "However", ice in deep troughs… (or the equivalent)

Line 513: This first sentence is very awkward to read. Please rephrase for clarity.

Line 536: Perhaps a word is missing here. I am not sure what you say is consolidated.

Section 4.2: This is just a suggestion, but perhaps consider moving this section to the end of the discussion. Placing it here breaks up the flow of your nice discussion, and seems disjointed, especially since you have not even explained how you correct basal melt and the implications.

Lines 570-573: These sentences are awkwardly phrased, and their meaning is not clear. Please rephrase them.

Line 580: allow "to represent the" regional conditions => allow "the representation of" regional conditions

Line 617: is compensated by "accumulation on" grounded ice ?

Line 649: "ea" => "sea"

Lines 654-657: These conclusions should be rephrased to be very clear for the reader. They are a bit confusing to read as written. I suggest that you use these sentences to directly answer, for the reader, the question that you pose on line 637.

Line 656: Please specify which "discrepancy" in particular you refer to here

Lines 659-660: Awkward, please rephrase. Maybe "How precipitation is specified in ice-sheet simulations is crucial to the outcome of numerical simulations of Antarctica's sea-level contribution." Or something similar.

Line 668: where "marginal ice wanes due to ocean warming"

Line 669: "as average" => "on average"

Figures 2, A1, A2 captions: Please start each of these captions with the sentence that you place towards the end. i.e. "Anomalies of the CMIP5 … (mean, max, min) " Or a similar sentence that summarizes where the anomalies are from. That is, if all three figures were next to each other, the reader would be able to know how they are different by just reading the first sentence in the caption. Right now, the reader has to read very far into the caption to differentiate the figures.

Lines 1069-1070: The sentence is written twice.

Line 1071: "importance" => "consequence"

Figure A5 caption: "depicts the" => "is depicted in the"

Figure A10 caption: "lists the" table A1 => "is listed in" Table A1

---

## Author Response (AR2)

The following is a review of "Precipitation Ansatz dependent Future Sea Level Contribution by Antarctica based on CMIP5 Model Forcing" By C. B. Rodehacke and others.

This manuscript describes the execution of an ensemble of ice-sheet model simulations. The experiment is designed to explore how the treatment of, and estimating method for, precipitation in a continental ice-sheet model simulation may impact future estimates of Antarctic mass balance and potential for sea-level contribution. The authors make use of the state-of-the-art PISM model of the Antarctic Ice Sheet, and run two different precipitation strategies with a range of emission scenarios, Earth system models, and ocean forcing. The manuscript includes a thorough examination of the model response to these variations, and the authors spatially and temporally investigate the differences resulting from use of the two different precipitation-forcing methods. The authors highlight which areas are susceptible to mass loss, no matter which forcing is applied (i.e. West Antarctica). However, in the rest of Antarctica, the authors find a varying degree of different precipitation patterns within their ensemble. Most importantly, results using the different precipitation methods do not agree on whether future forcing would result in overall thickening or thinning of the East Antarctic Ice Sheet. These results suggest that ice-sheet model projections may have a strong dependence on how precipitation is determined, and the simulation of future precipitation may constitute a significant uncertainty in projections of the Antarctic Ice Sheet.

Overall, I find this is a well-designed study. The authors cover a wide-range of variations in their experiments, and the results are comprehensively discussed. The figures are also very helpful in representing all the results from the full set of simulations, and they represent a thorough depiction of model results. The results presented are interesting and will certainty have an impact on how ice-sheet projections are conducted in the future. Therefore, I support publication of this manuscript in ESD, with suggested edits. Please find comments and suggestions to the authors below.

*Thank you very much for the review and the encouraging comments. Our comments are in blue/italic. After responding to the raised issues, the differences between the former and the revised manuscript are highlighted.*

General Comments:

In general, I find that the authors do a good job of describing their experiments and that they use language effectively to convey their points. There are some locations in the text, however, where I feel that the sentences are awkwardly phrased, and I note a number of these below. I have gone through the reviewer comments and responses from the authors with respect to the first submission of this proposal. Overall, I find that the authors have done a good job of responding to the comments and suggestions from the reviewers. However, I do note that in many cases where confusing and/or awkward wording is present, this phrasing is the result of new text that was added during the revision process. Since I am suggesting a number of edits below, I urge the authors to read their newest revision and to make sure that new edits are clear and specific, and that the additional phrases flow with the rest of the surrounding text. Also, the authors should make sure to use precise language, especially when referring to ocean vs. atmospheric temperatures; when using the term "model", which could refer to any number of models utilized in this study; when describing (and enumerating) the ensemble simulations; and when referring to the years and types of model forcing (i.e. years of forcing, and whether it is a transient forcing or an average forcing).

*Following your suggestion, we add to nearly every occurrence of temperature either air or ocean to distinguish clearly between ocean and air temperatures.*

*Guided by reviewers comments and our aim to reduce ambiguity, we renamed "CMIP5 ensemble" into "CMIP5 data set" or "CMIP5 models" while we keep the label ensemble for the ice sheet simulations.*

In addition, I noted that there are a few important comments that the authors did not address in their response to Reviewer 1. In my opinion these were important points, and I think they should result in additional edits during this round of reviews.

More specifically, I think that the authors could still put some time into improving the readability of this long manuscript, by separating the sections in a more comprehensive way. Specifically, the Results and Discussion section contains methods, results, and discussion. I suggest that the methods contained in the Results be placed in the Methods and Materials section, and the Method and Materials section be organized into sub-sections (e.g. separating the forcing with the Ice Sheet Model description, as commented below). Similarly, the Results and Discussion can be separated into a Results and a Discussion section, and much of the current Conclusion section (which involves discussion of study limitations) can be placed into its own subsection of the Discussion (commented on more below).

*Following your suggestion, we have split the article in smaller confined units by introducing so-called "subsubsection" in the Latex source code. In addition, we move the suggested text blocks. Please see also your related specific comments below.*

With respect to the Conclusion, I agree with Reviewer 1 that the conclusions should be only a couple paragraphs and summary the main conclusions for the reader (in case those were lost in the many pages of text). The Conclusion section does do this now, but the many limitations take away from the interesting summary of your findings. Since the text is so long, the authors should take a serious look at reorganization of the manuscript, with the reader in mind. This should allow them to strengthen their conclusion as well, and include comments on what their results suggest about uncertainties in ice-sheet model projections (see additional comments to this point below). A strong, clear conclusion will help put this important work in context and take claim to some well-deserved findings.

*The rewritten Conclusion is shorter and concise.*

Finally, the title, as stated, has an awkward word ordering. Reviewer 1 noted this, and I agree. I suggest something like "Future Sea Level Contribution from Antarctica due to CMIP5 Model Forcing and its dependence on Precipitation Ansatz." Having a title that is readable will be beneficial for attracting readers and will help your manuscript make its intended impact on the modeling community once your study is published.

*Thank you very much for the suggested title. We have adapted it and use "Future Sea Level Contribution from Antarctica inferred from CMIP5 Model Forcing and its Dependence on Precipitation Ansatz."*

Additional Specific comments/suggestions:

Abstract- In general, this abstract summarizes many of the results. However, the wording could be made clearer with simplification of the sentences (for instance, your plain language summary is very clear). Also, in some cases you could directly state your results (linespecific comments below). Finally, I suggest you use ansatz in the abstract, so that a general reader would know what your title means (or at least deduce it from how it is used).

*Addressed by the rewritten abstract.*

Line 4:  Missing word – nine CMIP5 "models"?  Also, since it is the projections that are ranging not the models, please rearrange the sentence as something like: "future projections, ranging from strong mitigation efforts to business-as-usual, from nine CMIP5 earth system models to run an ensemble…", to be clearer.

*Indeed, this is much clearer. We consider this suggestion in the entirely rewritten abstract.*

Line 5: "In contrast to various former studies, only the historical (1850–2005) and scenario (2006–2100) forcing drive our ensemble of simulations, which neglects unavoidable continuous warming consistent with the higher climate scenarios beyond the year 2100."   I was not sure what this sentence meant until I read your response to Reviewer 1. Please try to state this point in a simpler way – that is that you, for instance, 'run your simulations with forcing derived from 1850-2100 CMIP5 output, so results past 2100, in contrast to previous studies, do not represent projections.'

*We consider this suggestion in the entirely rewritten abstract: "Since CMIP5 projections cover only the period from 1850 to 2100, our, until the year 5000, prolonged simulations neglect any unavoidable continuous warming consistent with the higher climate scenarios beyond the year 2100. Hence, past the year 2100, our study may underestimate ice loss."*

Line 8: "The spatially and temporally varying climatic forcing" – this is also a vague statement. I think here that you are trying to say that you run the full ensemble, using various rcp scenarios, derivation of anomalies, and models to investigate the full spread of model realizations. Please be more specific.

*We consider this suggestion in the entirely rewritten abstract: "In contrast to various former studies [...], our simulations consider the spatial structure in the forcing coined by various climate patterns. This fundamental difference reproduces regions of decreasing precipitation despite general warming."*

Line 11: "…in a broad marginal strip…" Please be more specific about where this strip is located.

*We have rewritten the entire abstract and write here: "along the coast"*

Line 13:  The change to referencing boundary conditions here is confusing. Could you say "forcing" instead to be clear that this refers to the same CMIP5 forcing you refer to earlier in the abstract?

*Here we write: "... between the applied precipitation methods."*

Line 15:  as "an" invariant scaling constant

*Done as suggested.*

Line 27-28: "The discrepancy of the simulation results between both methods describing the precipitation illustrates the uncertainty of the possible range of future precipitation growth in a warming atmosphere."  This sentence is a very nice summary of your results. Please add something similar into the actual abstract, since currently, there is no equivalent concluding statement.

*As suggested, we close the rephrased abstract with: "The discrepancies in response to both precipitation ansatzes illustrate the principal uncertainty in projections of Antarctica's sea-level contribution."*

Line 36: perhaps (Church et al., 2013a) should come after "behavior" since the past IPCC reports did not consider ice sheet models. Then, perhaps reference some ice sheet models that have been used for future projections (or maybe ISMIP6) after "ice-sheet models".

*Done as suggested.*

Line 42:  This could be stated more clearly. For this study, for example, you construct an ensemble that includes the full range of 21st century spatial and temporal patterns of atmosphere/ocean exhibited by CMIP5 models, and this ensemble is used to drive hundreds of PISM simulations (instead of just running one run or a small subset of runs).

*Rephrased: "We use the historical climate scenario (1850–2004) followed by three future climate scenarios(RCP2.6, RCP4.5, RCP8.5; 2005–2100) from a compilation of nine CMIP5 models (Table 1) to drive for each climate projection one simulation with the Parallel Ice Sheet Model (PISM, e.g., Bueler and Brown, 2009; Winkelmann et al., 2011). We compare these ice sheet simulations considering the spatial inhomogeneities in transient climate forcing with more traditional simulations in which the precipitation forcing anomaly is scaled with the temperature forcing anomaly."*

Line 63: Adding a concluding statement to this paragraph would be helpful to show your point. Stating something about the fact that these studies suggest that the scaling is highly variable, across ocean and land, and would be expected to not be well captured over Antarctica with a single scaling.

*Following your suggestion, we add: " Since the interplay between thermodynamic and atmospheric dynamics governs the scaling, it is unlikely to represent this scaling by a single value across Antarctica."*

Line 93: surface mass balance (SMB) is "estimated to be"

*Done.*

Line 105:  Specify that here you refer to WAIS.

*We write: "To conclude, across the West Antarctic Ice Sheet, the ..."*

Line 109: And dynamic grounding line migration. Also, note the years for which these quoted fractions are relevant.

*We add "dynamical grounding line migration" as process and highlight the missing period "1995-2009."*

Line 130: "In particular, the ansatz of the precipitation determines whether the global sea level rises or falls."  Isn't this the conclusion of this manuscript? Or are you saying here that this is your hypothesis?  Or is there past documentation that this is the case?  Maybe you could say something like, "Here we quantify how the ansatz of the precipitation determines whether the global sea level rises or falls".

*We use: "Here we test our hypothesis that the ansatz (assumption used to describe a phenomenon temporary to solve a problem) of the precipitation determines whether the global sea level rises or falls."*

Line 130: Even though "ansatz" is the perfect word for your context, it should be defined for the reader. There is not much benefit to having the reader not understand. I suggest this is defined either in the Abstract or the Introduction (where forcing is discussed). Also, your conclusions (perhaps in a new

conclusion section) should again use word "ansatz" to summarize what your extensive set of experiments has found. Currently, it is only used once in the entire manuscript, which is a shame – you should take advantage of such a perfect term in your discussion and conclusion.

*Thanks for this encouraging comment because I/we have thought to drop it entirely, but now we keep it and improve the text as suggested. Please, see also former comment.*

Line 131:  Like 2), 1) should start with "We", so something like "We utilize both the temperature and the precipitation anomalies from CMIP5 models on top of the reference background distributions (see Table 2) that were used to drive the ice-sheet model during spin-up."

*We replace the text as suggested.*

Line 134:  Please state again for 2) that the anomalies are placed on top of the background climate.

*We add: " Also, the second set of anomalies are added to the reference fields (see Table 2)."*

Line 136-137:  "In some cases, negative temperature scaling is considered unrealistic (Frieler et al., 2012)."  I might be missing something here, but it is not clear to me how this statement is relevant. Could you be more specific in the text how this statement follows?

*Clarified: "In these pure thermodynamical frameworks, negative temperature scaling is unexpected (Frieler et al., 2012); however,atmospheric dynamics may not dominate, which questions the usage of a constant scaling across Antarctica."*

Line 143:  Please specify "Beyond 2100", instead of "Afterwards"

*Done.*

Line 143:  Please rephrase "to keep the natural variability."  The intended logic is not clear to the reader.

*We hope to have this issue addressed by the rephrased paragraph.*

Line 144-147: "we use either" – You use this phrasing a number of times, but it sounds like you choose either one or the other (due to some criteria or randomly). What you actually mean is that you run one set of runs with the first 50 years and then another set with the second 50 years (as a variation on your ensemble).   You also say "Additionally, the number of scenarios is twice as large, since the mean states of the first and last 50 years show in general marginal differences. Anomaly forcing is computed relative to either the first or last 50 years of the control run. In the following, the first 50 years act generally as reference." Which I think says that you run a set of experiments for each of the first and second 50 years, and that for each set of anomalies, they are put on top of whichever is the background climate for the 1850-2100 runs. The way it is worded presently is awkward and unclear, and I do not think it effectively illustrates your point. Please try to make this clearer in the text and rephrase. I think this wording was, in part, why Reviewer 1 could not compute how many runs were actually conducted in the ensemble.

*The entire paragraph is rephrased. Also, we reduce the usage of "either." Please inspect the text for details.*

Line 149:  Instead of "triggers", please use "would be expected to trigger"

*Replaced.*

Line 153: Before, "Our simulations do not reflect this ongoing warming" please add something similar to "Note well," or an equivalent to highlight this sentence for the reader.

*Thanks for proposing this extension.*

Line 154:  After "Also," please be clear about why you are including this discussion. Something like: "Also, over longer timescales, there are feedbacks that are not captured by our simulations, for instance …"

*Integrated as suggested.*

Line 159: Please reiterate in this last sentence why it is true. For example: "Therefore, since only 21st century climate conditions are used to force the ensemble after 2100, our ensemble of ice sheet simulations beyond this year should not be considered a projection."

*Thanks for the suggestion, which we use.*

Line 167:  Are these identical because there is no meltwater runoff occurring (that is, you are only showing the precipitation ultimately?), and the fact that the surface topography is the same?  It is not clear to me what point is being made by including this statement in the text. Can you elucidate in the text if it is indeed important to include here?

*We move part of the text addressing the surface mass balance into the new subsection "Surface Mass Balance" as part of the Methods section after the introduction of the Parallel Ice Sheet Model. It also motivates the comparison. You find: "The surface mass balance (SMB) is computed via the PDD method, where the hydrological year starts on day 91. The PDD factor for snow and ice are 0.3296 cm(IE) Kelvin−1 day−1 and 0.8792 cm(IE) Kelvin−1 day−1, respectively. The temporal evolving annual 2m-air temperature standard deviation is derived from daily CMIP5 model values for each CMIP5 model at each ice sheet model grid-cell.*

*The reference data set (Table 2) drives three special ice sheet control runs "control 1", "control 2," and "control 3" (Table 3).These are performed to check whether a disturbance occurs when we replace the SMB used during the spin-up. The simulation named "control 1" is a continuation of the spin-up, where the SMB (Figure A16a) equals the precipitation from the reference data set (Table 2). The utilization of the PDD approach provides the SMB in "control 2". In the simulation "control 3", the SMB is computed via PDD and considers a potential height difference between the reference data set and the evolving ice sheet surface (Figure A16b). For the height difference, we consider a lapse rate of −7 K km−1. Since the height difference is zero at the beginning of this test, it initially does not influence the SMB. However, a lowering ice sheet surface in progressing simulations increases the air temperature used to compute the SMB via PDD. All the SMB distributions ("control 1" to "control3") are numerically identical across Antarctica (Figure A16c) because Antarctica is too cold to experience melting via PDD (Figure 2a; A detailed analysis of the climate follows below in section 3.1: "CMIP5 Forcing Data Set"). Therefore, the altered computation does not trigger any signal, while the via PDD computed SMB allows for melting under a warming climate."*

Materials and Methods:  I suggest splitting this section up into subsections (see comments below), and moving some details from the results into this section. That is, this methods section should describe all of your experiments, just not the main experiments. The extra sensitivity experiments (i.e. the basal melting sensitivity test description) should at least be named or listed in an

organized way here, and then you can refer to them later in your results section.

*We have added a table of all simulation being part of the ice sheet model ensemble.*

Line 215-Line 221:  Please specify "ocean temperatures" instead of just temperatures throughout this paragraph. Also, because this is a description of your forcing, it should probably be included in the Materials and Methods and not the Results.

*As indicated above, we have added a leading ocean or atmosphere to any occurrence of the word temperature to avoid ambiguity. However, we disagree with your suggestion to move the analysis of the processed forcing into the Material and Methods section because then we would have to move the scaling analysis for the same reason. Therefore, we would like to keep the following string of results: Analysis of climate subtracted from the CMIP5 data set, mean temperature-scaling of the precipitation found across Antarctica, deduced scaling across defined regions for different scenarios and individual models, the relationship between use precipitation method and ice sheet thickness changes, and finally the relation between the precipitation method and its impacts on the simulated sea-level contribution. We agree to move other parts, as suggested in the remaining review.*

Line 233-245:  This first paragraph describes the methods for temperature scaling. Since your methods section discusses two points:  the model forcing and the ice-sheet model setup, I urge the authors to have two sections within Materials and Methods, one to discuss the forcing, and including these scaling equations, and another section to discuss the ice sheet model. Breaking up the section would improve readability.

*We agree that it is best to separate these parts. Done.*

Line 249: Should this be "(last 50 years)"?  Since below you compare these results against if you replace it with the first 50 years (line 250)? Or does this mean if you replace it with the full transient forcing of the first 50 years (instead of the average) the results do not change? Please check the wording of these sentences with respect to the first/last 50 years since the current version is confusing.

*We're sorry for being unclear. We clarify: "To test the result's robustness, we exchange the baseline: beginning of the historical period instead of the first 50~years of piControl. Both the first 50 years of piControl and the historical forcing (1850–1899) start (probably) from the same state but are subject to diverging forcing, e.g., in atmospheric greenhouse gases and volcanic events (such as Krakatau in 1883; Henderson and Henderson, 2009). Despite replacing the baseline, the values change only slightly. Since the results are very similar when exchanging the baseline, we restrict the analysis to anomalies relative to the first 50 years of the piControl climate and consider the results robust.."*

Line 295-Line 315, and Line 349-356:  These paragraphs are examples of those that could be moved to the current Conclusion section, which I suggest you change to a "Discussion" section.

*Move as suggested into the new section "Discussion."*

Sections 3.5-3.6:  These sections are also more Discussion, since they put together results from a wide range of your figures. I suggest separating these into a discussion section, and allowing the discussion section to have sub-sections so that it is organized into your thoughtful topic areas. In this case, the parts of the current conclusion could be a final subsection to the

Discussion (that is one could be "Limitations" which are more appropriately included in the Discussion not conclusion), allowing the actual Conclusion to be a much shorter summary of the important findings.

*We agree and, therefore, have split the former section "Conclusion" into "Discussion" and "Conclusion" and added a subsection "Limitations" in "Discussion".*

Line 470: "Anyhow" is informal. Please use a different phrase here.

*Replaced by "Nevertheless."*

Line 518: "corrected" should be used instead of "correct"

*Done*

Table 1: Again, the statement "the first or last 50 years" makes it sounds like it is either one or the other, instead of both (that is, I am looking for a column in your table to tell me which one of those you chose). The same for "PISM1Eq (Figure A10) or PISM2Eq (Figure A11)". Please rephrase to make it clear that both are done in all of these cases.

*The revised table caption of table 1 and the table caption of the new table 3 should have address this issue.*

Line 633: Some concluding words about what this means for uncertainty in projections would strengthen your conclusion. For example, something similar to or expanding upon "The discrepancy of the simulation results between both methods describing the precipitation illustrates the uncertainty of the possible range of future precipitation growth in a warming atmosphere."

*Thanks for this suggestion which we take up.*

Table 1: In addition, "e.g. section 3.2: "Precipitation scaling"" – this is a perfect example of the misplacement of the description of the scaling. In this case, in describing all the experiments, the references should point to sections of the Methods, not to the Results.

*Addressed by the reorganization of the entire manuscript.*

Table 4: A reference to Fig. 1 would be appropriate in this caption, since your regions are defined nicely in that figure.

*Added as suggested.*

Figure 11, A13: It is not clear what distinguishes an ice shelf as "fringing". Please specify in the text or use more appropriate terminology.

*We drop "fringing" in these figure captions.*

Figure 12: Please reference a specific section or location instead of a reference to "please see text for details".

*Corrected all unclear cross references in captions.*

Section A3: This section does not really qualify as "Additional Discussion". Perhaps it should be an Appendix B for additional methods?

*We have reworked the section heads in the appendix.*

Line 974: "the first 30 years of the transient historical period (1850–2005)".

This is confusing the way it is written. It would be clearer to specify (1850-1879) as the exact 30 years you refer to. Also, please clarify if you mean the transient forcing or the average over this period.

*Rephrased: "If we alternatively compute the anomalies relative to averaged first 30~years of the historical period (1850—1879)."*

Line 976:  Reference here the location (Section) of the manuscript where you describe the quantification of uncertainties due to the driving model. Also please specify that you mean the driving model for calculating precipitation forcing (Is that the case, actually?). Or do you mean the Earth System Model? Actually, I am not positive which model you mean here.

*We have extended the text to address the issue: "The choice of the baseline (first or last 50~years of piControl or first 30~years of the historical period) to compute the scaling distribution is of minor importance. However, selecting the forcing data set from the pool of CMIP5 models determines the scaling distribution overwhelmingly. The across Antarctica averaged scaling factors reveal that the scatter range for one model is much smaller than the scaling values' distance among models (Figure 5). "*

Line 985:  Please reference here the figure that illustrates this point.

*We add "(see scatter range in Figure 5)"*

Line 1005:  What does the term "trends" mean in this context?  Please specify in the text or use different term.

*Rephrased: "For individual ensemble members the temporal evolution of the calving rate is ..."*

Line 1015-1017:  Please specify the years that correspond to each estimate.

*We added the period covered by the reported estimates.*

Line 1031: "However, only for MIROC-ESM the reference state" is awkward. Please rephrase, that is: "However, the reference state only matters for MIROC-ESM…"

*Corrected.*

Line 1032: "not negligible" => "a non-negligible"

*Corrected.*

Line 1034: "amounts" => "amount to"

*Replaced.*

Line 1040: "The temporal evolution of the actual basal melting rate (Figure A13) increases until 2100 and falls back afterward onto the value of the year 2071 because we apply the last 30-years-forcing recurrently after 2100." Is awkwardly phrased. One suggestion:  "The basal melting rates increase until 2100, but then suddenly decrease back to 2071 values, since by experimental design, the last 30 years of forcing (2071-2100) is repeated after year 2100."  - or something similar for clarity.

*Thanks for the suggestion which we adopt.*

Line 1061: "with a lower strength" => "with lower emissions", maybe?  "Strength" used in this context is vague.

*We use: "Scenarios with reduced radiative forcing reach the minimum later, ..."*

Line 1128:  This seems like a conclusion that could be included in the main text, and stated in a similar manner.

*It has been moved into the newly created section "Discussion."*

Figure A5, A6, A7:  Please specify "the year 2000", Is the "the simulated sea level for each individual simulation at the year 2000," or something similar.

*Rephrased as suggested.*

*Please note that the next pages highlight the differences between the former and the revised manuscript.*

X

**Future Sea Level Contribution  from Antarctica  inferred from CMIP5 Model Forcing  and its Dependence on Precipitation Ansatz**

Christian B. Rodehacke[1,2], Madlene Pfeiffer[1], Tido Semmler[1], Özgür Gurses[1], and Thomas Kleiner[1]

[1]Alfred Wegener Institute Helmholtz Centre for Polar and Marine Research, D-27570 Bremerhaven, Germany
[2]Danish Meteorological Institute, DK-2100 Copenhagen Ø, Denmark

**Correspondence:** Christian Rodehacke (christian.rodehacke@awi.de)

**Abstract.** Various observational estimates indicate growing mass loss at Antarctica's margins but also heavier precipitation across the continent. Simulated future projections reveal that heavier precipitation, fallen on Antarctica, may counteract amplified iceberg discharge and increased basal melting of floating ice shelves driven by a warming ocean. Here, we  test how the ansatz (implementation in a mathematical framework) of the precipitation boundary condition shapes Antarctica's sea-level contribution in an ensemble of ice-sheet simulations. We test two precipitation conditions. We either apply the precipitation anomalies coming from CMIP5 models directly or scale the precipitation by the air temperature anomalies from the CMIP5 models. In the scaling approach, it is common to use a relative precipitation increment per degree warming as an invariant scaling constant. From nine CMIP5 models, we use future climate projections, ranging from strong mitigation efforts to business-as-usual, to  perform simulations from 1850 to 5000. We take advantage of individual climate projections by exploiting its full temporal and spatial structure. The CMIP5 projections beyond 2100 are prolonged with reiterated forcing that includes decadal variability; hence, our study may underestimate ice loss past 2100. In contrast to various former studies ~~, only the historical (1850–2005) and scenario (2006–2100) forcing drive our ensemble of simulations , which neglects unavoidable continuous warming consistent with the higher climate scenarios beyond the year 2100. We test how the precipitation boundary condition determines Antarctica's sea-level contribution. The spatially and temporally varying climatic forcing drives ice-sheet simulations, such that our ensemble represents all climate patterns, which is fundamentally different from using spatial means as forcing~~ that apply an evolving temporal forcing averaged spatially across the entire Antarctic Ice Sheet, our simulations consider the spatial structure in the forcing coined by various climate patterns. This fundamental difference reproduces regions of decreasing precipitation despite general warming. Regardless of the applied boundary and forcing conditions, our ensemble study suggests that some areas will lose ice in the future, such as the glaciers from the West Antarctic Ice Sheet draining into the Amundsen Sea. In general, the simulated ice-sheet thickness grows  along the coast, where incoming storms deliver topographically controlled precipitation.  There the ice thickness differences are largest between the applied precipitation  methods. On average  , Antarctica shrinks for all future scenarios if the  air temperature anomalies scale the

precipitation. In contrast, Antarctica gains mass in our simulations if we apply the simulated precipitation anomalies  directly. The analysis reveals that the mean scaling inferred from climate models is larger than the commonly used values deduced from ice cores; besides, it varies spatially: Highest scaling across the East Antarctic Ice Sheet  and lowest scaling around the Siple Coast  east of the Ross Ice Shelf. The discrepancies in response to both precipitation ansatzes illustrate the principal uncertainty in projections of Antarctica's sea-level contribution.

**Plain Language Summary** In the warmer future, the ice sheet of Antarctica will lose more ice at the margin, because more icebergs may calve and the warming ocean melts more floating ice shelves from below. However, the hydrological cycle is also stronger in a warmer world. As a consequence, more snowfall precipitates on Antarctica, which may balance the amplified marginal ice loss. In this study, we have used future climate scenarios from various global climate models to perform numerous ice-sheet simulations. These simulations represent the Antarctic Ice Sheet. We analyze whether Antarctica will grow or shrink. In all our simulations, we find that certain areas will lose ice under all circumstances. However, depending on the method used to describe the precipitation reaching Antarctica in our simulations, parts of the Antarctic Ice Sheet may either grow or shrink in the future. The discrepancy of the simulation results between both methods describing the precipitation illustrates the uncertainty of the possible range of future precipitation growth in a warming atmosphere. Furthermore, the dissimilarity is pronounced differently between the West Antarctic and East Antarctic Ice Sheet. Since we use only the available climate scenarios until the year 2100, any additional warming after 2100 may turn the ice gain into an ice loss under a strongly changing climate.

**1 Introduction**

[revised manuscript text omitted]

$$q = q(t_{ref}) = \frac{F_{ref}(t_{ref})}{F_{org}(t_{ref})}. \tag{D2}$$

1405 The corrected flux $F_{cor}$ using Equation D1 is defined as

$$F_{cor}(t) = r(t) \cdot F_{ref}(t_{ref}), \tag{D3}$$

so that the flux difference $\Delta F(t)$ is

$$\begin{aligned} \Delta F(t) &= F_{cor}(t) - F_{org}(t) \\ &= F_{org}(t) \left[ \frac{F_{ref}(t_{ref})}{F_{org}(t_{ref})} - 1 \right]. \end{aligned}$$

1410 With Equation D2 we obtain

$$\Delta F(t) = F_{org}(t) \left[ q - 1 \right]. \tag{D4}$$

To relate the sea-level change to the ice mass evolution, we define the ratio $p(t)$ of the sea level temporal deviation to the ice mass temporal deviation as

$$p(t) = \frac{\dfrac{dz_l(t)}{dt}}{\dfrac{dm_{ice}(t)}{dt}}, \tag{D5}$$

1415 where $z_l$ is the sea level and $m_{ice}$ the total ice mass, which includes grounded and floating ice. We use here $p = \mathrm{median}(p(t))$ so that each ensemble member is characterized by one value for its entire time series. If $p = \frac{1}{\rho A_{oce}}$, $100\,\%$ of flux difference (Equation D4) contributes immediately to the sea level of the global ocean with an area of $A_{oce}$.

The total ice mass ($m_\text{ice}$) changes are driven by four terms

$$\frac{m_\text{ice}}{dt} = \underbrace{\left[ F^{SMB}(t) + F^G(t) \right]}_{\text{unchanged under correction}} + F^B(t) + F^D(t),$$

where $F^{SMB}(t)$ is the surface mass balance flux and $F^G(t)$ the basal mass flux of grounded ice (Figure A9). We assume that these two terms in the brackets do not change regardless of the applied corrections to the last two terms $F^B$ and $F^D$. Hence the difference in the ice mass change is

$$\Delta \frac{dm_\text{ice}}{dt} = \Delta F^B(t) + \Delta F^D(t) \tag{D6}$$

Now we relate the temporal evolution of the sea level to the total ice mass changes by utilizing the Equation D5

$$\frac{z_{l\text{cor}}}{dt} = \frac{z_{l\text{org}}}{dt} + p(t) \cdot \left[ \Delta F^B_\text{cor}(t) + \Delta F^D_\text{cor}(t) \right],$$

so that we obtain:

$$z_{l\text{cor}} = z_{l\text{org}} + \Delta z_l(t), \tag{D7}$$

where the sea-level difference $\Delta z_l(t)$ is

$$\Delta z_l(t) = \int_{t_0}^{t} p(\hat{t}) \left[ \Delta F^B_\text{cor}(\hat{t}) + \Delta F^D_\text{cor}(\hat{t}) \right] d\hat{t}. \tag{D8}$$

The Figures A5 and A7 depict the sea-level difference for two cases. If the additional mass loss contributes immediately to a rising sea level (Figure A7), the corresponding sea level rise of $30\,\text{cm}$ would be larger than the actual sea level rise since 1850 of about $20\,\text{cm}$ (Church and White, 2011).This case is not realistic because a melting floating ice shelf does not impact the sea level. Only the flow of grounded ice across the grounding line, to feed an ice shelf, or the direct loss of grounded ice contributes to the sea level.

In contrast, the sea level changes hardly (Figure A5) if the deduced ratio $\overline{p(t)}$, which corresponds to the ratio defined in Equation D5. It is computed for each ensemble mean as median of its time series. Whether the ratio between ice loss and sea-level rise is constant under amplified basal melting of ice shelves is an open question. Strongly intensified ocean-driven ice loss will probably cause a retreating grounding line on a longer time scale, which ultimately releases grounded ice into the sea and increases the sea level.

Figure A8 shows the proportion of the deduced ratio to the $100\,\%$ ratio. Only very few ensemble member lose about 15% of the maximum value of $p = 1/(\rho \cdot A_\text{oce})$. In contrast, the mean and median value of this proportion is generally less then $5\,\%$. For all ensemble members driven by the precipitation anomaly, this proportion is on average $4.7\,\%$ with a median of $3.9\,\%$. It is even lower for ensemble members driven by the temperature-scaled precipitation. The median amounts $0.7\,\%$ and the corresponding mean is $0.9\,\%$. Please note that some ensemble members under the temperature-scaled precipitation are subject to a negative scaling. This result confirms the above presented low positive and negative scaling seen for restricted regions (Figure 5). It highlights also that simulations driven by temperature-scaled precipitation could show unexpected results.

**D1**

**Appendix E:** **Appendix Figures**

[Figure]

**Figure A1.** Atmospheric (a, b) and oceanographic (c) reference forcing;  CMIP5 data set maximum anomalies (d–l). The top row represents the reference fields to spin-up the ice-sheet model (Table 2). The 2m-air temperature (a) and the total precipitation (b) are mean fields from the regional RACMO model, while the ocean temperatures come from the World Ocean Atlas 2009 (c). Below each reference field, the related maximum anomalies are compiled for the period 2071–2100. Here, the second (third and fourth) row shows the anomalies for RCP8.5 (RCP4.5, RCP2.6). The dark-gray line follows the current coastline. All potential ocean temperatures (c, f, i, l) are a vertical mean of the depth interval from $150\,\mathrm{m}$ to $500\,\mathrm{m}$. The white contour lines in the anomaly plots highlight the following thresholds. 2m-air temperature (d, g, j): $8^\circ$C; total precipitation (e, h, k) $50\,\mathrm{cm\,year^{-1}}$. All these anomalies are the  CMIP5 data set maximum of the models listed in Table 1; CCSM4 is not part of RCP2.6. Figure 2 shows the corresponding  mean fields, where the white contour line in the precipitation field corresponds to $30\,\mathrm{cm\,year^{-1}}$.

[revised manuscript text omitted]